# Learning Correlated Reward Models: Statistical Barriers and Opportunities

**Yeshwanth Cherapanamjeri**
MIT

**Constantinos Daskalakis**
MIT

**Gabriele Farina**
MIT

**Sobhan Mohammadpour**
MIT

## Abstract

Random Utility Models (RUMs) are a classical framework for modeling user preferences and play a key role in reward modeling for Reinforcement Learning from Human Feedback (RLHF). However, a crucial shortcoming of many of these techniques is the Independence of Irrelevant Alternatives (IIA) assumption, which collapses *all* human preferences to a universal underlying utility function, yielding a coarse approximation of the range of human preferences. On the other hand, statistical and computational guarantees for models avoiding this assumption are scarce. In this paper, we investigate the statistical and computational challenges of learning a *correlated* probit model, a fundamental RUM that avoids the IIA assumption. First, we establish that the classical data collection paradigm of pairwise preference data is *fundamentally insufficient* to learn correlational information, explaining the lack of statistical and computational guarantees in this setting. Next, we demonstrate that *best-of-three* preference data provably overcomes these shortcomings, and devise a statistically and computationally efficient estimator with near-optimal performance. These results highlight the benefits of higher-order preference data in learning correlated utilities, allowing for more fine-grained modeling of human preferences. Finally, we validate these theoretical guarantees on several real-world datasets, demonstrating improved personalization of human preferences.

## 1 Introduction

Random Utility Models (RUMs) (Train, 2009) are the leading framework for modelling human preferences. Classical applications dating back to the 1960s include mathematical psychology (Luce, 1959), transportation science (Ben-Akiva, 1973), econometrics (McFadden, 1980), and marketing (Rust and Zahorik, 1993). More recently, they are a core component of the Reinforcement Learning from Human Feedback (RLHF) pipeline (Christiano et al., 2017), used to align Large Language Models (LLMs) with human preferences. Unfortunately, many classical RUMs adopt the Independence of Irrelevant Alternatives (IIA) assumption. In the context of LLMs, the IIA posits a universal underlying utility function for all users of a system. A growing body of work highlights the limits of this approach (Bai et al., 2022; Casper et al., 2023; Sorensen et al., 2024; Anwar et al., 2024; Conitzer et al., 2024) in capturing the full range of human preferences.

Formally, a RUM models utilities over a set of $n$ items as a high-dimensional random vector, $X$[1]. A classical choice is the *logit* model, where each component $X_i$ is independent and distributed according to a Gumbel distribution $X_i \sim \text{Gumbel}(\mu_i)$. The IIA assumption implies that for any unobserved item (or response), the model assigns the same average utility irrespective of the user's past preferences.

In this paper, we investigate the statistical challenges of learning correlated utility models, focusing in particular on the correlated probit model where $X \sim \mathcal{N}(\mu, \Sigma)$. By explicitly modeling correlations, these techniques allow past user behavior to inform future responses, providing a more accu-

---

[1]For RLHF, the items correspond to (prompt, response) pairs

rate representation of human preferences. However, in contrast to the logit model, little is known about the statistical and computational challenges of learning these models: i) *What data is needed to learn them?* and ii) *How many samples do we need?*

We make key advancements on several fronts: 1) We show that the conventional paradigm of pairwise preference data is fundamentally insufficient to learn these models, 2) On the other hand, three-way-preference data is both necessary and sufficient and finally, 3) Validate the efficacy of higher-order preference data in accurately modelling correlated user preferences. Our results highlight fundamental drawbacks in conventional data collection pipelines and suggest deliberate amendments to address these shortcomings. Prior to our work, even *identifiability* results, disregarding any statistical considerations, did not exist. Hence, we establish the *first* identifiability guarantees and complement them with a near-optimal polynomial-time estimator.

## 2 RELATED WORK

A wealth of ideas exists for modeling rewards. However, we focus on choice modeling, where subjective *preferences* between items are expressed. Consequently, we find a treatment of reward models like step-wise reward models (Havrilla et al., 2024) and process reward models (Luo et al., 2023) beyond the scope of this work. We are interested in models that can make *counterfactual* choices from preference data. Outside of machine learning, choice models have been successfully applied to planning problems like public transport scheduling (Wei et al., 2022), effect of price markups (Gallego and Wang, 2014), airline scheduling (Wei et al., 2020), EV charging station placement (Lamontagne et al., 2023), and toll placement Osorio and Atasoy (2021).

*Random Utility Models (RUMs)* are among the most common models of human behavior in mathematical psychology, econometrics, transportation, and marketing (e.g., see McFadden, 2001, a Nobel Prize lecture on RUMs). RUMs impose structure on the choice model by assuming that the agents are rational, meaning that when confronted with a finite set of *alternatives*, they choose the one that maximizes their (latent) utility. The agent's unobserved utility is commonly modeled as a random variable to acknowledge the heterogeneity of utilities. Formally, an agent is assumed to have a random utility vector $X$, where $X_i$ is the utility of choosing some alternative $i$ from the set of all possible alternatives $\mathcal{C}$, and, when presented with a subset $\mathcal{R} \subseteq \mathcal{C}$ of alternatives, the agent chooses the alternative $I_{\mathcal{R}}$ satisfying

$$I_{\mathcal{R}} \in \arg\max_{i \in \mathcal{R}} X_i.$$

Given observations of the form $(\mathcal{R}^t, I_{\mathcal{R}^t}^t)$, $t = 1, \ldots, T$, a common challenge is to *learn* the distribution of the latent $X$. Obtaining an estimate $\tilde{X}$ of $X$ not only gives us an estimate $\tilde{I}_{\mathcal{R}}$ of $I_{\mathcal{R}}$ for any subset of alternatives $\mathcal{R}$, but it also allows making predictions about the welfare of $\mathcal{R}$, $G(\mathcal{R}) := \max_{i \in \mathcal{R}} X_i$. It is important to note that learning the distribution of $X$ from preference data is inherently underconstrained as adding the same potentially stochastic value to all components of $X$ yields no change in the distribution of $I$.

The most popular RUM is the *logit* (Train, 2009, Chapter 3) or also sometimes referred to as Luce-Shephard-McFadden model, or, in the binary case (i.e. $|\mathcal{C}| = 2$), the Bradley-Terry model. The logit is the only RUM that satisfies *independence of irrelevant alternatives (IIA)* (Luce, 1959), that is, the ratio of the probabilities of choosing two alternatives is independent of the presence of other alternatives. The logit enjoys many useful properties, for instance, it can be learned using pairwise comparisons, it can learn any distribution over the set of full alternatives[2], and is easy to calculate via a softmax. Unfortunately, if the utilities are correlated, the logit cannot be used to make correct counterfactual predictions. A classical example showcasing this impossibility is the red bus/blue bus problem introduced by Debreu (1960). If half of the populations prefer red buses over blue buses and half of the population prefers red buses made by company A over those made by company B, then IIA implies that 2/3 of the population prefers red buses made by A or B over blue buses, a contradiction. In practice, as shown by Benson et al. (2016), many datasets do not satisfy the IIA assumption. We continue our discussion on choice modeling in Appendix A

---

[2]by setting the utilities to the log probabilities

## 3 PROBLEM SETUP, AND THE NEED FOR BEST-OF-THREE OBSERVATIONS

Here, we avoid the restrictive IIA assumption and assume, instead, that the utilities follow the classical probit model with $X \sim \mathcal{N}(\mu, \Sigma)$. We aim to recover $(\mu, \Sigma)$ from only the *choices* $\arg\max_{\{i \in \mathcal{R}\}} X_i$ made for various subsets $\mathcal{R} \subset [n]$. For binary choice, $\mathcal{R}$ is pairs of elements $(i, j) \in [n]^2$. Similarly, for the three-way choice setting, $\mathcal{R}$ is triplets $(i, j, k) \in [n]^3$. Note that $(\mu, \Sigma)$ are not fully identifiable in this setting: for any $X \sim \mathcal{N}(\mu, \Sigma)$, $X'$ defined as follows, induces the same ranking probabilities for *any* $t > 0$, $t\left(X - \frac{1}{n}\sum_i X_i\right)$. Hence, we adopt the following necessary normalization $\mu, \Sigma$.

**Assumption 3.1.** For $X \sim \mathcal{N}(\mu, \Sigma)$, we assume, without changing the choice distributions, that

$$\langle \mu, \mathbf{1} \rangle = 0, \quad \Sigma \mathbf{1} = 0, \text{ and } \quad \text{Tr}(\Sigma) = n.$$

Consequently, $X$ lives on the hyperplane $\mathbf{1}^\top X = 0$. Furthermore, we assume $\Sigma$ has rank $n - 1$.

Unfortunately, the classical paradigm of pairwise comparisons ($|\mathcal{R}| = 2$) is insufficient for recovering $(\mu, \Sigma)$, even accounting for Assumption 3.1. Its proof is deferred to Appendix E.2.

**Theorem 3.2.** *For any $n \geqslant 3$ and $\mu, \Sigma$ satisfying Assumption 3.1, there exists an infinite set $\mathcal{S}$:*

$$\forall i, j \in [n], \mu', \Sigma' \in \mathcal{S} : \Pr_{X \sim \mathcal{N}(\mu, \Sigma)} \{X_i \geqslant X_j\} = \Pr_{X \sim \mathcal{N}(\mu', \Sigma')} \{X_i \geqslant X_j\}.$$

Theorem 3.2 indicates that pairwise comparison data is fundamentally unable to account for correlations in the probit model. While the full proof of Theorem 3.2 is somewhat involved, the particular setting with $\mu = 0$ is instructive. Here, observe that for *any* pair of distinct alternatives, $i, j$, the probability that $X_i \geqslant X_j$ is $1/2$ *irrespective* of the value of $\Sigma$. Hence, correlational information is fundamentally impossible to learn given only access to pairwise comparisons.

## 4 IDENTIFIABILITY

Theorem 3.2 prompts the natural question: *can higher order preference data help?* We answer the question in the affirmative and show that best-of-three observations (Theorem 4.4) are sufficient *and* from Theorem 3.2, necessary to recover $\mu, \Sigma$. Observing that:

$$\mathbb{P}\{X_i \geqslant X_j \geqslant X_k\} = \mathbb{P}\{X_j \geqslant X_k\} - \mathbb{P}\{X_j \geqslant X_i, X_k\}, \quad \text{(RANK-PROB)}$$

we may assume access to three-way *ranking* probabilities.

### 4.1 THE CASE OF THREE ALTERNATIVES

As a stepping stone towards identifying probit models with an arbitrary number $n$ of alternatives, we focus on the case $n = 3$ and establish the following theorem.

**Theorem 4.1.** $(\mu, \Sigma)$ *are uniquely identifiable from the three-way observation probabilities.*

This result will serve as the basis for the general case in Subsection 4.2. A probit model satisfying the constraints in Assumption 3.1 defines a *bivariate* normal distribution that lies in the plane defined by $\mathbf{1}^\top X = 0$. We start by projecting the utilities onto a lower-dimensional space:

$$\begin{bmatrix} V_1 \\ V_2 \end{bmatrix} := \underbrace{\begin{bmatrix} 1/\sqrt{6} & 1/\sqrt{6} & -2/\sqrt{6} \\ 1/\sqrt{2} & -1/\sqrt{2} & 0 \end{bmatrix}}_{=: P} \begin{bmatrix} X_1 \\ X_2 \\ X_3 \end{bmatrix} \quad \leftrightarrow \quad \begin{bmatrix} X_1 \\ X_2 \\ X_3 \end{bmatrix} = \underbrace{\begin{bmatrix} 1/\sqrt{6} & 1/\sqrt{2} \\ 1/\sqrt{6} & -1/\sqrt{2} \\ -2/\sqrt{6} & 0 \end{bmatrix}}_{=: P^\top} \begin{bmatrix} V_1 \\ V_2 \end{bmatrix}. \quad (1)$$

Under the transformation $P$, the probit model $X \sim \mathcal{N}(\mu, \Sigma)$ is mapped to some bivariate normal distribution $V \sim \mathcal{N}(\dot{\mu}, \dot{\Sigma})$. Upon identification of the parameters $\dot{\mu}$ and $\dot{\Sigma}$ in two dimensions, we recover the original parameters $\mu$ and $\Sigma$ through the transformation:

$$\mu = P^\top \dot{\mu}, \quad \text{and} \quad \Sigma = P^\top \dot{\Sigma} P.$$

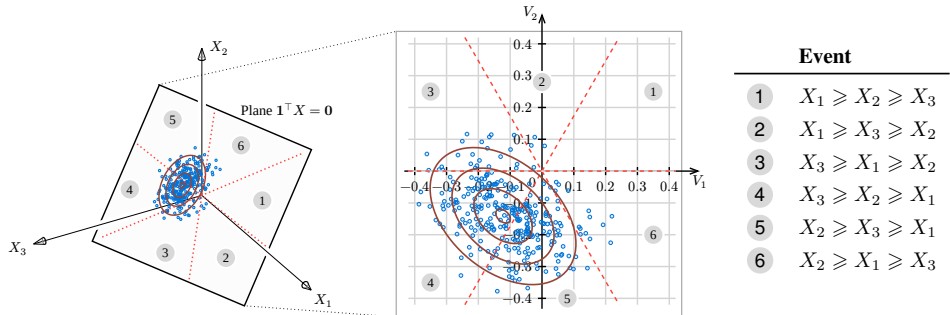

Figure 1: The probabilities $\mathbb{P}\{X_i \geqslant X_j \geqslant X_k\}$, for permutations, $(i, j, k)$, of $\{1, 2, 3\}$, correspond to the probability mass in each of the six slices of the plane denoted ① through ⑥.

Defining,

$$c_1 := \begin{bmatrix} 0 \\ 1 \end{bmatrix}, \qquad c_2 := \begin{bmatrix} \sqrt{3}/2 \\ -1/2 \end{bmatrix}, \qquad c_3 := \begin{bmatrix} \sqrt{3}/2 \\ 1/2 \end{bmatrix},$$

the probability of the events $\{X_i \geqslant X_j\}$, are satisfy the following:

$$\mathbb{P}\{X_1 \geqslant X_2\} = \mathbb{P}\{c_1^\top V \geqslant 0\}, \quad \mathbb{P}\{X_2 \geqslant X_3\} = \mathbb{P}\{c_2^\top V \geqslant 0\}, \quad \mathbb{P}\{X_1 \geqslant X_3\} = \mathbb{P}\{c_3^\top V \geqslant 0\},$$

with each three-way permutation corresponding to the intersections of the corresponding halfspaces. The projection and the partitioning of the 2-dimensional space are illustrated in Figure 1.

We will further transform $V$, rendering the distribution *isotropic*. Define $\dot\Sigma^{-1/2}$ as *a* solution of

$$\dot\Sigma^{-1/2}\dot\Sigma(\dot\Sigma^{-1/2})^\top = I.$$

Note that $\dot\Sigma^{-1/2}$ is unique up to an orthonormal transformation on the left. We will fix a convenient choice subsequently. For now, observe that $\dot\Sigma^{-1/2}$ and $\dot\Sigma^{1/2} := ((\dot\Sigma^{-1/2})^{-1})^\top$ satisfy:

$$\forall x_1, x_2 \in \mathbb{R}^2 : \langle x_1, x_2 \rangle = \langle \dot\Sigma^{1/2} x_1, \dot\Sigma^{-1/2} x_2 \rangle.$$

Setting $x_1 = c_i$ and $x_2 = V$, we now observe:

$$\langle \dot\Sigma^{1/2} c_i, \dot\Sigma^{-1/2} V \rangle \text{ with } \dot\Sigma^{-1/2} V \sim \mathcal{N}(\widetilde\mu, I) \text{ where } \widetilde\mu := \dot\Sigma^{-1/2}\dot\mu.$$

Lastly, note that:

$$\forall i \in [3] : \langle c_i, V \rangle \geqslant 0 \iff \langle \widetilde{c}_i, \dot\Sigma^{-1/2} V \rangle \geqslant 0 \text{ where } \widetilde{c}_i := \frac{\dot\Sigma^{1/2} c_i}{\|\dot\Sigma^{1/2} c_i\|}.$$

Note that $\dot\Sigma^{-1/2} V$ is isotropic. Defining,

$$\alpha_i := \widetilde{c}_i^\top \widetilde\mu \text{ and } \alpha_{ij} := \widetilde{c}_i^\top \widetilde{c}_j,$$

Our first technical result identifies $\alpha_i$ from observational data.

**Lemma 4.2.** *The quantities $\alpha_i$ are identifiable from three-way ranking probabilities.*

*Proof.* Note that

$$\mathbb{P}\{c_i^\top V \geqslant 0\} = \mathbb{P}\Big\{\langle \Sigma^{-1/2} V, \widetilde{c}_i \rangle \geqslant 0\Big\} = \mathbb{P}_{g \sim \mathcal{N}(0,1)}\{g + \langle \widetilde{c}_i, \widetilde\mu \rangle \geqslant 0\} = \Phi(\langle \widetilde{c}_i, \widetilde\mu \rangle).$$

As $\Phi(\cdot)$ is a strictly increasing function, $\langle \widetilde{c}_i, \widetilde\mu \rangle$ is recoverable from the observables. □

In the next lemma, we use the values $\alpha_i$ from Lemma 4.2 to establish a monotonic relationship between $\langle \widetilde{c}_i, \widetilde{c}_j \rangle$ and the probability of the events in Figure 1. However, the precise choice of events depends on the $\alpha_i$. Note that for any distinct $i, j \in [3]$, there exist signings $s_i, s_j \in \{\pm 1\}$ such that $s_i \langle \widetilde{c}_i, \widetilde\mu \rangle, s_j \langle \widetilde{c}_j, \widetilde\mu \rangle \geqslant 0$ and the events $\Big\{s_i \langle \widetilde{c}_i, \dot\Sigma^{-1/2} V \rangle, s_j \langle \widetilde{c}_j, \dot\Sigma^{-1/2} V \rangle \leqslant 0\Big\}$ correspond to combinations of one or more events in Figure 1 and is hence, observable. By instantiating the subsequent lemma with the signed vectors $s_i \widetilde{c}_i, s_j \widetilde{c}_j$, we recover the $\alpha_{ij}$ by relating it to the probability of the event $\Big\{s_i \langle \widetilde{c}_i, \dot\Sigma^{-1/2} V \rangle, s_j \langle \widetilde{c}_j, \dot\Sigma^{-1/2} V \rangle \leqslant 0\Big\}$. Its proof is deferred to Appendix E.1.

**Lemma 4.3.** *Let $v_1, v_2 \in \mathbb{R}^2$ be two independent unit vectors and $\xi \in \mathbb{R}^2$ be such that $d_1 := \langle v_1, \xi \rangle, d_2 := \langle v_2, \xi \rangle \geqslant 0$. Then,*

$$\alpha_{12} := \langle v_1, v_2 \rangle$$

*is identifiable from $d_1, d_2$, and*

$$\gamma_{12} := \mathbb{P}\{\langle X, v_1 \rangle, \langle X, v_2 \rangle \leqslant 0\} \text{ for } X \sim \mathcal{N}(\xi, I).$$

*Proof of Theorem 4.1.* We will show that $\dot{\Sigma}^{1/2}$ is recoverable from the observation probabilities. $\dot{\Sigma}$ is then obtainable from the fact that $\dot{\Sigma} = (\dot{\Sigma}^{1/2})^\top \dot{\Sigma}^{1/2}$. From Lemma 4.2, we recover $\alpha_i = \langle \widetilde{c}_i, \widetilde{\mu} \rangle$ for $i \in [3]$ and from Lemma 4.3, we recover $\alpha_{ij} = \langle \widetilde{c}_i, \widetilde{c}_j \rangle$ for $i \neq j \in [3]$. Without loss of generality, we assume that for some $s, t > 0$ by relaxing the scale constraint $\mathrm{Tr} \, \dot{\Sigma} = 1$:

$$\dot{\Sigma}^{1/2} c_1 = \begin{bmatrix} 1 \\ 0 \end{bmatrix}, \quad \dot{\Sigma}^{1/2} c_2 = s \begin{bmatrix} \alpha_{12} \\ \sqrt{1 - \alpha_{12}^2} \end{bmatrix}, \quad \dot{\Sigma}^{1/2} c_3 = t \begin{bmatrix} \alpha_{13} \\ \sqrt{1 - \alpha_{13}^2} \end{bmatrix}. \tag{2}$$

The signs of the second component of the vector are determined by fixing the sign of $\dot{\Sigma}^{1/2} c_2$ (this is an orthonormal transformation) and for $\dot{\Sigma}^{1/2} c_3$, follows from the fact that $c_3 = c_1 + c_2$. Noting that $\alpha_{13} \neq \alpha_{12}$ (as otherwise, we obtain permutation observations with 0 probability), we obtain:

$$\begin{bmatrix} 1 \\ 0 \end{bmatrix} + s \begin{bmatrix} \alpha_{12} \\ \sqrt{1 - \alpha_{12}^2} \end{bmatrix} = t \begin{bmatrix} \alpha_{13} \\ \sqrt{1 - \alpha_{13}^2} \end{bmatrix} \implies \begin{bmatrix} 1 \\ 0 \end{bmatrix} = \begin{bmatrix} \alpha_{13} & \alpha_{12} \\ \sqrt{1 - \alpha_{13}^2} & \sqrt{1 - \alpha_{12}^2} \end{bmatrix} \begin{bmatrix} t \\ -s \end{bmatrix}.$$

Since the above forms an invertible system, we obtain the values of $s, t > 0$. This now enables the recovery of $\dot{\Sigma}$ (and consequently, $\Sigma$) via the invertible system:

$$\dot{\Sigma}^{1/2} \cdot \begin{bmatrix} c_1 & c_2 \end{bmatrix} = \begin{bmatrix} 1 & s\alpha_{12} \\ 0 & s\sqrt{1 - \alpha_{12}^2} \end{bmatrix}$$

To recover $\dot{\mu}$, note that $\widetilde{c}_1$ and $\widetilde{c}_2$ form a basis for $\mathbb{R}^2$ and observing that these are determined by Eq. (2), we identify $\widetilde{\mu}$. Noting that $\dot{\Sigma}^{-1/2} \dot{\mu} = \widetilde{\mu}$, we recover $\dot{\mu}$ and $\mu$ by our previous discussion. $\qquad \square$

## 4.2 IDENTIFIABILITY: MORE THAN THREE ALTERNATIVES

We now utilize Theorem 4.1 to establish identifiability for an arbitrary number of alternatives, $n$. Theorem 4.1 recovers $\mu, \Sigma$ restricted to tuples of 3 alternatives up to the symmetries in Assumption 3.1. However, the sub-matrices of $\mu, \Sigma$ (or more precisely, $\mu$ and $\Sigma$ restricted to these alternatives) could potentially violate these assumptions. Hence, we establish that there is a *unique* pair, $\mu, \Sigma$, consistent with the equivalence class of parameters identified by Theorem 4.1 for all choices of 3 alternatives. In what follows, we will use $\bar{\mu}_{ijk}$ and $\bar{\Sigma}_{ijk}$ to denote the parameters $\mu$ and $\Sigma$ restricted to the tuple $i, j$ and $k$, projected onto the two-dimensional subspace orthogonal to $\mathbf{1}$.

**Theorem 4.4.** *$\mu, \Sigma$ are uniquely identifiable from the three-way observation probabilities.*

*Proof.* First, note for any distinct indices $i, j, k$, Theorem 4.1 yields $\widetilde{\mu}_{ijk}, \widetilde{\Sigma}_{ijk}$ satisfying:

$$t_{ijk} \widetilde{\mu}_{ijk} = \bar{\mu}_{ijk} \text{ and } t_{ijk}^2 \widetilde{\Sigma}_{ijk} = \bar{\Sigma}_{ijk}$$

for some scaling $t_{ijk} > 0$. Denoting $c_{ij} = e_i - e_j$, we have:

$$c_{ij}^\top t_{ijk}^2 \widetilde{\Sigma}_{ijk} c_{ij} = c_{ij}^\top \bar{\Sigma}_{ijk} c_{ij} = c_{ij}^\top \Sigma c_{ij} \implies t_{ijk}^2 = \frac{c_{ij}^\top \Sigma c_{ij}}{c_{ij}^\top \widetilde{\Sigma}_{ijk} c_{ij}}.$$

The first step uses the fact that $c_{ij}^\top \bar{\Sigma}_{ijk} c_{ij} = c_{ij}^\top \Sigma_{ijk} c_{ij}$. Since scaling $\Sigma$ (and $\mu$) produces the same solution, we may assume without loss of generality that $t_{123} = 1$. Noting that this allows the recovery of $c_{12}^\top \Sigma c_{12}$, the above equality enables determination of the $t_{12k}$ for all $k \in [n]$. This further enables the determination of $t_{1jk}$ for all distinct $k, j \in [n]$. Another application of the argument finally yields the values of $t_{ijk}$ for all distinct $i, j, k \in [n]$.

Furthermore, observe that for all distinct $i, j, k \in [n]$ with $\sigma_{ij} := \Sigma_{ij}$:

$$c_{ij}^\top t_{ijk}^2 \widetilde{\Sigma}_{ijk} c_{ij} = c_{ij}^\top \Sigma c_{ij} = \sigma_{ii} + \sigma_{jj} - 2\sigma_{ij}.$$

Furthermore, noting that $c_{ii} = \mathbf{0}$, we determine $\alpha_{ij} = c_{ij}^\top \Sigma c_{ij}$ for all $i, j \in [n]$. Observe that

$$\sum_{j=1}^n c_{ij}^\top \Sigma c_{ij} = n\sigma_{ii} + \mathrm{Tr}(\Sigma) - 2\sum_{j=1}^n \sigma_{ij} = n\sigma_{ii} + \mathrm{Tr}(\Sigma).$$

where the last step is due to fact that $\mathbf{1}^\top \mu = 0, \Sigma \mathbf{1} = \mathbf{0}$. By further summing over $i$, we get

$$\sum_{i=1}^n \sum_{j=1}^n c_{ij}^\top \Sigma c_{ij} = n\,\mathrm{Tr}(\Sigma) + n\,\mathrm{Tr}(\Sigma)$$

which now enables recovery of $\mathrm{Tr}(\Sigma)$ and from the above discussion $\sigma_{ii}$ and consequently, $\sigma_{ij}$ for all $i, j$. Hence, we have identified $\Sigma$. For the mean, observe again the following:

$$c_{ij}^\top (t_{ijk}\widetilde{\mu}_{ijk}) = c_{ij}^\top \bar{\mu}_{ijk} = \mu_i - \mu_j.$$

The last step is valid because all of the elements of $\mu_{ijk}$ were shifted by the same amount to obtain $\bar{\mu}_{ijk}$, and consequently, their difference did not change. Since $t_{ijk}$ are determined, this determines $\mu_1 - \mu_j$ for all $j$ (note the choice of $\mu_1$ is arbitrary). Letting $\beta_j = \mu_1 - \mu_j$, we have:

$$\sum_{j=1}^n \beta_j = n\mu_1 - \sum_{j=1}^n \mu_j = n\mu_1.$$

This determines $\mu_1$ and, consequently, the remaining $\mu_j$. This concludes the proof as $(\mu, \Sigma)$ have been recovered up to the universal symmetries previously described. $\qquad\square$

## 5 Finite-sample Guarantees: Upper and Lower Bounds

In this section, we extend our *identifiability* results Section 4 to *estimation*. The key conceptual advancement over Section 4 is in the aggregation procedure which combines the standardized estimators of the 3 item setting (Subsection 4.1) into a solution for the $n$ item setting (Subsection 4.2). The procedure described in Subsection 4.2 is wasteful as it requires estimating *all* $3 \times 3$ sub-matrices of $\Sigma$, resulting in a total of $O(n^3)$ total $3 \times 3$ matrices. We show that this can be substantially improved upon using a much smaller number ($\widetilde{O}(n^2)$) of sub-matrices in Theorem 5.2. We then establish a statistical *lower bound* demonstrating the near-optimality of our estimator. We start with our estimation guarantees. In addition to Assumption 3.1, we require the following assumption:

**Assumption 5.1.** (Observability) For any three alternatives $i, j, k \in [n]$ and $X \sim \mathcal{N}(\mu, \Sigma)$:

$$\mathbb{P}\{i > j > k\} \geqslant \gamma$$

for some $\gamma > 0$.

Intuitively, Assumption 5.1 avoids worst-case scenarios where an overwhelming difference in utilities renders a choice unobservable, making some parameters impossible to estimate. We now state our main estimation guarantee for the general $n$-item setting.

**Theorem 5.2.** *Let $(\mu^*, \Sigma^*)$ satisfy Assumptions 3.1 and 5.1 with $\mathrm{Tr}(\Sigma^*) = n$. Then, for any $\varepsilon, \delta \in (0, 1/2]$, given there is a polynomial time algorithm which when allowed $N$ observations of rank-3 permutations from $\mathrm{Choice}(\mu^*, \Sigma^*)$ of its choice, returns estimates, $(\overline{\mu}, \overline{\Sigma})$, satisfying:*

$$\|\overline{\mu} - \mu^*\|_\infty \leqslant \varepsilon \text{ and } \|\overline{\Sigma} - \Sigma^*\|_\infty \leqslant \varepsilon$$

*with probability at least $1 - \delta$ as long as $N \geqslant Cn^2\varepsilon^{-2}\gamma^{-24}\log(n/\delta)\log^6(n/(\gamma\varepsilon))$.*

Next, we present a lower bound that shows that Theorem 5.2 is near-optimal. We introduce some notation before proceeding. An estimator, $T$, is parameterized by a set of tuples $\{((i, j, k), N_{ijk}) : i, j, k \in [n], i \neq j \neq k \neq i\}$. The estimator then obtains $N_{ijk}$ samples from $\mathrm{Choice}(\mu_{ijk}^*, \Sigma_{ijk}^*)$ for each $i, j, k$ and outputs estimates $(\overline{\mu}, \overline{\Sigma})$. Furthermore, we denote:

$$N_{ij} := \sum_k N_{ijk}.$$

As before, we adopt Assumption 3.1 for normalization and state our main lower bound below.

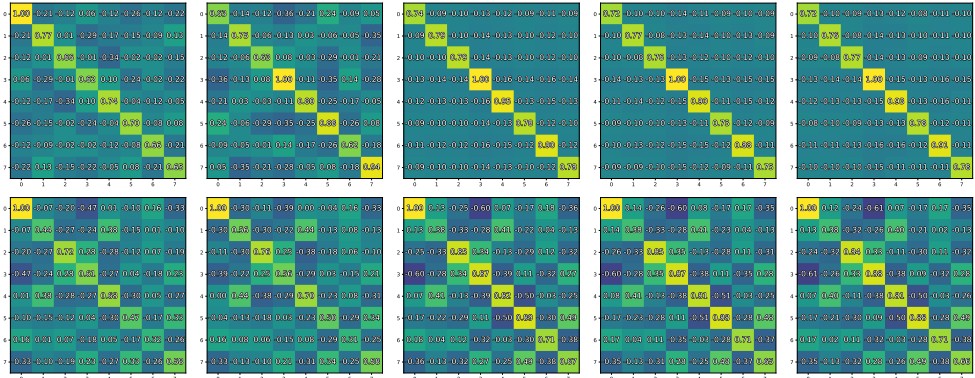

Figure 2: Center: ground truth covariance matrix, two figures on the right, covariance matrix learned from best-of-three observations, two figure on the left, covariance matrix learned from pairwise comparisons. We reproduce larger version these plots in Appendix F.

**Theorem 5.3.** *Let* $n \in \mathbb{N}$, $\delta, \varepsilon \in (0, 2^{-4})$. *For any estimator $T$ parameterized by* $\{((i, j, k), N_{ijk})\}_{i \neq j \neq k \neq i}$ *such that $N_{i^*, j^*} \leqslant \varepsilon^{-2} \log(1/\delta)/4$ for some $i^* \neq j^* \in [n]$, there exist two choice models, parameterized by $(\mathbf{0}, \Sigma^1)$ and $(\mathbf{0}, \Sigma^2)$ with:*

$$\mathrm{Tr}(\Sigma^1) = \mathrm{Tr}(\Sigma^2) = n \text{ and } \|\Sigma^1 - \Sigma^2\|_\infty \geqslant \varepsilon,$$

*satisfying:*

$$\max_{\Sigma \in \{\Sigma^1, \Sigma^2\}} \mathbb{P}_{\boldsymbol{X} \sim (T, \mathrm{Choice}(\mathbf{0}, \Sigma))} \left\{ \|T(\boldsymbol{X}) - \Sigma\|_\infty \geqslant \frac{\varepsilon}{4} \right\} \geqslant \delta.$$

We pause for a few remarks. Firstly, note that each pair $(i, j)$ must appear in at least $\Omega(\varepsilon^{-2})$ experiments. Since each sample only accounts for $O(1)$ pairs, this result implies a lower bound of $\Omega(n^2)$ on the number of unique experiments for any successful estimator matching Theorem 5.2. Finally, note that this argument also implies that the total number of samples is at least $\Omega(n^2 \varepsilon^{-2} \log(1/\delta))$ again matching Theorem 5.2 in terms of $n, d,$ and $\delta$.

## 6 EXPERIMENTS

We empirically evaluate modeling correlations. We use datasets of ratings and rankings. In rating datasets users are asked to assign a numerical value to alternatives. In contrasts, ranking datasets contain the ordered list of preferences. We convert rankings to ratings and vice versa by using the index as rating and assuming alternatives with higher ratings are preferred over alternatives with lower ratings. The data gathering process can either be organic i.e., users rate movies they watch, or structured, every user ranks all food items on the menu.

We use two types of models on these datasets: RUMs and matrix completion models, both trained with gradient based methods. For RUMs, we use logit and probit models learned from two way comparisons and probit models learned from three way comparisons. In all cases we use the maximum likelihood estimate. RUMs do not take the user as input and directly create a model for the population. Matrix completion (Koren et al., 2009) learns the ratings for all users in the training set, we then use the counts of users in the training set to approximate the probability of certain events. For sanity checks, we include a model we refer to as direct, which is similar to matrix completion, except we compute the desired quantities directly on the training set. We argue that this model is unrealistic in some scenarios, for instance where the set of alternative and user is large or when the ratings are grouped but anonymous. In particular, we find that matrix completion needs many observations per user to learn the distribution of choices. Still, since these matrix completion models the whole population, we expect them to be able to learn correlated preferences.

In the first set of experiments we sample six choices from a user, we present the ranking of four of the alternatives as context and predict the preference over the remaining two alternatives. We start with synthetic data sampled from probit model as we can compare our results to the ground

| structure | | logit accuracy quantile | | | matrix completion accuracy quantile | | | probit (pairwise) accuracy quantile | | | probit (best-of-three) accuracy quantile | | |
|---|---|---|---|---|---|---|---|---|---|---|---|---|---|
| $\mu$ | $\Sigma$ | 0.25 | 0.50 | 0.75 | 0.25 | 0.50 | 0.75 | 0.25 | 0.50 | 0.75 | 0.25 | 0.50 | 0.75 |
| 0 | I | 0.50 | 0.50 | 0.50 | 0.50 | 0.50 | 0.50 | 0.50 | 0.50 | 0.50 | 0.50 | 0.50 | 0.50 |
| 0 | bin | 0.50 | 0.50 | 0.50 | 0.78 | 0.78 | 0.78 | 0.45 | 0.50 | 0.54 | 0.79 | 0.79 | 0.79 |
| 0 | rI | 0.50 | 0.50 | 0.50 | 0.50 | 0.50 | 0.51 | 0.50 | 0.50 | 0.50 | 0.50 | 0.50 | 0.50 |
| 0 | r | 0.50 | 0.50 | 0.50 | 0.67 | 0.67 | 0.67 | 0.47 | 0.51 | 0.55 | 0.67 | 0.67 | 0.67 |
| r | I | 0.70 | 0.70 | 0.71 | 0.68 | 0.70 | 0.70 | 0.66 | 0.66 | 0.67 | 0.68 | 0.70 | 0.70 |
| r | bin | 0.79 | 0.79 | 0.80 | 0.94 | 0.95 | 0.95 | 0.95 | 0.95 | 0.95 | 0.94 | 0.95 | 0.95 |
| r | rI | 0.72 | 0.72 | 0.72 | 0.71 | 0.71 | 0.72 | 0.69 | 0.70 | 0.71 | 0.70 | 0.70 | 0.72 |
| r | r | 0.63 | 0.63 | 0.64 | 0.71 | 0.71 | 0.71 | 0.67 | 0.68 | 0.69 | 0.71 | 0.71 | 0.71 |

Table 1: Accuracy of different methods on the synthetic dataset. Notice how the direct method matches best-of-three probit.

| dds. | var. | feat. | logit accuracy quantile | | | matrix completion accuracy quantile | | | direct accuracy quantile | | | probit (pairwise) accuracy quantile | | | probit (best-of-three) accuracy quantile | | |
|---|---|---|---|---|---|---|---|---|---|---|---|---|---|---|---|---|---|
| | | | 0.25 | 0.50 | 0.75 | 0.25 | 0.50 | 0.75 | 0.25 | 0.50 | 0.75 | 0.25 | 0.50 | 0.75 | 0.25 | 0.50 | 0.75 |
| jokes | onehot | onehot | 0.61 | 0.61 | 0.61 | 0.61 | 0.61 | 0.61 | 0.61 | 0.62 | 0.63 | 0.59 | 0.59 | 0.59 | 0.61 | 0.61 | 0.61 |
| ml | 1k | llm | 0.57 | 0.57 | 0.57 | 0.59 | 0.59 | 0.60 | 0.57 | 0.57 | 0.58 | 0.56 | 0.56 | 0.56 | 0.57 | 0.57 | 0.57 |
| ml | 10k | llm | 0.59 | 0.59 | 0.59 | 0.60 | 0.60 | 0.60 | 0.60 | 0.60 | 0.60 | 0.57 | 0.57 | 0.57 | 0.59 | 0.59 | 0.59 |
| ml | 50k | llm | 0.59 | 0.59 | 0.59 | 0.58 | 0.58 | 0.58 | 0.59 | 0.59 | 0.60 | 0.55 | 0.55 | 0.56 | 0.59 | 0.59 | 0.59 |
| ml | 1k | onehot | 0.62 | 0.62 | 0.62 | 0.60 | 0.60 | 0.60 | 0.57 | 0.58 | 0.58 | 0.60 | 0.60 | 0.60 | 0.61 | 0.61 | 0.62 |
| ml | 10k | onehot | 0.61 | 0.61 | 0.61 | 0.60 | 0.60 | 0.60 | 0.58 | 0.59 | 0.59 | 0.59 | 0.59 | 0.59 | 0.61 | 0.61 | 0.61 |
| ml | 50k | onehot | 0.59 | 0.59 | 0.59 | 0.58 | 0.58 | 0.58 | 0.58 | 0.58 | 0.59 | 0.55 | 0.55 | 0.56 | 0.59 | 0.59 | 0.60 |
| nf | 10k | llm | 0.59 | 0.59 | 0.59 | 0.59 | 0.59 | 0.59 | 0.56 | 0.56 | 0.57 | 0.58 | 0.58 | 0.58 | 0.59 | 0.59 | 0.59 |
| nf | 100k | llm | 0.61 | 0.61 | 0.61 | 0.61 | 0.61 | 0.61 | 0.61 | 0.61 | 0.61 | 0.59 | 0.59 | 0.59 | 0.61 | 0.61 | 0.61 |
| nf | 150k | llm | 0.61 | 0.61 | 0.61 | 0.60 | 0.60 | 0.60 | 0.60 | 0.60 | 0.61 | 0.56 | 0.56 | 0.56 | 0.61 | 0.61 | 0.61 |
| nf | 10k | onehot | 0.61 | 0.61 | 0.62 | 0.59 | 0.59 | 0.59 | 0.57 | 0.57 | 0.57 | 0.59 | 0.59 | 0.60 | 0.61 | 0.61 | 0.61 |
| nf | 100k | onehot | 0.62 | 0.62 | 0.62 | 0.61 | 0.61 | 0.61 | 0.61 | 0.61 | 0.61 | 0.59 | 0.59 | 0.59 | 0.62 | 0.62 | 0.62 |
| nf | 150k | onehot | 0.61 | 0.61 | 0.61 | 0.60 | 0.60 | 0.61 | 0.59 | 0.60 | 0.60 | 0.56 | 0.56 | 0.56 | 0.61 | 0.61 | 0.61 |
| sushi | B | default | 0.65 | 0.65 | 0.65 | 0.66 | 0.66 | 0.66 | 0.51 | 0.51 | 0.51 | 0.64 | 0.64 | 0.64 | 0.65 | 0.65 | 0.65 |
| sushi | A | onehot | 0.65 | 0.66 | 0.66 | 0.67 | 0.67 | 0.68 | 0.68 | 0.68 | 0.68 | 0.65 | 0.65 | 0.65 | 0.68 | 0.68 | 0.68 |
| sushi | B | onehot | 0.67 | 0.67 | 0.68 | 0.67 | 0.68 | 0.68 | 0.50 | 0.50 | 0.50 | 0.66 | 0.66 | 0.66 | 0.67 | 0.67 | 0.68 |

Table 2: Prediction accuracy for different model and datasets. The column labeled "var." denotes the variants. For movie datasets, nf (Netflix) and ml (MovieLens) the number denotes the number of ratings a movie needed to be included. For the sushi dataset, the dataset is split into 2 categories. The column labeled "feat." denotes the features fed to the model. Rows marked with llm use the embedding of `Qwen3-Embedding-0.6B` (Zhang et al., 2025).

truth. In Figure 2, we show the recovered covariance matrices with pairwise and best-of-three observations using a probit RUM. We can see that the model learned from pairwise comparisons learns nonexistent correlations in the uncorrelated case while in the correlated case, initialization can play an important role as apparent by the change in value in the first cell of the second row. We report the performance in Table 1. The location $\mu$ can either be zero or sampled randomly, the covariance matrix can either be the identity matrix I, diagonal random rI, plus minus one bin, and random r. In all cases the best-of-three probit matches the performance direct method (with simulation).

For real data, we use the data from a sushi preference dataset (Kamishima, 2003), the eigen-taste joke dataset (Goldberg et al., 2001), Netflix prize (Netflix, Inc., 2006), and the MovieLens dataset (Harper and Konstan, 2015). The sushi dataset is the only ranking dataset. It's A variant is the only dataset where every user rank every alternative. The results are presented in Table 2. We observe that training with best-of-three observations lead to great improvement over probit models trained with best-of-two observations. The mean effect are strong in these datasets and the improvement over the logit is modest. Table 3 lists some movies with strong correlations in the Netflix and MovieLense datasets. In particular we note that we find sequels like Spider-Man (2002) and Spider-Man 2 (2004) or Kill Bill Vol. 1 (2003) and Kill Bill Vol. 2 (2004) to have strong positive correlations while we find a strong negative correlation between more critically acclaimed movies like Memento (2000) and Eternal Sunshine of the Spotless Mind (2004) compared to Hollywood comedy and blockbusters like Perl Harbor (2001) or Cheaper by the Dozen (2003). In Figure 7, we see that sea urchins are very divisive and that a preference for cucumber sushi negatively correlates with preference for toro (fatty tuna) sushi.

We finish this section with a welfare maximization experiment. We evaluate the welfare maximizing subset of alternatives given a fixed size, i.e., solve the following equation for some fixed $N$, $\max_{\mathcal{R} \subseteq \mathcal{C} : |\mathcal{R}| = N} \mathbb{E}\left[\max_{i \in \mathcal{R}} X_i\right]$. We use the models from the previous experiments to generate wel-

| Netflix | | | | | |
|---|---|---|---|---|---|
| | | cor. | | | cor. |
| Independence Day (1996) | Lost in Translation (2003) | -0.45 | Spider-Man (2002) | Spider-Man 2 (2004) | 0.56 |
| Maid in Manhattan (2002) | The Royal Tenenbaums (2001) | -0.42 | Adaptation (2002) | Lost in Translation (2003) | 0.57 |
| Miss Congeniality (2000) | The Matrix (1999) | -0.41 | Kill Bill: Vol. 1 (2003) | Kill Bill: Vol. 2 (2004) | 0.57 |
| Memento (2000) | Pearl Harbor (2001) | -0.41 | Fight Club (1999) | The Usual Suspects (1995) | 0.59 |
| Double Jeopardy (1999) | Eternal Sunshine of the Spo... | -0.40 | Lord of the Rings: The Retu... | Lord of the Rings: The Two ... | 0.64 |
| Cheaper by the Dozen (2003) | Memento (2000) | -0.40 | Finding Nemo (Widescreen) (... | Shrek (Full-screen) (2001) | 0.65 |
| Independence Day (1996) | The Royal Tenenbaums (2001) | -0.40 | American Beauty (1999) | Being John Malkovich (1999) | 0.66 |
| Anchorman: The Legend of Ro... | Speed (1994) | -0.40 | Lord of the Rings: The Fell... | Lord of the Rings: The Two ... | 0.66 |
| MovieLens | | | | | |
| Much Ado About Nothing (1993) | Truth About Cats & Dogs, Th... | -0.46 | Braveheart (1995) | Little Women (1994) | 0.53 |
| Much Ado About Nothing (1993) | What's Eating Gilbert Grape... | -0.42 | Crow, The (1994) | Muriel's Wedding (1994) | 0.53 |
| Dangerous Minds (1995) | Snow White and the Seven Dw... | -0.42 | Clerks (1994) | Father of the Bride Part II... | 0.54 |
| Mrs. Doubtfire (1993) | Nightmare Before Christmas,... | -0.39 | Johnny Mnemonic (1995) | Robin Hood: Men in Tights (... | 0.55 |
| Little Women (1994) | Seven (a.k.a. Se7en) (1995) | -0.39 | Congo (1995) | Speed (1994) | 0.55 |
| Ace Ventura: When Nature Ca... | Little Women (1994) | -0.39 | City Slickers II: The Legen... | Little Women (1994) | 0.59 |
| Congo (1995) | Get Shorty (1995) | -0.38 | Father of the Bride Part II... | GoldenEye (1995) | 0.61 |
| Johnny Mnemonic (1995) | Much Ado About Nothing (1993) | -0.38 | Hudsucker Proxy, The (1994) | True Romance (1993) | 0.61 |

Table 3: Left to right, movies with the lowest and highest correlations in a probit learned from the Netflix (top) and MovieLens (bottom) datasets. A table with more movie pair is produced in Appendix F.

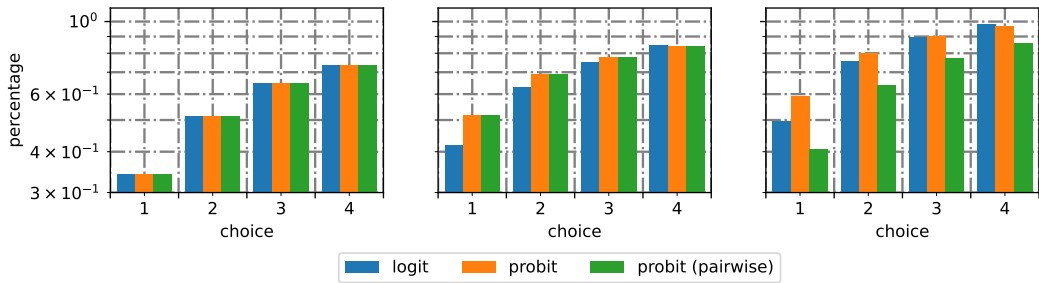

Figure 3: Preference of the welfare maximizing choices, left to right for $N = 1, 2, 3$.

fare maximizing subsets of size 1 to 3 in the sushi-A dataset. Figure 3 illustrates what percentage of the test population have their $i$th preferred alternative in the set, e.g., the bar with $x = 4$ corresponds to the ratio of people who have at least their third favorite alternative in the set. We acknowledge that this quantity does not directly correlate with welfare as welfare is sensitive to the magnitude of utility change whereas this plot is not. It might not make significant utility difference for someone if their second or third choice are present but it affects this plot. Nonetheless, we see that for subsets of size 2 and 3, more people have their preferred sushi on the menu. The optimal size one menu, for all RUMs, is toro (fatty tuna), the optimal size 2 menu according to the logit is toro (fatty tuna) and maguro (tuna) while our probit suggests uni (sea urchin) and maguro (fatty tuna) instead. Looking at the output of the model in Figure 7 we see that the utility of uni (sea urchin) has high variance and correlates negatively with the utility of toro (fatty tuna) and maguro (tuna). This difference in menu highlights the problem with not modeling correlations properly, a menu of two alternatives that are high utility is not much better than one with only one alternative if they are very correlated. The probit learned from pairwise comparisons instead suggests anago (eel) and toro (fatty tuna) instead.

## 7    CONCLUSIONS AND FUTURE DIRECTIONS

In this paper, we investigated the methodological shortcomings of learning correlated reward models from simple preference data. We established the statistical deficiency of pairwise comparisons for identifying correlation and showed that best-of-three preferences are provably sufficient. Our analysis culminated in a statistically and computationally efficient estimator for this setting, complete with finite-sample guarantees. Ultimately, our findings show that modeling preference correlations is not only feasible but essential for enabling more advanced applications like in-context learning and the strategic sampling of alternatives.

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

## A   PRIOR WORK

There has been considerable interest in scaling probit estimation, with recent techniques extending the method from ten alternatives (e.g., Natarajan et al., 2000) to one hundred (e.g., Ding et al., 2024). As we show in the next section these methods cannot actually recover the correlations properly.

The nested logit (NL) relaxes the IIA axiom by grouping the alternatives in the groups and assuming IIA between the elements of the same group or the groups themselves. In the cat, golden retriever,

and borzoi example, this can mean that first, the agents choose whether they prefer cats or dots and then choose between gold retrievers and borzois. In this structure, IIA holds between cats and dogs and between golden retrievers and borzois, but not between borzois and cats. In its generality, the NL is described by a tree whose leaves are the alternatives. To choose an alternative, an agent starts at the root and must select one of the current node's children using an IIA model. The agent repeats this process until they reach a leaf (alternative). Ben-Akiva (1973) showed that the welfare and choice probabilities of the NL can be calculated in a bottom-up fashion as the value of each node is the log-sum-exp of the values of its children. Benson et al. (2016) show that at least $\Omega(n^2)$ statistical tests are required to recover the structure of the NL.

The mixed logit McFadden and Train (2000) relaxes the IIA axioms by assuming that there exists an unobserved utility that would validate IIA. Since we do not observe these utilities, we have to marginalize them. The mixed logit is defined as

$$\mathbb{P}\left\{I_{\mathcal{R}} = i\right\} = \mathbb{P}\left\{\forall_j u_i + \epsilon_i + A_i \geqslant u_i + \epsilon_i + A_j\right\} = \int \mathrm{sm}_\tau(u + A)_i \, \mathrm{d}\,\mathbb{P}\{A\}.$$

In practice, the integral is evaluated using numerical methods, and the mixed logit is learned using MLE. McFadden and Train (2000) showed that if they take the limit of $\tau$ to 0, this model converges to a RUM with utility $A$ because the softmax converges to the indicator function, returning one when $A_i \geqslant A_j$ for all $j$ and zero otherwise.

Over the years, many choice models have been developed and RUMs are by no means the only class of choice models. We can characterize them by

1. representation power, for instance, the logit cannot represent correlated choices,
2. number of parameters, while the logit has $n = |\mathcal{C}|$ parameters, the probit has $n(n + 3)/2$ parameters, and
3. whether they provide some notion of welfare.

*Representative agents models (RAM)*, are defined as

$$\mathbb{P}\left\{I_{\mathcal{R}} = i\right\} = \left\{\arg\max_{\pi \in \Delta(\mathcal{R})} \langle \pi, u_{\mathcal{R}} \rangle - \Omega_{\mathcal{R}}(\pi)\right\}_i,$$

for some utility vector $u$ restricted the set of alternative $\mathcal{R}$, some strictly convex regularizer $\Omega_{\mathcal{R}}$, and the probability simplex over $\mathcal{R}$ denoted by $\Delta(\mathcal{R})$. RAMs also provide an expected welfare function, but learning a regularizer for all possible sets of alternatives may not be trivial. Taking $\Omega$ to be the negative entropy yields the same model as the logit (Anderson et al., 1988). Generally, an equivalent RAM exists for every RUM but not vice-versa (Hofbauer and Sandholm, 2002). As noted by works like Sørensen and Fosgerau (2022), working with RAMs may be more convenient for mathematical purposes. We refer to Feng et al. (2017) for a class halfway between RAMs and RUMs.

Tversky (1972) introduces a different class of models, where each agent *splits* the set of alternatives in a desirable and an undesirable subset. While the model has an exponential number of parameters (in terms of the number of alternatives) and requires a different type of data, it is subset-consistent for all possible choice models.

A wealth of work exists on approximations for the probit model starting with Clark (1961). Clark approximates $\max(X, Y)$ with a normal distribution. We refer to Kamakura (1989) and Bunch and Kitamura (1991) for references to other approximation methods. We note that Bunch and Kitamura (1991) finds that none of the approximations consistently outperform others, and all have critical failure modes. Compared to simulation-based methods, one of the most significant weaknesses of this type of work is the inability to trade compute for accuracy.

Train (2009, Chapter 5) and Bunch (1991) mention several ways of removing the symmetries in estimating probit models and how to verify that the parametrization is "correct". Interestingly, they omit our reparametrization. McFadden (1980) claim that the primary issue with the probit is evaluating and differentiating the choice probabilities.

**Notation.** We use $\mathbb{R}$ and $\mathbb{N}$ to denote the set of real and natural numbers respectively. For $n \in \mathbb{N}$, $\mathbb{R}^n$ and $\mathbb{R}^{n \times n}$ denote the set of $n$-dimensional vectors and $n \times n$ dimensional matrices respectively.

Furthermore, $\mathbb{S}^n$ and $\mathcal{S}^{n \times n}$ correspond to the set of unit vectors and positive semi-definite matrices of dimension $n$. For $n \in \mathbb{N}$, $[n]$ is shorthand for $\{1, \ldots, n\}$. We denote the pdf and cdf of a standard Gaussian as $\phi$ and $\Phi$, respectively. For a matrix $A$, $\text{Tr}(A)$ and $\det(A)$ denote the trace and determinant of $A$. For two vectors $x$ and $y$ in $\mathbb{R}^n$, $\langle x, y \rangle$ is their inner product. We use TV and KL for the total variation and Kullback–Leibler divergence. Lastly, $\text{Choice}(\mu, \Sigma)$, for $\mu \in \mathbb{R}^n$ and $\Sigma \in \mathbb{R}^{n \times n}$, refers to the choice model defined by a RUM whose utilities follow a multivariate normal distribution $\mathcal{N}(\mu, \Sigma)$ whose mean is $\mu$ and covariance matrix is $\Sigma$.

## B    ESTIMATION WITH THREE ALTERNATIVES - PROOF OF THEOREM B.1

Here, we extend the arguments of Theorem 4.1 presented in Subsection 4.1 to the setting of estimation, where one aims to. Recall, our observability assumption from Section 5:

**Assumption 5.1.** (Observability) For any three alternatives $i, j, k \in [n]$ and $X \sim \mathcal{N}(\mu, \Sigma)$:

$$\mathbb{P}\{i > j > k\} \geqslant \gamma$$

for some $\gamma > 0$.

The main theorem of the section is stated below:

**Theorem B.1.** *Let $(\mu, \Sigma) \in \mathbb{R}^3 \times \mathcal{S}^{3 \times 3}$ satisfy Assumption 5.1 for some $\gamma > 0$ and $\varepsilon, \delta \in (0, 1/2]$. Then, there is an algorithm which when given $n$ i.i.d samples from $\text{Choice}(\mu, \Sigma)$, returns estimates, $(\widehat{\mu}, \widehat{\Sigma})$ satisfying for some $t > 0$:*

$$\|t\mu - \widehat{\mu}\| \leqslant \varepsilon$$
$$(1 - \varepsilon)t^2 \Sigma \preccurlyeq \widehat{\Sigma} \preccurlyeq (1 + \varepsilon)t^2 \Sigma$$
$$\text{Tr}(t^2 \Sigma), \text{Tr}(\widehat{\Sigma}) \geqslant \frac{1}{2}$$

*with probability at least $1 - \delta$ as long as $n \geqslant C \varepsilon^{-2} \gamma^{-20} \log(1/\delta) \log^2(1/(\gamma \varepsilon))$.*

We begin with some useful technical results. The first concerns the maximizer of the minimum of the Gaussian masses of two symmetric cones defined by the intersection of two half-spaces.

**Lemma B.2.** *Let $v_1, v_2 \in \mathbb{S}^2$. Then, we have:*

$$\underset{\mu}{\arg\max} \left( \min \left\{ \mathbb{P}_{X \sim \mathcal{N}(\mu, I)}(\langle X, v_1 \rangle, \langle X, v_2 \rangle \geqslant 0), \mathbb{P}_{X \sim \mathcal{N}(\mu, I)}(\langle X, v_1 \rangle, \langle X, v_2 \rangle \leqslant 0) \right\} \right) = \mathbf{0}.$$

*Proof.* Consider any $\mu \neq 0$. Note, that without loss of generality, we may assume:

$$v_1 = \begin{bmatrix} \sin(\theta) \\ -\cos(\theta) \end{bmatrix}, \; v_2 = \begin{bmatrix} \sin(\theta) \\ \cos(\theta) \end{bmatrix}.$$

for some $\theta \in [0, \pi/2]$. Now, observe:

$$\xi_1 := \mathbb{P}_{X \sim \mathcal{N}(\mu, I)}(\langle X, v_1 \rangle, \langle X, v_2 \rangle \geqslant 0) = \frac{1}{2\pi} \int_0^\infty \int_{-x\tan\theta}^{x\tan\theta} \exp\left\{ -\frac{(x - \mu_1)^2 + (y - \mu_2)^2}{2} \right\} dy dx$$

$$\xi_2 := \mathbb{P}_{X \sim \mathcal{N}(\mu, I)}(\langle X, v_1 \rangle, \langle X, v_2 \rangle \leqslant 0) = \frac{1}{2\pi} \int_0^\infty \int_{-x\tan\theta}^{x\tan\theta} \exp\left\{ -\frac{(-x - \mu_1)^2 + (y - \mu_2)^2}{2} \right\} dy dx$$

When $\mu_1 > 0$, $(x + \mu_1)^2 > x^2$ for all $x \geqslant 0$ and consequently:

$$\xi_2 \leqslant \frac{1}{2\pi} \int_0^\infty \int_{-x\tan\theta}^{x\tan\theta} \exp\left\{ -\frac{x^2 + (y - \mu_2)^2}{2} \right\} dy dx.$$

Similarly for $\mu_1 < 0$ with $\xi_1$ taking the place of $\xi_2$. Therefore, for any $\mu_2$:

$$\underset{\mu_1}{\arg\max} \min(\xi_1, \xi_2) = 0.$$

**Claim B.3.** We have for any $t > 0$:

$$\arg\max_{\mu} \int_{-t}^{t} \exp\left\{-\frac{(x-\mu)^2}{2}\right\} dx = 0.$$

*Proof.* First, define

$$f(\mu) := \int_{-t}^{t} \exp\left\{-\frac{(x-\mu)^2}{2}\right\} dx.$$

We now have:

$$\frac{d}{d\mu} f(\mu) = \int_{-t}^{t} (x-\mu) \exp\left\{-\frac{(x-\mu)^2}{2}\right\} dx$$

$$= \int_{(t+\mu)^2/2}^{(t-\mu)^2/2} \exp\{-u\} du = \exp\left\{-\frac{(t+\mu)^2}{2}\right\} - \exp\left\{-\frac{(t-\mu)^2}{2}\right\}.$$

Hence, we have:

$$\frac{d}{d\mu} f(\mu) \begin{cases} < 0 & \text{if } \mu > 0 \\ > 0 & \text{if } \mu < 0 \\ 0 & \text{otherwise} \end{cases}.$$

Hence, $\mu = 0$ is the unique minimizer of $f(\mu)$ concluding the proof of the claim. $\qquad\square$

From Claim B.3, we get for any $\mu_2 \neq 0$:

$$\xi_2 < \frac{1}{2\pi} \int_0^{\infty} \int_{-x\tan\theta}^{x\tan\theta} \exp\left\{-\frac{(x^2+y^2)}{2}\right\} dy dx$$

$$= \mathbb{P}_{X\sim\mathcal{N}(0,I)}\{\langle X, v_1\rangle, \langle X, v_2\rangle \geqslant 0\} = \mathbb{P}_{X\sim\mathcal{N}(0,I)}\{\langle X, v_1\rangle, \langle X, v_2\rangle \leqslant 0\}$$

concluding the proof of the lemma. $\qquad\square$

The following lemma establishes the probabilistic concentration properties required for our result with the parameter $\widetilde{\varepsilon}$ to be chosen subsequently.

**Lemma B.4.** *There exists a constant $C > 0$ such that:*

$$\forall v_1, v_2 \in \{\pm c_i\}_{i=1}^3 \text{ s.t } v_1 \not\parallel v_2 : \left|\frac{1}{n}\sum_{i=1}^n \mathbf{1}\{\langle X_i, v_1\rangle, \langle X_i, v_2\rangle \geqslant 0\} - \mathbb{P}_{X\sim\mathcal{N}(\mu,\Sigma)}\{\langle X, v_1\rangle, \langle X, v_2\rangle \geqslant 0\}\right| \leqslant \widetilde{\varepsilon}$$

$$\forall v \in \{\pm c_i\}_{i=1}^3 : \left|\frac{1}{n}\sum_{i=1}^n \mathbf{1}\{\langle X_i, v\rangle \geqslant 0\} - \mathbb{P}_{X\sim\mathcal{N}(\mu,\Sigma)}\{\langle X, v\rangle \geqslant 0\}\right| \leqslant \widetilde{\varepsilon}$$

*with probability at least $1 - \delta$ if $n \geqslant C\log(1/\delta)/\widetilde{\varepsilon}^2$.*

*Proof.* The lemma follows from Hoeffding's inequality (Theorem E.1) along with a union bound over the 18 events in the conclusion of the lemma. $\qquad\square$

The next lemma shows that permutation probabilities remain stable under mild perturbations of the center of a Gaussian.

**Lemma B.5.** *Let $\mu \in \mathbb{R}^2$ and $S$ be a set such that:*

$$\mathbb{P}_{X\sim\mathcal{N}(\mu,I)}\{X \in S\} \geqslant \gamma.$$

*Then, we have:*

$$\forall \mu' \in \mathbb{B}(\mu, \varepsilon) : \left|\mathbb{P}_{X\sim\mathcal{N}(\mu',I)}\{X \in S\} - \mathbb{P}_{X\sim\mathcal{N}(\mu,I)}\{X \in S\}\right| \leqslant 4\varepsilon\sqrt{\log(1/(\gamma\varepsilon))} \cdot \mathbb{P}_{X\sim\mathcal{N}(\mu,I)}\{X \in S\}.$$

*Proof.* Let $\mu = 0$ without loss of generality and defining $r = 2(\sqrt{\log(1/(\gamma\varepsilon))} + \varepsilon)$, we have:

$$
\begin{aligned}
\mathbb{P}\{X \in S\} &= \frac{1}{2\pi} \int_S \exp\left\{-\frac{\|x - \mu'\|^2}{2}\right\} dx \\
&= \frac{1}{2\pi}\left(\int_{S \cap \mathbb{B}(0,r)} \exp\left\{-\frac{\|x - \mu'\|^2}{2}\right\} dx + \int_{S \setminus \mathbb{B}(0,r)} \exp\left\{-\frac{\|x - \mu'\|^2}{2}\right\} dx\right) \\
&\leqslant \frac{1}{2\pi} \int_{S \cap \mathbb{B}(0,r)} \exp\left\{-\frac{\|x - \mu'\|^2}{2}\right\} dx + \mathbb{P}_{G \sim \mathcal{N}(\mu', I)}\{\|G\| \geqslant r\} \\
&\leqslant \frac{1}{2\pi} \int_{S \cap \mathbb{B}(0,r)} \exp\left\{-\frac{\|x - \mu'\|^2}{2}\right\} dx + \mathbb{P}_{G \sim \mathcal{N}(0, I)}\{\|G\| \geqslant r - \varepsilon\} \\
&\leqslant \frac{1}{2\pi} \int_{S \cap \mathbb{B}(0,r)} \exp\left\{-\frac{\|x\|^2 + \|\mu\|^2 - 2\langle x, \mu'\rangle}{2}\right\} dx + \mathbb{P}_{G \sim \mathcal{N}(0, I)}\{\|G\| \geqslant r - \varepsilon\} \\
&\leqslant \frac{1}{2\pi} \int_{S \cap \mathbb{B}(0,r)} \exp\left\{-\frac{\|x\|^2 - 2\langle x, \mu'\rangle}{2}\right\} dx + \gamma\varepsilon \\
&\leqslant \frac{1}{2\pi} \int_{S \cap \mathbb{B}(0,r)} \exp\left\{\varepsilon r\right\} \exp\left\{-\frac{\|x\|^2}{2}\right\} dx + \gamma\varepsilon \\
&\leqslant \exp\left\{\varepsilon r\right\} \mathbb{P}_{X \sim \mathcal{N}(0, I)}\{X \in S\} + \gamma\varepsilon \leqslant (\exp(\varepsilon r) + \varepsilon) \cdot \mathbb{P}_{X \sim \mathcal{N}(0, I)}\{X \in S\}.
\end{aligned}
$$

For the lower bound, we have:

$$
\begin{aligned}
\mathbb{P}\{X \in S\} &= \frac{1}{2\pi} \int_S \exp\left\{-\frac{\|x - \mu'\|^2}{2}\right\} dx \\
&= \frac{1}{2\pi}\left(\int_{S \cap \mathbb{B}(0,r)} \exp\left\{-\frac{\|x - \mu'\|^2}{2}\right\} dx + \int_{S \setminus \mathbb{B}(0,r)} \exp\left\{-\frac{\|x - \mu'\|^2}{2}\right\} dx\right) \\
&\geqslant \frac{1}{2\pi} \int_{S \cap \mathbb{B}(0,r)} \exp\left\{-\frac{\|x - \mu'\|^2}{2}\right\} dx \\
&= \frac{1}{2\pi} \int_{S \cap \mathbb{B}(0,r)} \exp\left\{-\frac{\|x\|^2 + \|\mu\|^2 - 2\langle x, \mu'\rangle}{2}\right\} dx \\
&\geqslant \frac{1}{2\pi} \int_{S \cap \mathbb{B}(0,r)} \exp\left\{-\varepsilon(\varepsilon + r)\right\} \exp\left\{-\frac{\|x\|^2}{2}\right\} dx \\
&= \frac{\exp\left\{-\varepsilon(\varepsilon + r)\right\}}{2\pi}\left(\int_S \exp\left\{-\frac{\|x\|^2}{2}\right\} dx - \int_{S \setminus \mathbb{B}(0,r)} \exp\left\{-\frac{\|x\|^2}{2}\right\} dx\right) \\
&\geqslant \frac{\exp\left\{-\varepsilon(\varepsilon + r)\right\}}{2\pi}\left(\int_S \exp\left\{-\frac{\|x\|^2}{2}\right\} dx - \int_{\mathbb{R}^d \setminus \mathbb{B}(0,r)} \exp\left\{-\frac{\|x\|^2}{2}\right\} dx\right) \\
&= \exp\left\{-\varepsilon(\varepsilon + r)\right\}\left(\mathbb{P}_{X \sim \mathcal{N}(0, I)}\{X \in S\} - \mathbb{P}_{X \sim \mathcal{N}(0, I)}\{\|X\| \geqslant r\}\right) \\
&\geqslant (1 - \varepsilon(\varepsilon + r))(1 - \varepsilon)\mathbb{P}_{X \sim \mathcal{N}(0, I)}\{X \in S\} \\
&\geqslant (1 - 2\varepsilon(\varepsilon + r))\mathbb{P}_{X \sim \mathcal{N}(0, I)}\{X \in S\}
\end{aligned}
$$

which concludes the proof of the result. $\qquad\square$

We also require the upper bound on the Gaussian tail.

**Lemma B.6.** *For all $t \geqslant 0$, we have:*

$$
\mathbb{P}_{X \sim \mathcal{N}(0,1)}\{X \geqslant t\} \leqslant \frac{1}{2} \exp\left\{-\frac{t^2}{2}\right\}.
$$

We will now adapt the three steps outlined in Section 4 for estimation. These proofs are similar to those for identifiability but are substantially more technical. We will adopt a more streamlined

notation for this proof. Recall that Section 4 required two projections, the first to reduce the dimensionality to 2 (Figure 1) and another to reduce the 2-dimensional Gaussian to the isotropic setting. Here, we will instead work with a single transformation. We start by defining:

$$c_1 := \begin{bmatrix} 1 \\ -1 \\ 0 \end{bmatrix}, \ c_2 := \begin{bmatrix} 0 \\ 1 \\ -1 \end{bmatrix}, \ c_3 := \begin{bmatrix} 1 \\ 0 \\ -1 \end{bmatrix}.$$

For the utility vector $x \sim \mathcal{N}(\mu, \Sigma)$, we observe the permutation $u_1 \succ u_2 \succ u_3$ when $\langle c_1, x \rangle > 0$ and $\langle c_2, x \rangle > 0$. Similar conditions may be derived for all other permutations. Note that $c_1, c_2$ and $c_3$ are linearly dependent $c_1 + c_2 = c_3$, and any two form a basis for the subspace orthogonal to $\mathbf{1}$.

Consider the linear transformations $\Sigma^{-1/2} \in \mathbb{R}^{2 \times 3}$ and $\Sigma^{1/2} \in \mathbb{R}^{2 \times 3}$ where

$$\Sigma^{-1/2} \Sigma (\Sigma^{-1/2})^\top = \mathrm{Id}(2), \tag{3}$$

$$\Sigma^{-1/2} (\Sigma^{1/2})^\top = \mathrm{Id}(2) \tag{4}$$

$$\Sigma^{1/2} c_1 = t_1 \begin{bmatrix} 1 \\ 0 \end{bmatrix}, \ \text{and} \tag{5}$$

$$\Sigma^{1/2} c_2 = \begin{bmatrix} \star \\ t_2 \end{bmatrix} \ \text{where } t_2 \geqslant 0. \tag{6}$$

Since we assume that $\Sigma$ has rank 2, (3) must have a solution. Since the left-hand side of (3) is invariant to rotations and reflections, without loss of generality, (5) and (6) can be satisfied. Observe that this implies a similar property for $c_3$ as $c_1 + c_2 = c_3$.

Let $\widetilde{\mu} := \Sigma^{-1/2} \mu$ and consider the transformed vectors:

$$\widetilde{c}_1 = \frac{\Sigma^{1/2} c_1}{\|\Sigma^{1/2} c_1\|}, \ \widetilde{c}_2 = \frac{\Sigma^{1/2} c_2}{\|\Sigma^{1/2} c_2\|}, \ \text{and } \widetilde{c}_3 = \frac{\Sigma^{1/2} c_3}{\|\Sigma^{1/2} c_3\|}.$$

We now show that the quantities $\langle \widetilde{c}_i, \widetilde{\mu} \rangle$ are estimable under the conclusion of Lemma B.4. This extends the identification result from Lemma 4.2.

**Lemma B.7.** *Defining the estimates:*

$$\widehat{\alpha}_i := \Phi^{-1} \left\{ \sum_{j=1}^n \mathbf{1}\{\langle X_j, c_i \rangle \geqslant 0\} \right\} \ \text{and } \alpha_i := \langle \widetilde{c}_i, \widetilde{\mu} \rangle,$$

*we have:*

$$|\widehat{\alpha}_i - \alpha_i| \leqslant \sqrt{2\pi} \cdot \frac{\widetilde{\varepsilon}}{\gamma}$$

$$-\sqrt{2 \log(1/\gamma)} \leqslant \widehat{\alpha}_i, \alpha_i \leqslant \sqrt{2 \log(1/\gamma)}.$$

*Proof.* Recall from Lemma 4.2:

$$\alpha_i = \Phi^{-1}(\mathbb{P}_{X \sim \mathcal{N}(\mu, \Sigma)}\{\langle X, c_i \rangle \geqslant 0\}).$$

Furthermore, we have from Assumption 5.1 and Lemma B.4:

$$\frac{\gamma}{2} \leqslant \frac{1}{n} \sum_{j=1}^n \mathbf{1}\{\langle X_j, c_i \rangle \geqslant 0\} \leqslant 1 - \frac{\gamma}{2}.$$

Therefore, we get from Lemma B.6:

$$-\sqrt{2 \log(1/\gamma)} \leqslant \widehat{\alpha}_i, \alpha_i \leqslant \sqrt{2 \log(1/\gamma)}.$$

Using this fact, we obtain:

$$\widetilde{\varepsilon} \geqslant |\Phi(\alpha_i) - \Phi(\widehat{\alpha}_i)| = \left| \int_{\alpha_i}^{\widehat{\alpha}_i} \phi(x) dx \right| \geqslant \frac{1}{\sqrt{2\pi}} |\alpha_i - \widehat{\alpha}_i| \cdot \min_{x \in [\alpha_i, \widehat{\alpha}_i]} \exp\left\{ -\frac{x^2}{2} \right\} \geqslant \frac{1}{\sqrt{2\pi}} |\alpha_i - \widehat{\alpha}_i| \gamma.$$

By re-arranging the above, we get:

$$|\alpha_i - \widehat{\alpha}_i| \leqslant \sqrt{2\pi} \cdot \frac{\widetilde{\varepsilon}}{\gamma}$$

concluding the proof of the lemma. $\qquad \square$

Next, we adapt the second step of our identification argument (Lemma 4.3) to estimation. As before, this step is substantially more involved to account for the statistical noise.

**Lemma B.8.** *Let $\mu \in \mathbb{R}^2$ and $v_1, v_2$ be two linearly independent unit vectors. Suppose, further that estimates $\widehat{d}_1 \geqslant \widehat{d}_2 \geqslant 0$ are known estimates such that:*

$$|\widehat{d}_1 - \langle v_1, \mu \rangle| \leqslant \varepsilon \text{ and } |\widehat{d}_2 - \langle v_2, \mu \rangle| \leqslant \varepsilon.$$

*Furthermore, assume that the following hold:*

$$\forall u_1 \in \{\pm v_1\}, u_2 \in \{\pm v_2\} : \mathbb{P}_{X \sim \mathcal{N}(\mu, I)}\{\langle X, u_1 \rangle, \langle X, u_2 \rangle \geqslant 0\} \geqslant \gamma$$

*and we have access to estimates, $\{\widehat{\gamma}_{s_1, s_2}\}_{s_1 \in \{\pm v_1\}, s_2 \in \{\pm v_2\}}$ with:*

$$\left|\widehat{\gamma}_{s_1, s_2} - \mathbb{P}_{X \sim \mathcal{N}(\mu, I)}\{\langle X, s_1 \rangle, \langle X, s_2 \rangle \geqslant 0\}\right| \leqslant \varepsilon.$$

*Then, we may recover estimate $\widehat{\beta}$ such that:*

$$|\widehat{\beta} - \langle v_1, v_2 \rangle| \leqslant C \frac{\varepsilon}{\gamma^4} \sqrt{\log(1/(\gamma \varepsilon))}$$

*from $\widehat{d}_1$, $\widehat{d}_2$, and $\{\widehat{\gamma}_{s_1, s_2}\}$.*

*Proof.* As in the proof of Lemma 4.3, we may assume $v_1 = e_2$ and $(v_2)_1 \geqslant 0$ without loss of generality. Since, $v_2$ is a unit vector, it may be parameterized as $v_2 = v(\theta^*)$ for $\theta^* \in [0, \pi]$ with $v(\theta)$ defined as follows:

$$v(\theta) := \begin{bmatrix} \sin(\theta) \\ -\cos(\theta) \end{bmatrix} \text{ for } \theta \in [0, \pi].$$

Now, defining $\widehat{\mu}(\theta)$:

$$\widehat{\mu}(\theta) := \begin{bmatrix} \widetilde{\mu}(\theta) \\ \widehat{d}_1 \end{bmatrix} \text{ where } \widetilde{\mu}(\theta) := \frac{\widehat{d}_1 \cos(\theta) + \widehat{d}_2}{\sin(\theta)}.$$

Our estimate of $\theta^*$ (with $\cos(\pi - \theta^*) = \langle v_1, v_2 \rangle$) is defined below:

$$\widehat{\theta} := \arg\min_{\theta} \max_{s_1 \in \{\pm v_1\}, s_2 \in \{\pm v_2\}} \ell_{s_1, s_2}(\theta) \text{ where } \ell_{s_1, s_2}(\theta) := \left|\mathbb{P}_{X \sim \mathcal{N}(\widehat{\mu}(\theta), I)}\{\langle X, s_1 \rangle, \langle X, s_2 \rangle \geqslant 0\} - \widehat{\gamma}_{s_1, s_2}\right|.$$

Our proof that $\widehat{\theta}$ is close to $\theta^*$ will proceed in two steps:

1. In the first step, we show that $\theta$ is a *solution* with small error

2. Next, we show that all points outside a small interval around $\theta$ have large error.

**Claim B.9.** We have:

$$\forall s_1 \in \{\pm v_1\}, s_2 \in \{\pm v_2\} : \ell_{s_1, s_2}(\theta^*) \leqslant \frac{64\varepsilon}{\gamma} \sqrt{\log(1/(\gamma \varepsilon))}$$

$$\|\mu - \widehat{\mu}(\theta^*)\| \leqslant 4 \frac{\varepsilon}{\gamma}.$$

*Proof.* Observe from the rotational symmetry of a standard Gaussian that for any $u_1, u_2 \in \mathbb{S}^1$:

$$\mathbb{P}_{X \sim \mathcal{N}(0, I)}\{\langle X, u_1 \rangle, \langle X, u_2 \rangle \geqslant 0\} = \frac{1}{2} - \frac{\arccos(\langle u_1, u_2 \rangle)}{2\pi}.$$

Hence, applying Lemma B.2 with the vectors $(v_1, v_2)$ yields:

$$\frac{\pi - \theta^*}{2\pi} \geqslant \gamma \implies \theta^* \leqslant \pi - 2\pi\gamma.$$

Similarly, applying Lemma B.2 with the vectors $(v_1, -v_2)$ yields:

$$\theta^* \geqslant 2\pi\gamma.$$

Hence, we get:

$$2\pi\gamma \leqslant \theta^* \leqslant \pi - 2\pi\gamma \implies \sin(\theta^*) \geqslant \frac{2}{\pi} \cdot 2\pi\gamma = 4\gamma.$$

Now, observing that:

$$\mu = \begin{bmatrix} \frac{d_1 \cos(\theta^*) + d_2}{\sin(\theta^*)} \\ d_1 \end{bmatrix},$$

we get:

$$\|\mu - \widehat{\mu}(\theta^*)\| \leqslant \left| \frac{d_1 \cos(\theta^*) + d_2}{\sin(\theta^*)} - \frac{\widehat{d}_1 \cos(\theta^*) + \widehat{d}_2}{\sin(\theta^*)} \right| + \left| d_1 - \widehat{d}_1 \right|$$

$$\leqslant |d_1 - \widehat{d}_1||\cot(\theta^*)| + |d_2 - \widehat{d}_2||\csc(\theta^*)| + \varepsilon \leqslant 4\frac{\varepsilon}{\gamma}$$

yielding the first claim. The second now follows from Lemma B.5. $\qquad\square$

We now proceed to the second step of the proof. We start by bounding $\widetilde{\mu}(\theta^*)$.

**Claim B.10.** We have:

$$|\widetilde{\mu}(\theta^*)| \leqslant |\widetilde{\mu}(\theta^*)| \leqslant \sqrt{2\log(1/(2\gamma))} + \frac{\varepsilon}{\gamma}$$

*Proof.* Furthermore, observe that:

$$\forall \theta \in \left[0, \frac{\pi}{2}\right] : \{x : \langle x, v_1 \rangle, \langle x, v_2 \rangle \geqslant 0\} \subseteq \{x : x_1 \geqslant 0\}$$

$$\forall \theta \in \left[\frac{\pi}{2}, \pi\right] : \{x : \langle x, -v_1 \rangle, \langle x, v_2 \rangle \geqslant 0\} \subseteq \{x : x_1 \geqslant 0\}.$$

As a consequence, in both cases:

$$\mathbb{P}_{X \sim \mathcal{N}(\mu, I)}\{X_1 \geqslant 0\} \geqslant \gamma.$$

Hence, we get from Lemma B.6:

$$\mu_1 \geqslant -\sqrt{2\log(1/2\gamma)}.$$

Similarly, we get

$$\forall \theta \in \left[0, \frac{\pi}{2}\right] : \{x : \langle x, -v_1 \rangle, \langle x, -v_2 \rangle \geqslant 0\} \subseteq \{x : x_1 \leqslant 0\}$$

$$\forall \theta \in \left[\frac{\pi}{2}, \pi\right] : \{x : \langle x, v_1 \rangle, \langle x, -v_2 \rangle \geqslant 0\} \subseteq \{x : x_1 \leqslant 0\}.$$

We, hence, get as a consequence:

$$\mathbb{P}_{X \sim \mathcal{N}(\mu, I)}\{X_1 \leqslant 0\} \geqslant \gamma.$$

Lemma B.6 similarly yields:

$$\mu_1 \leqslant \sqrt{2\log(1/(2\gamma))}.$$

Note, as before:

$$\mu = \begin{bmatrix} \frac{d_1 \cos(\theta^*) + d_2}{\sin(\theta^*)} \\ d_1 \end{bmatrix}.$$

Finally, we get from Claim B.9:

$$\|\mu - \widehat{\mu}(\theta^*)\| \leqslant \frac{\varepsilon}{\gamma} \implies |\widetilde{\mu}(\theta^*)| \leqslant \sqrt{2\log(1/(2\gamma))} + \frac{\varepsilon}{\gamma}$$

concluding the proof of the claim. $\qquad\square$

Similar to Lemma 4.3, define:

$$\widehat{\gamma}(\theta) \coloneqq \mathbb{P}_{X \sim \mathcal{N}(\widehat{\mu}(\theta), I)}\{\langle X, v_1 \rangle, \langle X, v_2 \rangle \leqslant 0\}.$$

The next claim lower bounds the derivative of $\widehat{\gamma}$ in an interval around $\theta^*$.

**Claim B.11.** We have:

$$\forall \theta \in (0, \pi) : \widehat{\gamma}'(\theta) > 0$$

$$\forall \theta \in [\theta_l, \theta_h] : \widehat{\gamma}'(\theta) \geqslant \frac{\gamma^3}{16}$$

$$\text{with } \theta_l = \theta^* - \nu, \quad \theta_h = \theta^* + \nu, \quad \nu = C \frac{\varepsilon \sqrt{\log(1/(\gamma\varepsilon))}}{\gamma^4}.$$

*Proof.* The proof that $\widehat{\gamma}'(\theta) > 0$ for all $\theta \in (0, \pi)$ is identical to that of the corresponding result in Lemma 4.3. We now proceed to the second claim. Observe as in the proof of Lemma 4.3 that:

$$\widehat{\gamma}'(\theta) \geqslant \int_0^\infty \frac{1}{2\pi} \exp\left\{ -\frac{(r\cos(\theta) + \widetilde{\mu}(\theta))^2 + (r\sin(\theta) + \widehat{d}_1)^2}{2} \right\} r\,dr$$

$$= \exp\left\{ -\frac{\widetilde{\mu}(\theta)^2 + \widehat{d}_1^2}{2} \right\} \int_0^\infty \frac{1}{2\pi} \exp\left\{ -\frac{r^2 + 2r(\cos(\theta)\widetilde{\mu}(\theta) + \sin(\theta)\widehat{d}_1)}{2} \right\} r\,dr$$

$$\geqslant \exp\left\{ -\frac{\widetilde{\mu}(\theta)^2 + \widehat{d}_1^2}{2} \right\} \int_0^1 \frac{1}{2\pi} \exp\left\{ -\frac{r^2 + 2r(\cos(\theta)\widetilde{\mu}(\theta) + \sin(\theta)\widehat{d}_1)}{2} \right\} r\,dr$$

$$\geqslant \exp\left\{ -\frac{\widetilde{\mu}(\theta)^2 + \widehat{d}_1^2 + 2(|\widetilde{\mu}(\theta)| + |\widehat{d}_1|)}{2} \right\} \int_0^1 \frac{1}{2\pi} \exp\left\{ -\frac{r^2}{2} \right\} r\,dr$$

$$\geqslant \exp\left\{ -\frac{\widetilde{\mu}(\theta)^2 + \widehat{d}_1^2 + 2(|\widetilde{\mu}(\theta)| + |\widehat{d}_1|)}{2} \right\} \cdot \frac{1}{16} \geqslant \gamma^3 \cdot \frac{1}{16} = \frac{\gamma^3}{16}$$

where the last inequality follows from the following:

$$|\widetilde{\mu}(\theta)| \leqslant |\widetilde{\mu}(\theta^*)| + (\theta_h - \theta_l) \max_{\theta \in [\theta_l, \theta_h]} |\widetilde{\mu}'(\theta)|$$

$$= |\widetilde{\mu}(\theta^*)| + (\theta_h - \theta_l) \max_{\theta \in [\theta_l, \theta_h]} \left| \frac{\widehat{d}_1 - \widehat{d}_2 \cos(\theta)}{\sin^2(\theta)} \right| \leqslant |\widetilde{\mu}(\theta^*)| + (\theta_h - \theta_l) \cdot \frac{\sqrt{\log(2/\gamma)}}{\gamma^2}$$

concluding the proof of the claim with Claim B.10. $\qquad\square$

From Claim B.11, we get that:

$$\forall \theta \geqslant \theta_h : \widehat{\gamma}(\theta) \geqslant \widehat{\gamma}(\theta^*) + \frac{\nu\gamma^3}{16}$$

$$\forall \theta \leqslant \theta_l : \widehat{\gamma}(\theta) \geqslant \widehat{\gamma}(\theta^*) - \frac{\nu\gamma^3}{16}.$$

Consequently, we obtain from Claim B.9 and picking $C'$ large enough:

$$\forall \theta \geqslant \theta_h : \widehat{\gamma}(\theta) - \widehat{\gamma}_{-v_1, -v_2} \geqslant \frac{\nu\gamma^3}{16} - \frac{64\varepsilon}{\gamma}\sqrt{\log(1/(\gamma\varepsilon))} \geqslant \frac{128\varepsilon}{\gamma}\sqrt{\log(1/(\gamma\varepsilon))}$$

$$\forall \theta \leqslant \theta_l : \widehat{\gamma}_{-v_1, -v_2} - \widehat{\gamma}(\theta) \leqslant \frac{64\varepsilon}{\gamma}\sqrt{\log(1/(\gamma\varepsilon))} - \frac{\nu\gamma^3}{16} \leqslant -\frac{128\varepsilon}{\gamma}\sqrt{\log(1/(\gamma\varepsilon))}.$$

Hence, we get that $\widehat{\theta} \in [\theta_l, \theta_h]$. Defining $\widehat{\beta} = -\cos(\widehat{\theta})$, the conclusion follows from the 1-Lipschitzness of the function $-\cos(\theta)$. $\qquad\square$

*Proof of Theorem B.1.* The proof proceeds similarly to that of Theorem 4.1. Let $\widetilde{\varepsilon}$ be a parameter that will be chosen subsequently. We have from Lemma B.4 that:

$$\forall v_1, v_2 \in \{\pm c_i\}_{i=1}^3 \text{ s.t } v_1 \not\Vert v_2 : \left| \frac{1}{n}\sum_{i=1}^n \mathbf{1}\{\langle X_i, v_1\rangle, \langle X_i, v_2\rangle \geqslant 0\} - \mathbb{P}_{X \sim \mathcal{N}(\mu, \Sigma)}\{\langle X, v_1\rangle, \langle X, v_2\rangle \geqslant 0\} \right| \leqslant \widetilde{\varepsilon}$$

$$\forall v \in \{\pm c_i\}_{i=1}^3 : \left| \frac{1}{n} \sum_{i=1}^n \mathbf{1}\{\langle X_i, v \rangle \geqslant 0\} - \mathbb{P}_{X \sim \mathcal{N}(\mu, \Sigma)}\{\langle X, v \rangle \geqslant 0\} \right| \leqslant \widetilde{\varepsilon}$$

with probability at least $1 - \delta$ when $n \geqslant C(\log(1/\delta)/\widetilde{\varepsilon}^2)$. We condition on this event in the remainder of the proof. Recall the definition of the vectors:

$$\widetilde{c}_i = \frac{\Sigma^{1/2} c_i}{\|\Sigma^{1/2} c_i\|} \text{ and } \widetilde{\mu} = \Sigma^{-1/2}\mu.$$

From Lemma B.7, we obtain estimates, $\widehat{\alpha}_i$ such that:

$$|\langle \widetilde{\mu}, \widetilde{c}_i \rangle - \widehat{\alpha}_i| \leqslant \frac{8\widetilde{\varepsilon}}{\gamma} =: \varepsilon'.$$

An application of Lemma B.8 now yields estimates $\widehat{\beta}_{ij}$ for all $i, j \in [3]$ such that:

$$|\widehat{\beta}_{ij} - \langle \widetilde{c}_i, \widetilde{c}_j \rangle| \leqslant C \frac{\widetilde{\varepsilon}}{\gamma^4} \sqrt{\log(1/(\gamma\varepsilon))} =: \varepsilon^\dagger.$$

We now fix the normalization of $\Sigma^{1/2}$ and assume without loss of generality that:

$$\Sigma^{1/2} c_1 = \begin{bmatrix} 1 \\ 0 \end{bmatrix}, \ \Sigma^{1/2} c_2 = s \begin{bmatrix} \beta_{12} \\ \sqrt{1 - \beta_{12}^2} \end{bmatrix}, \text{ and } \Sigma^{1/2} c_3 = t \begin{bmatrix} \beta_{13} \\ \sqrt{1 - \beta_{13}^2} \end{bmatrix}$$

Noting that $c_1 + c_2 = c_3$, we get similarly to before:

$$\begin{bmatrix} 1 \\ 0 \end{bmatrix} = \begin{bmatrix} \beta_{13} & \beta_{12} \\ \sqrt{1 - \beta_{13}^2} & \sqrt{1 - \beta_{12}^2} \end{bmatrix} \begin{bmatrix} t \\ -s \end{bmatrix}.$$

As observed in Theorem 4.1, the above equations allow for *identifiability* of $\Sigma$. However, for *estimation*, we incur errors due to inaccurate estimates of quantities $\beta_{ij}$ and $s$. We start with the corresponding *empirical* approximation of the above equality:

$$\begin{bmatrix} 1 \\ 0 \end{bmatrix} = \begin{bmatrix} \widehat{\beta}_{13} & \widehat{\beta}_{12} \\ \sqrt{1 - \widehat{\beta}_{13}^2} & \sqrt{1 - \widehat{\beta}_{12}^2} \end{bmatrix} \begin{bmatrix} \widehat{t} \\ -\widehat{s}. \end{bmatrix}$$

In our first claim, we bound the approximation error of the above matrices by their empirical counterparts. Define:

$$\forall i, j \in [3] \text{ s.t } i \neq j : B_{ij} := \begin{bmatrix} \beta_{1i} & \beta_{1j} \\ \sqrt{1 - \beta_{1i}^2} & \sqrt{1 - \beta_{1j}^2} \end{bmatrix} \text{ and } \widehat{B}_{ij} := \begin{bmatrix} \widehat{\beta}_{1i} & \widehat{\beta}_{1j} \\ \sqrt{1 - \widehat{\beta}_{1i}^2} & \sqrt{1 - \widehat{\beta}_{1j}^2} \end{bmatrix}.$$

Our approximation guarantees are stated below:

**Claim B.12.** We have:

$$|\det(B_{ij})| \geqslant \frac{2\gamma}{\pi} \text{ and } |\det(\widehat{B}_{ij})| \geqslant \frac{\gamma}{\pi}$$

$$\|B_{ij}\|, \|\widehat{B}_{ij}\| \leqslant 2$$

$$\|B_{ij} - \widehat{B}_{ij}\| \leqslant \frac{4\varepsilon^\dagger}{\gamma}$$

$$\|B_{ij}^{-1}\| \leqslant \frac{\pi}{\gamma} \text{ and } \|\widehat{B}_{ij}^{-1}\| \leqslant \frac{2\pi}{\gamma}$$

$$\|B_{ij}^{-1} - \widehat{B}_{ij}^{-1}\| \leqslant \frac{8\varepsilon^\dagger \pi^2}{\gamma^3}.$$

*Proof.* We have as a consequence of Lemma B.2 and Assumption 5.1 that $\langle \widetilde{c}_i, \widetilde{c}_j \rangle = \cos(\theta_{ij})$ for some $\gamma \leqslant \theta_{ij} \leqslant \pi - \gamma$. Similarly, we get from the approximation guarantees of Lemma B.8 that the empirical estimates satisfy $\langle \widehat{c}_i, \widehat{c}_j \rangle = \cos(\widehat{\theta}_{ij})$ for $\gamma/2 \leqslant \widehat{\theta}_{ij} \leqslant \pi - \gamma/2$. Consequently, we have:

$$|\det(B_{ij})| \geqslant |\sin(\theta_{ij})| \geqslant \frac{2\gamma}{\pi} \text{ and } |\det(\widehat{B}_{ij})| \geqslant \sin(\widehat{\theta}_{ij}) \geqslant \frac{\gamma}{\pi}$$

establishing the first result of the claim. For the second, we have:

$$\|B_{ij}\| \leqslant \|B_{ij}\|_F = \sqrt{2}$$

and similarly for $\widehat{B}_{ij}$. For the third, observe again from the guarantees of $\theta_{ij}$ and $\widehat{\theta}$:

$$\left| \sqrt{1 - \beta_{ij}^2} - \sqrt{1 - \widehat{\beta}_{ij}^2} \right| = \left| \frac{(\beta_{ij} + \widehat{\beta}_{ij})(\beta_{ij} - \widehat{\beta}_{ij})}{\sqrt{1 - \beta_{ij}^2} + \sqrt{1 - \widehat{\beta}_{ij}^2}} \right| \leqslant \frac{2\varepsilon^\dagger}{\gamma}$$

and consequently,

$$\|B_{ij} - \widehat{B}_{ij}\| \leqslant \|B_{ij} - \widehat{B}_{ij}\|_F \leqslant 2 \cdot \frac{2\varepsilon^\dagger}{\gamma} = \frac{4\varepsilon^\dagger}{\gamma}.$$

For the fourth claim, we have:

$$\|B_{ij}^{-1}\| = \frac{1}{\sigma_{\min}(B_{ij})} = \frac{\|B_{ij}\|}{|\det(B_{ij})|} \leqslant \frac{\pi}{\gamma}$$

and similarly, for $\widehat{B}_{ij}$. For the final claim, we have:

$$\frac{\gamma}{\pi} \|B_{ij}^{-1} - \widehat{B}_{ij}^{-1}\| \leqslant \sigma_{\min} \|B_{ij}^{-1} - \widehat{B}_{ij}^{-1}\| \leqslant \|B_{ij}(B_{ij}^{-1} - \widehat{B}_{ij}^{-1})\| = \|I - B_{ij}\widehat{B}_{ij}^{-1}\|$$

$$= \|I - \widehat{B}_{ij}\widehat{B}_{ij}^{-1} - (B_{ij} - \widehat{B}_{ij})\widehat{B}_{ij}^{-1}\| = \|(B_{ij} - \widehat{B}_{ij})\widehat{B}_{ij}^{-1}\|$$

$$\leqslant \|B_{ij} - \widehat{B}_{ij}\| \|\widehat{B}_{ij}^{-1}\| \leqslant \frac{2\pi}{\gamma} \|B_{ij} - \widehat{B}_{ij}\| \leqslant \frac{8\pi\varepsilon^\dagger}{\gamma^2}.$$

$\square$

Our next claim bounds the magnitude of $s$ and the relative error of its approximation by $\widehat{s}$.

**Claim B.13.** We have:

$$\frac{\pi}{\gamma} \geqslant \max\{s, t\} \geqslant \frac{1}{2} \text{ and } |s - \widehat{s}|, |t - \widehat{t}| \leqslant \frac{8\varepsilon^\dagger \pi^2}{\gamma^3}.$$

*Proof.* For the first claim, observe the following

$$\beta_{13}t - \beta_{12}s = 1 \implies s + t \geqslant 1,$$

implying the lower bound. For the upper bound:

$$\max\{s, t\} \leqslant \left\| B_{23}^{-1} \begin{bmatrix} 1 \\ 0 \end{bmatrix} \right\| \leqslant \|B_{23}^{-1}\|$$

and the conclusion follows from Claim B.12. For the second, we have:

$$|s - \widehat{s}|, |t - \widehat{t}| \leqslant \left\| (B_{23}^{-1} - \widehat{B}_{23}^{-1}) \begin{bmatrix} 1 \\ 0 \end{bmatrix} \right\| \leqslant \|B_{23}^{-1} - \widehat{B}_{23}^{-1}\|$$

and the conclusion follows from Claim B.12. $\square$

The remainder of the proof proceeds in two cases: 1) $\widehat{s} \geqslant \widehat{t}$ or 2) $\widehat{t} > \widehat{s}$. We start with the first:

**Case 1:** $\widehat{s} \geqslant \widehat{t}$. Since $c_1$ and $c_2$ are a basis for the subspace orthogonal to $\mathbf{1}$, we may write:

$$\Sigma^{1/2} = AC \text{ where } C = \begin{bmatrix} c_1^\top \\ c_2^\top \end{bmatrix} \text{ and } A \in \mathbb{R}^{2 \times 2}.$$

Therefore, we get:

$$\Sigma^{1/2}C^\top = A(CC^\top) = \begin{bmatrix} 1 & s\beta_{12} \\ 0 & s\sqrt{1 - \beta_{12}^2} \end{bmatrix} \implies A = \begin{bmatrix} 1 & \beta_{12} \\ 0 & \sqrt{1 - \beta_{12}^2} \end{bmatrix} \begin{bmatrix} 1 & 0 \\ 0 & s \end{bmatrix} (CC^\top)^{-1}.$$

We may now write its empirical counterpart:

$$\widehat{A} = \begin{bmatrix} 1 & \widehat{\beta}_{12} \\ 0 & \sqrt{1 - \widehat{\beta}_{12}^2} \end{bmatrix} \begin{bmatrix} 1 & 0 \\ 0 & \widehat{s} \end{bmatrix} (CC^\top)^{-1}.$$

Noting that $\|(CC^\top)^{-1}\| \leqslant 1$, we get from Claims B.12 and B.13:

$$\|A - \widehat{A}\| \leqslant \left\| \widehat{B}_{12} \begin{bmatrix} 1 & 0 \\ 0 & \widehat{s} \end{bmatrix} - B_{12} \begin{bmatrix} 1 & 0 \\ 0 & s \end{bmatrix} \right\| = \left\| (\widehat{B}_{12} - B_{12}) \begin{bmatrix} 1 & 0 \\ 0 & \widehat{s} \end{bmatrix} - B_{12} \begin{bmatrix} 0 & 0 \\ 0 & s - \widehat{s} \end{bmatrix} \right\|$$

$$\leqslant \left\| (\widehat{B}_{12} - B_{12}) \begin{bmatrix} 1 & 0 \\ 0 & \widehat{s} \end{bmatrix} \right\| + \left\| \widehat{B}_{12} \begin{bmatrix} 0 & 0 \\ 0 & s - \widehat{s} \end{bmatrix} \right\|$$

$$\leqslant \|\widehat{B}_{12} - B_{12}\| \max\{1, \widehat{s}\} + \|\widehat{B}_{12}\| |s - \widehat{s}| \leqslant \frac{4\varepsilon^\dagger}{\gamma} \cdot \frac{2\pi}{\gamma} + 2 \cdot \frac{8\varepsilon^\dagger \pi^2}{\gamma^3} \leqslant 24 \frac{\varepsilon^\dagger \pi^2}{\gamma^3}.$$

Furthermore, we have for any $v \in \mathbb{R}^2$ with $\|v\| = 1$:

$$\|Av\| = \left\| B_{12} \begin{bmatrix} 1 & 0 \\ 0 & s \end{bmatrix} (CC^\top)^{-1} v \right\| \geqslant \sigma_{\min}(B_{12}) \cdot \sigma_{\min}\left( \begin{bmatrix} 1 & 0 \\ 0 & s \end{bmatrix} \right) \cdot \sigma_{\min}((CC^\top)^{-1})$$

$$\geqslant \frac{\gamma}{\pi} \cdot \min\{1, s\} \cdot \frac{1}{3} \geqslant \frac{\gamma}{12\pi}$$

where the inquality follows from an explicit calculation for $(CC^\top)^{-1}$, Claim B.13 for the middle term and Claim B.12 for the first term. Similarly, note that:

$$\|Av\| = \left\| B_{12} \begin{bmatrix} 1 & 0 \\ 0 & s \end{bmatrix} (CC^\top)^{-1} v \right\| \leqslant \sigma_{\max}(B_{12}) \cdot \sigma_{\max}\left( \begin{bmatrix} 1 & 0 \\ 0 & s \end{bmatrix} \right) \cdot \sigma_{\max}((CC^\top)^{-1})$$

$$\leqslant 2 \cdot \max\{1, s\} \cdot 1 \leqslant \frac{2\pi}{\gamma}$$

from Claims B.12 and B.13. Consequently, we obtain for all $\|v\| = 1$:

$$\left| v^\top A^\top A v - v^\top \widehat{A}^\top \widehat{A} v \right| = \left| \|Av\|^2 - \|\widehat{A}v\|^2 \right| = (\|Av\| + \|\widehat{A}v\|) \big| \|Av\| - \|\widehat{A}v\| \big|$$

$$\leqslant (\|A\| + \|\widehat{A}\|) \|(A - \widehat{A})v\| \leqslant \left( \frac{4\pi}{\gamma} + \|A - \widehat{A}\| \right) \|A - \widehat{A}\| \leqslant 120 \frac{\varepsilon^\dagger \pi^3}{\gamma^4}.$$

From the above three displays, we get:

$$(1 - \varepsilon^\ddagger) A^\top A \preccurlyeq \widehat{A}^\top \widehat{A} \preccurlyeq (1 + \varepsilon^\ddagger) A^\top A \text{ where } \varepsilon^\ddagger := \frac{2^{15} \pi^5 \varepsilon^\dagger}{\gamma^6}.$$

Finally, defining our estimate $\widehat{\Sigma}^{1/2} = \widehat{A}C$ and since $\Sigma = C^\top A^\top AC$, we get:

$$(1 - \varepsilon^\ddagger) \Sigma \preccurlyeq \widehat{\Sigma} \preccurlyeq (1 + \varepsilon^\ddagger) \Sigma.$$

**Case 2:** $\widehat{t} \geqslant \widehat{s}$. Here, we use $c_1$ and $c_3$ as a basis for the subspace orthogonal to $\mathbf{1}$ and write:

$$\Sigma^{1/2} = AC \text{ where } C = \begin{bmatrix} c_1^\top \\ c_3^\top \end{bmatrix} \text{ and } A \in \mathbb{R}^{2 \times 2}.$$

Therefore, we get:

$$\Sigma^{1/2} C^\top = A(CC^\top) = \begin{bmatrix} 1 & t\beta_{13} \\ 0 & t\sqrt{1 - \beta_{13}^2} \end{bmatrix} \implies A = \begin{bmatrix} 1 & \beta_{13} \\ 0 & \sqrt{1 - \beta_{13}^2} \end{bmatrix} \begin{bmatrix} 1 & 0 \\ 0 & t \end{bmatrix} (CC^\top)^{-1}.$$

The remainder of the proof is identical to the previous setting with $t$ taking the role of $s$ and $B_{13}$ (resp $\widehat{B}_{13}$) that of $B_{12}$ (resp $\widehat{B}_{13}$). This concludes the estimation guarantees for $\Sigma$.

**Estimating $\mu$:** To estimate $\mu$, we adopt the $B_{1i}$ and $C$ from the previous two cases and note:

$$\widetilde{\mu} = \Sigma^{-1/2} \mu, \ B_{1i}^\top \widetilde{\mu} = Q \implies \mu = (\Sigma^{1/2})^\top \widetilde{\mu} = C^\top A \widetilde{\mu} = C^\top A^\top (B_{1i}^{-1})^\top Q \text{ with } Q := \begin{bmatrix} \alpha_1 \\ \alpha_i \end{bmatrix}.$$

Empirically, we use the estimates:

$$\widehat{\mu} = C^\top \widehat{A}^\top (\widehat{B}_{1i}^{-1})^\top \widehat{Q} \text{ with } \widehat{Q} := \begin{bmatrix} \widehat{\alpha}_1 \\ \widehat{\alpha}_i \end{bmatrix}.$$

Therefore, we get that:

$$\|\widehat{\mu} - \mu\| = \left\| C^\top (A^\top (B_{1i}^{-1})^\top Q - \widehat{A}^\top (\widehat{B}_{1i}^{-1})^\top \widehat{Q}) \right\| \leqslant 2 \left\| A^\top (B_{1i}^{-1})^\top Q - \widehat{A}^\top (\widehat{B}_{1i}^{-1})^\top \widehat{Q} \right\|$$

$$\leqslant 2(\|A^\top (B_{1i}^{-1})^\top (Q - \widehat{Q})\| + \|A^\top (B_{1i}^{-1})^\top \widehat{Q} - \widehat{A}^\top (\widehat{B}_{1i}^{-1})^\top \widehat{Q}\|)$$

$$\leqslant 2(\|A^\top (B_{1i}^{-1})^\top \|\|(Q - \widehat{Q})\| + \|A^\top (B_{1i}^{-1})^\top - \widehat{A}^\top (\widehat{B}_{1i}^{-1})^\top \|\|\widehat{Q}\|)$$

$$\leqslant 4 \left( \left\| A^\top (B_{1i}^{-1})^\top \right\| \varepsilon' + \left\| A^\top (B_{1i}^{-1})^\top - \widehat{A}^\top (\widehat{B}_{1i}^{-1})^\top \right\| \sqrt{\log(1/\gamma)} \right). \tag{7}$$

For the first term, we have by Claim B.12:

$$\|A^\top (B_{1i}^{-1})^\top \| \leqslant \|A^\top \|\|(B_{1i}^{-1})^\top \| = \|A\|\|B_{1i}^{-1}\| \leqslant \frac{2\pi}{\gamma} \cdot \frac{\pi}{\gamma} = \frac{2\pi^2}{\gamma^2}.$$

For the second, we have:

$$\|A^\top (B_{1i}^{-1})^\top - \widehat{A}^\top (\widehat{B}_{1i}^{-1})^\top \| = \|B_{1i}^{-1} A - \widehat{B}_{1i}^{-1} \widehat{A}\| \leqslant \|B_{1i}^{-1}(A - \widehat{A})\| + \|(B_{1i}^{-1} - \widehat{B}_{1i}^{-1})\widehat{A}\|$$

$$\leqslant \|B_{1i}^{-1}\|\|A - \widehat{A}\| + \|B_{1i}^{-1} - \widehat{B}_{1i}^{-1}\|\|\widehat{A}\|$$

$$\leqslant \frac{\pi}{\gamma} \cdot \frac{24\varepsilon^\dagger \pi^2}{\gamma^3} + \frac{8\varepsilon^\dagger \pi^2}{\gamma^3} \cdot \frac{2\pi}{\gamma} = \frac{40\pi^3 \varepsilon^\dagger}{\gamma^4}.$$

Plugging this back into (7) yields:

$$\|\widehat{\mu} - \mu\| \leqslant \frac{200\pi^3 \sqrt{\log(1/\gamma)}}{\gamma^4} \varepsilon^\dagger.$$

Now, setting:

$$\widetilde{\varepsilon} := c \frac{\varepsilon \gamma^{10}}{\log(1/(\gamma \varepsilon))}$$

for some small constant $c$, the theorem follows by the definitions of $\varepsilon^\dagger$ and $\varepsilon^\ddagger$. □

## C  ESTIMATION - MULTIPLE ALTERNATIVES

In Appendix B, we extended Theorem 4.1 to an algorithm with estimation guarantees (Theorem B.1) that is robust to statistical noise. Hence, we now have a procedure that allows recovering the parameters $\mu, \Sigma$ on $3 \times 3$ sub-matrices (up to the symmetries described in Assumption 3.1). We now prove the main result of Section 5, Theorem 5.2, stated below for convenience, which uses the estimation algorithm for 3 items to construct one for the general $n$-item setting.

**Theorem 5.2.** *Let* $(\mu^*, \Sigma^*)$ *satisfy Assumptions 3.1 and 5.1 with* $\text{Tr}(\Sigma^*) = n$. *Then, for any* $\varepsilon, \delta \in (0, 1/2]$, *given there is a polynomial time algorithm which when allowed $N$ observations of rank-3 permutations from* $\text{Choice}(\mu^*, \Sigma^*)$ *of its choice, returns estimates,* $(\overline{\mu}, \overline{\Sigma})$, *satisfying:*

$$\|\overline{\mu} - \mu^*\|_\infty \leqslant \varepsilon \text{ and } \|\overline{\Sigma} - \Sigma^*\|_\infty \leqslant \varepsilon$$

*with probability at least* $1 - \delta$ *as long as* $N \geqslant C n^2 \varepsilon^{-2} \gamma^{-24} \log(n/\delta) \log^6(n/(\gamma \varepsilon))$.

In Theorem 4.4, we showed that $\mu, \Sigma$ can be recovered from *exact* specifications of *all* $3 \times 3$ sub-matrices. However, this approach has 2 substantial drawbacks:

1. It requires comparison data for *all* of the $\Omega(n^3)$ sub-matrices

2. The sub-matrices need to be specified *exactly* which is infeasible with statistical noise

To reduce the number of sub-matrices required, we formally relate our estimation guarantees to the structure of sub-graphs on the following graph:

$$G = (V, E)$$
$$V = \{\{i, j\} \in [n] \times [n] : i \neq j\}$$
$$E = \{(\{i, j\}, \{j, k\}) : i \neq j \neq k \neq i\}.$$

Observe that each potential sub-matrix corresponds to an edge of the graph and the vertices correspond to entries of $\Sigma$ that we aim to recover. Furthermore, note that a particular choice of sub-matrices corresponds to a sub-graph. The properties of this sub-graph relate to both of the aforementioned challenges:

1. The sub-graph being connected suffices for recovering $\Sigma$ from $3 \times 3$ sub-matrices

2. The error of this aggregation process scales *linearly* with the *diameter* of the sub-graph

The next result (efficiently) constructs a sub-graph with drastically fewer edges ($O(n^2)$ vs $O(n^3)$) while maintaining connectivity of the sub-graph with small (logarithmic) diameter.

**Lemma C.1.** *There exists a subgraph $G' = (V, E')$ with $E' \subset E$ satisfying:*

$$|E'| \leqslant n^2$$
$$\forall v_1, v_2 \in V : \text{dist}_{G'}(v_1, v_2) \leqslant 4(\log(n) + 1).$$

*Proof.* We will construct our edge set as follows:

1. For any $i \in [n]$, let $V_i = [n] \setminus \{i\}$

2. Consider a complete binary tree $T_i = (V_i, E_i)$ on $V_i$

3. For any $(j, k) \in E_i$, add edge $(\{i, j\}, \{i, k\})$ to $G'$.

First, observe that $E' \subset E$ by construction. Next, we have:

$$|E'| \leqslant n \cdot |E_i| = n(n - 2) < n^2.$$

For the final claim, we have from the construction of $E'$ and $T_i$ being a complete binary tree:

$$\forall i \neq j \neq k \neq i : \text{dist}_{G'}(\{i, j\}, \{i, k\}) \leqslant 2(\log(n) + 1).$$

Hence, we get:

$$\text{dist}_{G'}(\{i, j\}, \{k, l\}) \leqslant \text{dist}_{G'}(\{i, j\}, \{j, k\}) + \text{dist}_{G'}(\{j, k\}, \{k, l\}) \leqslant 4(\log(n) + 1)$$

concluding the proof of the lemma. $\square$

Before we proceed, we introduce some notation to ease exposition. We will now run the algorithm of Theorem B.1 on the tuples $(i, j, k)$ such that $(\{i, j\}, \{j, k\}) \in E'$. Let $(\widehat{\mu}_{ijk}, \widehat{\Sigma}_{ijk})$ be the estimates obtained through this process. We will abuse notation and write $(i, j, k) \in E'$ for the set of tuples. It will be convenient to index matrices with alternatives rather than numerical indices. We will typically use $c_{ij} = e_i - e_j$ with $e_i$ and $e_j$ corresponding to the vectors with 1 for item $i$ and $j$ respectively and 0 elsewhere. In the context estimates $\widehat{\Sigma}_{i,j,k}$, $e_i$ and $e_j$ (and consequently, $c_{i,j}$) will denote vectors in $\mathbb{R}^3$ for the positions corresponding to $i$ and $j$ while in the context of $\Sigma$, they will denote vectors in $\mathbb{R}^n$.

We now recover $\Sigma^*$ by solving the following convex program:

$$\underset{t, \Sigma}{\arg\min} \, t$$
$$\forall i, j, k \in E' : (1 - t)\frac{c_{ij}^\top \widehat{\Sigma}_{ijk} c_{ij}}{c_{ik}^\top \widehat{\Sigma}_{ijk} c_{ik}} \leqslant \frac{c_{ij}^\top \Sigma c_{ij}}{c_{ik}^\top \Sigma c_{ik}} \leqslant (1 + t)\frac{c_{ij}^\top \widehat{\Sigma}_{ijk} c_{ij}}{c_{ik}^\top \widehat{\Sigma}_{ijk} c_{ik}}$$
$$\text{Tr}(\Sigma) = n$$

$$\Sigma \mathbf{1} = \mathbf{0}$$

We assume that the estimates, $\widehat{\Sigma}_{ijk}$ are obtained by the algorithm of Theorem B.1 with failure probability $\delta' = \delta/(2n^2)$ and accuracy $\widehat{\varepsilon}$ for a parameter $\widetilde{\varepsilon}$ to be decided subsequently. Letting $\overline{\Sigma}$ denote the solution to the above (linear) program and noting that $\Sigma^*$ is a solution with $t = \widetilde{\varepsilon}$, we get that $\overline{\Sigma}$ has cost at most $\widetilde{\varepsilon}$. We require a brief technical result bounding the largest entry of $\Sigma$.

**Lemma C.2.** *We have:*

$$\Sigma_{ii} \leqslant \frac{100 \log(8/\gamma)}{\gamma^2}.$$

*Proof.* By Markov's inequality, there exist two indices $j, k$ such that $\Sigma_{jj}, \Sigma_{kk} \leqslant 2$. We start by bounding the distance between $\mu_j$ and $\mu_k$. Let $X \sim \mathcal{N}(\mu_{ijk}, \Sigma_{ijk})$ and we have:

$$\mathbb{P}\Big\{|X_j - \mu_j| \geqslant 2\sqrt{\log(4/\gamma)}\Big\} \leqslant \frac{\gamma}{4}$$
$$\mathbb{P}\Big\{|X_k - \mu_k| \geqslant 2\sqrt{\log(4/\gamma)}\Big\} \leqslant \frac{\gamma}{4}.$$

From the above two inequalities and Assumption 5.1:

$$\min\{\mathbb{P}\{X_j \geqslant X_k\}, \mathbb{P}\{X_j \leqslant X_k\}\} \geqslant \gamma,$$

we get the following set of inequalities by the probabilistic method:

$$\mu_j + 2\sqrt{\log(4/\gamma)} \geqslant \mu_k - 2\sqrt{\log(4/\gamma)} \implies \mu_k - \mu_j \leqslant 4\sqrt{\log(4/\gamma)}$$
$$\mu_j - 2\sqrt{\log(4/\gamma)} \leqslant \mu_k + 2\sqrt{\log(4/\gamma)} \implies \mu_k - \mu_j \geqslant -4\sqrt{\log(4/\gamma)}.$$

The above two inequalities yield:

$$|\mu_j - \mu_k| \leqslant 4\sqrt{\log(4/\gamma)}. \tag{8}$$

Next, we have again:

$$\mathbb{P}\Big\{|X_j - \mu_j| \geqslant 2\sqrt{\log(8/\gamma)}\Big\} \leqslant \frac{\gamma}{8}$$
$$\mathbb{P}\Big\{|X_k - \mu_k| \geqslant 2\sqrt{\log(8/\gamma)}\Big\} \leqslant \frac{\gamma}{8}.$$

By a union bound and using Eq. (8), we get:

$$\mathbb{P}\Big\{X_j, X_k \in [\mu_j - 8\sqrt{\log(8/\gamma)}, \mu_j + 8\sqrt{\log(8/\gamma)}]\Big\} \geqslant 1 - \frac{\gamma}{4}.$$

Denoting the event in the above probability by $E$, we now have from Assumption 5.1:

$$\gamma \leqslant \mathbb{P}\{X_i \in [X_j, X_k]\} = \mathbb{P}\{X_i \in [X_j, X_k] \cap E\} + \mathbb{P}\{X_i \in [X_j, X_k] \cap E^c\}$$
$$\leqslant \mathbb{P}\Big\{X_i \in \Big[\mu_j - 8\sqrt{\log(8/\gamma)}, \mu_j + 8\sqrt{\log(8/\gamma)}\Big]\Big\} + \mathbb{P}\{E^c\}$$
$$\leqslant \frac{1}{\sqrt{2\pi\Sigma_{ii}}} \cdot 16\sqrt{\log(8/\gamma)} + \frac{\gamma}{4}.$$

By rearranging the above inequality,

$$\sqrt{\Sigma_{ii}} \leqslant \frac{64}{3\sqrt{2\pi}\gamma}\sqrt{\log(8/\gamma)} \implies \Sigma_{ii} \leqslant \frac{100 \log(8/\gamma)}{\gamma^2}.$$

$\square$

We now show that $\overline{\Sigma}$ is close to $\Sigma^*$.

**Lemma C.3.** *We have:*

$$\forall i, j \in [n] : |\Sigma_{ij} - \Sigma^*_{ij}| \leqslant C\gamma^{-2}(\log(n) + 1)\log(8/\gamma)\widetilde{\varepsilon}.$$

*Proof.* First, observe from the fact that $\overline{\Sigma}\mathbf{1} = \mathbf{0}$:

$$c_{ij}^\top \overline{\Sigma} c_{ij} = \Sigma_{ii} + \Sigma_{jj} - 2\Sigma_{ij} \implies \sum_{j=1}^n c_{ij}^\top \overline{\Sigma} c_{ij} = n\overline{\Sigma}_{ii} + \mathrm{Tr}(\overline{\Sigma}) = n\overline{\Sigma}_{ii} + n. \tag{9}$$

Furthermore, we have from the guarantees of Theorem B.1 that:

$$\forall i, j, k \in E' : (1 - 4\widetilde{\varepsilon})\frac{c_{ij}^\top \Sigma^* c_{ij}}{c_{ik}^\top \Sigma^* c_{ik}} \leqslant \frac{c_{ij}^\top \widehat{\Sigma}_{ijk} c_{ij}}{c_{ik}^\top \widehat{\Sigma}_{ijk} c_{ik}} \leqslant (1 + 4\widetilde{\varepsilon})\frac{c_{ij}^\top \Sigma^* c_{ij}}{c_{ik}^\top \Sigma^* c_{ik}}.$$

Putting these together with the constraints on $\overline{\Sigma}$, we get:

$$\forall i, j, k \in E' : (1 - 6\widetilde{\varepsilon})\frac{c_{ij}^\top \Sigma^* c_{ij}}{c_{ik}^\top \Sigma^* c_{ik}} \leqslant \frac{c_{ij}^\top \overline{\Sigma} c_{ij}}{c_{ik}^\top \overline{\Sigma} c_{ik}} \leqslant (1 + 6\widetilde{\varepsilon})\frac{c_{ij}^\top \Sigma^* c_{ij}}{c_{ik}^\top \Sigma^* c_{ik}}.$$

From Lemma C.1, that for any $\{i,j\}$ and $\{k,l\}$, there exists a path $v_0 := \{i,j\}, v_1, \ldots, v_T = \{k,l\}$ with $T \leqslant 4(\log(n) + 1)$ with $(v_{i-1}, v_i) \in E'$ for all $i \in [T]$. Now observing that:

$$\frac{c_{v_0}^\top \overline{\Sigma} c_{v_0}}{c_{v_T}^\top \overline{\Sigma} c_{v_T}} = \prod_{i=1}^T \frac{c_{v_{i-1}}^\top \overline{\Sigma} c_{v_{i-1}}}{c_{v_i}^\top \overline{\Sigma} c_{v_i}}$$

$$\frac{c_{v_0}^\top \Sigma^* c_{v_0}}{c_{v_T}^\top \Sigma^* c_{v_T}} = \prod_{i=1}^T \frac{c_{v_{i-1}}^\top \Sigma^* c_{v_{i-1}}}{c_{v_i}^\top \Sigma^* c_{v_i}}$$

We get for any $i, j, k, l$ with $i \neq j$ and $k \neq l$:

$$(1 - \varepsilon^\dagger)\frac{c_{ij}^\top \Sigma^* c_{ij}}{c_{kl}^\top \Sigma^* c_{kl}} \leqslant \frac{c_{ij}^\top \overline{\Sigma} c_{ij}}{c_{kl}^\top \overline{\Sigma} c_{kl}} \leqslant (1 + \varepsilon^\dagger)\frac{c_{ij}^\top \Sigma^* c_{ij}}{c_{kl}^\top \Sigma^* c_{kl}} \text{ where } \varepsilon^\dagger = 50\widetilde{\varepsilon}(\log(n) + 1).$$

Now, consider the ratios:

$$\forall i \neq j : r_{ij} := \frac{c_{ij}^\top \overline{\Sigma} c_{ij}}{c_{ij}^\top \Sigma^* c_{ij}}, \ r_{\min} := \min_{ij} r_{ij} \text{ and } r_{\max} := \max_{ij} r_{ij}.$$

As a consequence, we have for any $i, j, k, l$:

$$r_{\min} \leqslant r_{\max} \leqslant (1 + \varepsilon^\dagger)r_{\min}.$$

Hence, we get:

$$\sum_{ij} c_{ij}^\top \overline{\Sigma} c_{ij} = n \sum_i (\overline{\Sigma}_{ii} + 1) = 2n^2 = \sum_{ij} c_{ij}^\top \Sigma^* c_{ij}$$

$$r_{\min}(2n^2) = r_{\min} \sum_{ij} c_{ij}^\top \Sigma^* c_{ij} \leqslant \sum_{ij} c_{ij}^\top \overline{\Sigma} c_{ij} = 2n^2 \leqslant r_{\max} \sum_{ij} c_{ij}^\top \Sigma^* c_{ij} \leqslant (1 + \varepsilon^\dagger)r_{\min}(2n^2).$$

Therefore, we get:

$$r_{\min} \leqslant 1 \leqslant r_{\max} \leqslant (1 + \varepsilon^\dagger)r_{\min} \implies (1 - \varepsilon^\dagger) \leqslant r_{\min} \leqslant r_{\max} \leqslant (1 + \varepsilon^\dagger).$$

Plugging this into to Eq. (9), we get:

$$(1 - \varepsilon^\dagger)n(\Sigma_{ii}^* + 1) = (1 - \varepsilon^\dagger) \sum_{j=1}^n c_{ij} \Sigma^* c_{ij} \leqslant n(\overline{\Sigma}_{ii} + 1) \leqslant (1 + \varepsilon^\dagger) \sum_{j=1}^n c_{ij} \Sigma^* c_{ij} = (1 + \varepsilon^\dagger)n(\Sigma_{ii}^* + 1).$$

By re-arranging the above, we get:

$$\forall i : |\Sigma_{ii}^* - \overline{\Sigma}_{ii}| \leqslant \varepsilon^\dagger(\Sigma_{ii}^* + 1).$$

For the off-diagonal terms, observe:

$$\overline{\Sigma}_{ij} = \frac{c_{ij}^\top \overline{\Sigma} c_{ij} - \overline{\Sigma}_{ii} - \overline{\Sigma}_{jj}}{2} \leqslant \frac{(1 + \varepsilon^\dagger)c_{ij}\Sigma^* c_{ij} - (1 - \varepsilon^\dagger)(\Sigma_{ii}^* + \Sigma_{jj}^*) + 2\varepsilon^\dagger}{2} \leqslant \Sigma_{ij}^* + \varepsilon^\dagger(2(\Sigma_{ii}^* + \Sigma_{jj}^*) + 1).$$

Similarly, we get:

$$\overline{\Sigma}_{ij} = \frac{c_{ij}^\top \overline{\Sigma} c_{ij} - \overline{\Sigma}_{ii} - \overline{\Sigma}_{jj}}{2} \geqslant \frac{(1 - \varepsilon^\dagger)c_{ij}\Sigma^* c_{ij} - (1 + \varepsilon^\dagger)(\Sigma_{ii}^* + \Sigma_{jj}^*) - 2\varepsilon^\dagger}{2} \leqslant \Sigma_{ij}^* - \varepsilon^\dagger(2(\Sigma_{ii}^* + \Sigma_{jj}^*) + 1).$$

Putting the above two displays together, we get:

$$|\Sigma_{ij} - \Sigma_{ij}^*| \leqslant 2\varepsilon^\dagger(\Sigma_{ii}^* + \Sigma_{jj}^* + 1)$$

which with Lemma C.2, concludes the proof of the lemma. $\qquad\square$

A similar approach now allows recovering $\mu^*$ with a related convex program.

**Lemma C.4.** *We have:*

$$\|\bar{\mu} - \mu^*\|_\infty \leqslant \frac{4000 \log(8/\gamma)}{\gamma} \varepsilon^\dagger.$$

*Proof.* We use $\mu_{ijk}$ and $\Sigma_{ijk}$ (and correspondingly for their starred and barred variants) to denote the restriction of $\mu$ and $\Sigma$ (and their variants) to the alternatives $i$, $j$, and $k$, *and projected* orthogonal to the vector $\mathbf{1}$. We estimate some rescaling parameters to obtain $\overline{\Sigma}$ from the 3-way estimates. Define:

$$\forall i, j, k \in E' : s_{ijk}^2 := \frac{\operatorname{Tr}(\overline{\Sigma}_{ijk})}{\operatorname{Tr}(\widehat{\Sigma}_{ijk})}.$$

The linear program we solve to find $\mu$ is defined below:

$$\underset{t,\mu}{\arg\min}\, t$$
$$\forall i, j, k \in E' : \|s_{ijk}\widehat{\mu}_{ijk} - \mu_{ijk}\|_\infty \leqslant t$$
$$\langle \mu, \mathbf{1} \rangle = 0.$$

Before we proceed, we prove an analogue of Lemma C.2 which bounds the mean of $\mu$.

**Lemma C.5.** *We have:*

$$\forall i \in [n] : |\mu_i| \leqslant \frac{40 \log(8/\gamma)}{\gamma}.$$

*Proof.* First, define:

$$i_M := \max_i \mu_i \text{ and } i_m = \min_i \mu_i.$$

Consider the random variable:

$$Y = X_{i_M} - X_{i_m} \text{ for } X \sim \mathcal{N}(\mu, \Sigma).$$

We have from Lemma C.2:

$$\mathbb{E}[Y] = \mu_{i_M} - \mu_{i_m} \text{ and } \operatorname{Var}(Y) \leqslant \frac{400 \log(8/\gamma)}{\gamma^2} =: \sigma^2.$$

However, we also have:

$$\mathbb{P}\{Y \leqslant 0\} \geqslant \gamma \implies \mathbb{P}_{Z \sim \mathcal{N}(0,\sigma^2)}\{Z \leqslant -(\mu_{i_M} - \mu_{i_m})\} \geqslant \gamma.$$

Hence, we get from Lemma B.6:

$$\|\mu\|_\infty \leqslant \mu_{i_M} - \mu_{i_m} \leqslant \sigma\sqrt{2\log(1/\gamma)},$$

concluding the proof of the lemma. $\qquad\square$

First, let:

$$\Sigma_{ijk}^* = \widetilde{s}_{ijk}^2 \widetilde{\Sigma}_{ijk}^*, \text{ and } \mu_{ijk}^* = \widetilde{s}_{ijk} \widetilde{\mu}_{ijk}^*$$

be such that:

$$(1 - \widetilde{\varepsilon})\widetilde{\Sigma}_{ijk}^* \preccurlyeq \widehat{\Sigma}_{ijk} \preccurlyeq (1 + \widetilde{\varepsilon})\widetilde{\Sigma}_{ijk}^*$$
$$\|\widetilde{\mu}_{ijk}^* - \widehat{\mu}_{ijk}\| \leqslant \widetilde{\varepsilon}$$

guaranteed by Theorem B.1. Note that:

$$\forall l, m \in \{i, j, k\} : (1 - \varepsilon^\dagger)c_{lm}^\top \Sigma_{ijk}^* c_{lm} \leqslant c_{lm}^\top \overline{\Sigma}_{ijk} c_{lm} \leqslant (1 + \varepsilon^\dagger)c_{lm}^\top \Sigma_{ijk}^* c_{lm}.$$

As a consequence, we have:

$$(1 - \varepsilon^\dagger)\sum_{lm} c_{lm}^\top \Sigma_{ijk}^* c_{lm} = (1 - \varepsilon^\dagger)6\operatorname{Tr}(\Sigma_{ijk}^*) \leqslant \sum_{lm} c_{lm}^\top \overline{\Sigma}_{ijk} c_{lm} = 6\operatorname{Tr}(\overline{\Sigma}_{ijk})$$
$$\leqslant (1 + \varepsilon^\dagger)\sum_{lm} c_{lm}^\top \Sigma_{ijk}^* c_{lm} = (1 + \varepsilon^\dagger)6\operatorname{Tr}(\Sigma_{ijk}^*).$$

As a consequence, we get:

$$(1 - \varepsilon^\dagger) \operatorname{Tr}(\Sigma^*_{ijk}) \leqslant \operatorname{Tr}(\bar{\Sigma}_{ijk}) \leqslant (1 + \varepsilon^\dagger) \operatorname{Tr}(\Sigma^*_{ijk}).$$

Similarly, we get that:

$$(1 - \widetilde{\varepsilon}) \operatorname{Tr}(\widetilde{\Sigma}^*_{ijk}) \leqslant \operatorname{Tr}(\widehat{\Sigma}_{ijk}) \leqslant (1 + \widetilde{\varepsilon}) \operatorname{Tr}(\widetilde{\Sigma}^*_{ijk}).$$

Consequently, we get that:

$$(1 - 4\varepsilon^\dagger)\widetilde{s}^2_{ijk} = (1 - 4\varepsilon^\dagger)\frac{\operatorname{Tr}(\Sigma^*_{ijk})}{\operatorname{Tr}(\widetilde{\Sigma}^*_{ijk})} \leqslant s^2_{ijk} \leqslant (1 + 4\varepsilon^\dagger)\frac{\operatorname{Tr}(\Sigma^*_{ijk})}{\operatorname{Tr}(\widetilde{\Sigma}^*_{ijk})} = (1 + 4\varepsilon^\dagger)\widetilde{s}^2_{ijk}.$$

Hence, we obtain:

$$
\begin{aligned}
\|s_{ijk}\widehat{\mu}_{ijk} - \mu^*_{ijk}\| &\leqslant \|\widetilde{s}_{ijk}\widehat{\mu}_{ijk} - \mu^*_{ijk}\| + |s_{ijk} - \widetilde{s}_{ijk}|\|\widehat{\mu}_{ijk}\| \\
&\leqslant \|\widetilde{s}_{ijk}\widetilde{\mu}^*_{ijk} - \mu^*_{ijk}\| + \widetilde{s}_{ijk}\|\widetilde{\mu}^*_{ijk} - \widehat{\mu}_{ijk}\| + 4\varepsilon^\dagger \widetilde{s}_{ijk}\|\widehat{\mu}_{ijk}\| \\
&\leqslant \|\mu^*_{ijk} - \mu^*_{ijk}\| + \widetilde{s}_{ijk}\widetilde{\varepsilon} + 4\varepsilon^\dagger \widetilde{s}_{ijk}\|\widehat{\mu}_{ijk}\| \\
&\leqslant 0 + \widetilde{s}_{ijk}\varepsilon^\dagger(1 + 4\|\widehat{\mu}_{ijk}\|).
\end{aligned}
\tag{10}
$$

For the final term, we get from the guarantees of Theorem B.1 that:

$$\operatorname{Tr}(\widehat{\Sigma}_{ijk}) \geqslant \frac{1}{2} \implies \operatorname{Tr}(\widetilde{\Sigma}^*_{ijk}) \geqslant \frac{1}{4} \implies \widetilde{s}^2_{ijk} \leqslant \frac{1200 \log(8/\gamma)}{\gamma^2}$$

and consequently, we get from Lemmas C.2 and C.5:

$$\widetilde{s}_{ijk}\|\widehat{\mu}_{ijk}\| \leqslant \widetilde{s}_{ijk}(\|\widetilde{\mu}^*_{ijk}\| + \widetilde{\varepsilon}) = \|\mu^*_{ijk}\| + \widetilde{s}_{ijk}\widetilde{\varepsilon} \leqslant \frac{160 \log(8/\gamma)}{\gamma} + \frac{40\sqrt{\log(8/\gamma)}}{\gamma} \leqslant \frac{200 \log(8/\gamma)}{\gamma}.$$

Plugging this into Eq. (10), we get:

$$\|s_{ijk}\widehat{\mu}_{ijk} - \mu^*_{ijk}\| \leqslant \frac{1000 \log(8/\gamma)}{\gamma}\varepsilon^\dagger =: \varepsilon^\ddagger.$$

To conclude the proof of the lemma, let:

$$i_M := \arg\max_i(\overline{\mu}_i - \mu^*_i) \text{ and } i_m := \arg\min_i(\overline{\mu}_i - \mu^*_i).$$

Note now that:

$$\sum_{i=1}^n \overline{\mu}_i = 0 = \sum_{i=1}^n \mu^*_i \implies (\overline{\mu}_{i_M} - \mu^*_{i_M}) \geqslant 0 \text{ and } (\overline{\mu}_{i_m} - \mu^*_{i_m}) \leqslant 0.$$

Furthermore, there exists $k$ such that $i_M, i_m, k \in E'$. Therefore, we have by the triangle inequality:

$$
\begin{aligned}
\|\widehat{\mu}_{i_M,i_m,k} - \mu^*_{i_M,i_m,k}\| \leqslant 2\varepsilon^\ddagger \implies 4\varepsilon^\ddagger &\geqslant c^\top_{i_M,i_m}(\widehat{\mu}_{i_M,i_m,k} - \mu^*_{i_M,i_m,k}) \\
&= (\overline{\mu}_{i_M} - \mu^*_{i_M}) - (\overline{\mu}_{i_m} - \mu^*_{i_m}) \geqslant \|\overline{\mu} - \mu^*\|_\infty
\end{aligned}
$$

where the last inequality follows from the fact that each of the terms in the sum are positive and one of them is equal to $\|\overline{\mu} - \mu^*\|_\infty$. $\qquad\square$

*Proof of Theorem 5.2.* The theorem follows from Lemmas C.3 and C.4 by setting:

$$\widetilde{\varepsilon} := c\frac{\gamma^2\varepsilon}{\log(8/\gamma)\log(n)} \text{ and } \delta' = \frac{\delta}{n^2}$$

and applying Theorem B.1 with parameters $\widetilde{\varepsilon}$ and $\delta'$ for the elements $i, j, k \in E'$ from Lemma C.1. A union bound yields the corresponding guarantees on failure probability failure probability. $\qquad\square$

# D  PROOF OF LOWER BOUND - THEOREM 5.3

Here, we provide the proof of Theorem 5.3 restated below for convenience which shows that the guarantees of Theorem 5.2 are almost tight.

**Theorem 5.3.** *Let* $n \in \mathbb{N}$, $\delta, \varepsilon \in (0, 2^{-4})$. *For any estimator* $T$ *parameterized by* $\{((i,j,k), N_{ijk})\}_{i \neq j \neq k \neq i}$ *such that* $N_{i^*,j^*} \leqslant \varepsilon^{-2} \log(1/\delta)/4$ *for some* $i^* \neq j^* \in [n]$, *there exist two choice models, parameterized by* $(\mathbf{0}, \Sigma^1)$ *and* $(\mathbf{0}, \Sigma^2)$ *with:*

$$\mathrm{Tr}(\Sigma^1) = \mathrm{Tr}(\Sigma^2) = n \text{ and } \|\Sigma^1 - \Sigma^2\|_\infty \geqslant \varepsilon,$$

*satisfying:*

$$\max_{\Sigma \in \{\Sigma^1, \Sigma^2\}} \mathbb{P}_{\mathbf{X} \sim (T, \mathrm{Choice}(\mathbf{0}, \Sigma))} \left\{ \|T(\mathbf{X}) - \Sigma\|_\infty \geqslant \frac{\varepsilon}{4} \right\} \geqslant \delta.$$

*Proof.* Let $T$ be an estimator with $N_{i^*,j^*} \leqslant \varepsilon^{-2} \log(1/\delta)/4$ for some $i^* \neq j^*$. We define:

$$\Sigma^1 = \frac{n}{n-1}\left(I - \frac{\mathbf{1}\mathbf{1}^\top}{n}\right) I \left(I - \frac{\mathbf{1}\mathbf{1}^\top}{n}\right)$$

$$\Sigma^2 = \frac{n}{(n-1) - 2(\varepsilon/n)}\left(I - \frac{\mathbf{1}\mathbf{1}^\top}{n}\right)(I + \varepsilon e_{i^*} e_{j^*}^\top + \varepsilon e_{j^*} e_{i^*}^\top)\left(I - \frac{\mathbf{1}\mathbf{1}^\top}{n}\right).$$

From the following calculations:

$$\mathrm{Tr}(\Sigma^1) = \frac{n}{n-1}\,\mathrm{Tr}\left(\left(I - \frac{\mathbf{1}\mathbf{1}^\top}{n}\right) I \left(I - \frac{\mathbf{1}\mathbf{1}^\top}{n}\right)\right) = \frac{n}{n-1}\,\mathrm{Tr}\left(I - \frac{\mathbf{1}\mathbf{1}^\top}{n}\right) = n$$

$$\mathrm{Tr}(\Sigma^2) = \frac{n}{(n-1) - 2(\varepsilon/n)}\,\mathrm{Tr}\left(\left(I - \frac{\mathbf{1}\mathbf{1}^\top}{n}\right)(I + \varepsilon e_{i^*} e_{j^*}^\top + \varepsilon e_{j^*} e_{i^*}^\top)\left(I - \frac{\mathbf{1}\mathbf{1}^\top}{n}\right)\right)$$

$$= \frac{n}{(n-1) - 2(\varepsilon/n)}\,\mathrm{Tr}\left((I + \varepsilon e_{i^*} e_{j^*}^\top + \varepsilon e_{j^*} e_{i^*}^\top)\left(I - \frac{\mathbf{1}\mathbf{1}^\top}{n}\right)\right)$$

$$= \frac{n}{(n-1) - 2(\varepsilon/n)}\left(n - 1 + \varepsilon\,\mathrm{Tr}\left((e_{i^*} e_{j^*}^\top + e_{j^*} e_{i^*}^\top)\left(I - \frac{\mathbf{1}\mathbf{1}^\top}{n}\right)\right)\right)$$

$$= \frac{n}{(n-1) - 2(\varepsilon/n)}\left(n - 1 - \varepsilon\,\mathrm{Tr}\left((e_{i^*} e_{j^*}^\top + e_{j^*} e_{i^*}^\top)\left(\frac{\mathbf{1}\mathbf{1}^\top}{n}\right)\right)\right)$$

$$= \frac{n}{(n-1) - 2(\varepsilon/n)}\left(n - 1 - 2\frac{\varepsilon}{n}\right) = n,$$

we see that $\Sigma^1$ and $\Sigma^2$ are normalized. Furthermore, we have:

$\|\Sigma^2 - \Sigma^1\|_\infty$

$$\geqslant e_{i^*}^\top (\Sigma^2 - \Sigma^1) e_{j^*} = e_{i^*}^\top \Sigma^2 e_{j^*} - \frac{n}{n-1} e_{i^*}^\top \left(I - \frac{\mathbf{1}\mathbf{1}^\top}{n}\right) I \left(I - \frac{\mathbf{1}\mathbf{1}^\top}{n}\right) e_{j^*}$$

$$= e_{i^*}^\top \Sigma^2 e_{j^*} - \frac{n}{n-1} e_{i^*}^\top \left(I - \frac{\mathbf{1}\mathbf{1}^\top}{n}\right) e_{j^*} = e_{i^*}^\top \Sigma^2 e_{j^*} + \frac{1}{n-1}$$

$$= \frac{n}{(n-1) - 2(\varepsilon/n)} e_{i^*}^\top \left(I - \frac{\mathbf{1}\mathbf{1}^\top}{n}\right)(I + \varepsilon e_{i^*} e_{j^*}^\top + \varepsilon e_{j^*} e_{i^*}^\top)\left(I - \frac{\mathbf{1}\mathbf{1}^\top}{n}\right) e_{j^*} + \frac{1}{n-1}$$

$$= \frac{n}{(n-1) - 2(\varepsilon/n)} \left(-\frac{1}{n} + e_{i^*}^\top \left(I - \frac{\mathbf{1}\mathbf{1}^\top}{n}\right)(\varepsilon e_{i^*} e_{j^*}^\top + \varepsilon e_{j^*} e_{i^*}^\top)\left(I - \frac{\mathbf{1}\mathbf{1}^\top}{n}\right) e_{j^*}\right) + \frac{1}{n-1}$$

$$= \frac{n\varepsilon}{(n-1) - 2(\varepsilon/n)} \left(e_{i^*}^\top \left(I - \frac{\mathbf{1}\mathbf{1}^\top}{n}\right)(e_{i^*} e_{j^*}^\top + e_{j^*} e_{i^*}^\top)\left(I - \frac{\mathbf{1}\mathbf{1}^\top}{n}\right) e_{j^*}\right) - \frac{2\varepsilon/n}{(n-1)((n-1) - 2\varepsilon/n)}$$

$$= \frac{n\varepsilon}{(n-1) - 2(\varepsilon/n)} \left(\left(1 - \frac{1}{n}\right)^2 + \left(e_{i^*}^\top \left(I - \frac{\mathbf{1}\mathbf{1}^\top}{n}\right) e_{j^*}\right)^2\right) - \frac{2\varepsilon/n}{(n-1)((n-1) - 2\varepsilon/n)}$$

$$= \frac{\varepsilon(n^2 - 2n + 2)}{n((n-1) - 2(\varepsilon/n))} - \frac{2\varepsilon}{n(n-1)((n-1) - 2\varepsilon/n)} = \frac{\varepsilon(n(n-1)(n-2) + 2(n-1) - 2)}{n(n-1)((n-1) - 2\varepsilon/n)}$$

$$\geqslant \frac{\varepsilon n(n-1)(n-2)}{n(n-1)((n-1)-2\varepsilon/n)} \geqslant \frac{\varepsilon(n-2)}{(n-1-2\varepsilon/n)} \geqslant \frac{\varepsilon}{2}.$$

We will now compute the KL-divergence between $(T, \mathrm{Choice}(\mathbf{0}, \Sigma^1))$ and $(T, \mathrm{Choice}(\mathbf{0}, \Sigma^2))$. We have by the definition of $T$:

$$\mathrm{KL}((T, \mathrm{Choice}(\mathbf{0}, \Sigma^1)) \| (T, \mathrm{Choice}(\mathbf{0}, \Sigma^2))) = \mathrm{KL}\left(\prod_{ijk}\prod_{l=1}^{N_{ijk}} X_l^{i,j,k} \| \prod_{ijk}\prod_{l=1}^{N_{ijk}} Y_l^{i,j,k}\right)$$

where

$$X_l^{i,j,k} \sim \mathrm{Choice}((\mathbf{0}, \Sigma_{ijk}^1)) \text{ and } Y_l^{i,j,k} \sim \mathrm{Choice}((\mathbf{0}, \Sigma_{ijk}^2))$$

and from the tensorization of the KL-divergence, we have:

$$\mathrm{KL}((T, \mathrm{Choice}(\mathbf{0}, \Sigma^1)) \| (T, \mathrm{Choice}(\mathbf{0}, \Sigma^2))) = \sum_{ijk} \mathrm{KL}\left(\prod_{l=1}^{N_{ijk}} X_l^{i,j,k} \| \prod_{l=1}^{N_{ijk}} Y_l^{i,j,k}\right)$$

$$= \sum_{ijk} N_{ijk}\,\mathrm{KL}\left(X^{i,j,k} \| Y^{i,j,k}\right).$$

We now observe from the definitions of $\Sigma^1$ and $\Sigma^2$ for all $i, j, k$:

$$X^{i,j,k} \sim \mathrm{Choice}(\mathbf{0}, I)$$

$$Y^{i,j,k} \sim \mathrm{Choice}(\mathbf{0}, \Sigma^{i,j,k})$$

$$\text{where } \Sigma^{i,j,k} = \begin{cases} I & \text{if } \{i^*, j^*\} \not\subset \{i, j, k\} \\ I + \varepsilon(e_{i^*}e_{j^*}^\top + e_{j^*}e_{i^*}^\top) & \text{otherwise} \end{cases}$$

Since $\Sigma^{i,j,k} = I$ for all $i, j, k$ such that $\{i^*, j^*\} \not\subset \{i, j, k\}$, we get:

$$\mathrm{KL}((T, \mathrm{Choice}(\mathbf{0}, \Sigma^1)) \| (T, \mathrm{Choice}(\mathbf{0}, \Sigma^2))) = \sum_{\substack{i,j,k \\ \{i^*,j^*\}\subset\{i,j,k\}}} N_{ijk}\,\mathrm{KL}\left(X^{i,j,k} \| Y^{i,j,k}\right).$$

Now, we have by the data-processing inequality and utilizing the closed form solution for the KL-divergence between two multivariate Gaussians (Pardo, 2006, Exercise 11 in Section 1.6) for $W \sim \mathcal{N}(\mathbf{0}, I)$ and $Z \sim \mathcal{N}(0, \Sigma^{i,j,k})$:

$$\mathrm{KL}\left(X^{i,j,k} \| Y^{i,j,k}\right) \leqslant \mathrm{KL}\left(W \| Z\right) = \frac{1}{2}\left(\mathrm{Tr}(I \cdot \Sigma^{i,j,k} - I) + \log\left(\frac{\det(I)}{\det(\Sigma^{i,j,k})}\right)\right)$$

$$= \frac{1}{2}\left(0 + \log\left(\frac{1}{1-\varepsilon^2}\right)\right) \leqslant \varepsilon^2$$

for $\varepsilon \leqslant 1/2$. By substituting this back, we get:

$$\mathrm{KL}((T, \mathrm{Choice}(\mathbf{0}, \Sigma^1)) \| (T, \mathrm{Choice}(\mathbf{0}, \Sigma^2))) \leqslant \sum_{\substack{i,j,k \\ \{i^*,j^*\}\subset\{i,j,k\}}} N_{ijk}\,\mathrm{KL}\left(X^{i,j,k} \| Y^{i,j,k}\right)$$

$$\leqslant \sum_{\substack{i,j,k \\ \{i^*,j^*\}\subset\{i,j,k\}}} N_{ijk}\varepsilon^2 = N_{i^*,j^*}\varepsilon^2.$$

We now get as a consequence of the Bretagnolle–Huber inequality (Lemma E.2):

$$\mathrm{TV}((T, \mathrm{Choice}(\mathbf{0}, \Sigma^1)), (T, \mathrm{Choice}(\mathbf{0}, \Sigma^2))) \leqslant 1 - \frac{1}{2}\exp\left\{-N_{i^*,j^*}\varepsilon^2\right\} \leqslant 1 - \frac{1}{2}\sqrt{\delta} \leqslant 1 - 2\delta.$$

We now have by the minimax characterization of the Bayes risk:

$$\max_{\Sigma \in \{\Sigma^1, \Sigma^2\}} \mathbb{P}_{\boldsymbol{X} \sim (T, \mathrm{Choice}(\mathbf{0}, \Sigma))}\left\{\|T(\boldsymbol{X}) - \Sigma\|_\infty \geqslant \frac{\varepsilon}{4}\right\} \geqslant \frac{1}{2}(1 - (1 - 2\delta)) = \delta$$

by an application of Le Cam's method ((Wainwright, 2019, Section 15.2)), concluding the proof. $\quad\square$

# E  MISCELLANEOUS TECHNICAL RESULTS

Here, we include technical results used in the proofs of our theorems. We start with Hoeffding's inequality.

**Theorem E.1.** *Boucheron et al. (2013) Let $X_1, \ldots, X_n$ be independent random variables such that $X_i$ takes values in $[a_1, b_i]$ almost surely for all $i \leqslant n$. Let*

$$S = \sum_{i=1}^{n} (X_i - \mathbb{E}[X_i]).$$

*Then for every $t \geqslant 0$:*

$$\mathbb{P}\{S \geqslant t\} \leqslant \exp\left(-\frac{2t^2}{\sum_{i=1}^{n}(b_i - a_i)^2}\right).$$

We will make use of the following inequality which will be crucial in obtaining the high-probability lower bounds in the respective statements:

**Lemma E.2** ( Bretagnolle and Huber (1979); Tsybakov (2009); Canonne (2023)). *We have for any two distributions, $P$ and $Q$:*

$$\mathrm{TV}(P, Q) \leqslant 1 - \frac{1}{2}\exp(-\mathrm{KL}(P\|Q)).$$

## E.1  PROOF OF LEMMA 4.3

**Lemma 4.3.** *Let $v_1, v_2 \in \mathbb{R}^2$ be two independent unit vectors and $\xi \in \mathbb{R}^2$ be such that $d_1 := \langle v_1, \xi\rangle, d_2 := \langle v_2, \xi\rangle \geqslant 0$. Then,*

$$\alpha_{12} := \langle v_1, v_2\rangle$$

*is identifiable from $d_1, d_2$, and*

$$\gamma_{12} := \mathbb{P}\{\langle X, v_1\rangle, \langle X, v_2\rangle \leqslant 0\} \text{ for } X \sim \mathcal{N}(\xi, I).$$

*Proof.* Without loss of generality, due to the invariance of an isotropic Gaussian to to rotations and reflections, we may assume $d_1 \geqslant d_2$, $v_1 = e_2$, and $(v_2)_1 \geqslant 0$. Since $v_2$ is a unit vector, it may be parameterized as $v_2 = v(\theta^*)$ for $\theta^* \in [0, \pi]$ with $v(\theta)$ defined as follows:

$$v(\theta) := \begin{bmatrix} \sin(\theta) \\ -\cos(\theta) \end{bmatrix} \text{ for } \theta \in [0, \pi].$$

Now, $\xi(\theta)$ satisfying the first two constraints imposed by $d_1, d_2$ for the vectors $v_1$ and $v(\theta)$ is given by the following expression:

$$\xi(\theta) := \begin{bmatrix} \widetilde{\xi}(\theta) \\ d_1 \end{bmatrix} \text{ where } \widetilde{\xi}(\theta) := \frac{d_1\cos(\theta) + d_2}{\sin(\theta)}.$$

**Claim E.3.** *We have for $\theta \in (0, \pi)$:*

$$\widetilde{\xi}'(\theta) < 0.$$

*Proof.* Observe from the chain rule that:

$$\widetilde{\xi}'(\theta) = \frac{-\cos(\theta)(d_1\cos(\theta) + d_2)}{\sin^2\theta} - d_1 = \frac{-d_1 - d_2\cos(\theta)}{\sin^2(\theta)} < 0$$

for $\theta \in (0, \pi)$ concluding the proof of the claim. $\qquad\square$

Continuing with the proof, define:

$$\gamma(\theta) := \mathbb{P}\{\langle X, v_1\rangle, \langle X, v(\theta)\rangle \leqslant 0\} \text{ for } X \sim \mathcal{N}(\widetilde{\xi}(\theta), I).$$

Our next claim shows that $\gamma(\theta)$ is also monotonic in $\theta$.

**Claim E.4.** We have for $\theta \in (0, \pi)$:
$$\gamma'(\theta) > 0.$$

*Proof.* First, observe the following:
$$\{x : \langle v_1, x \rangle, \langle v(\theta), x \rangle \leq 0\} := \{(r\cos(\alpha), r\sin(\alpha)) : r \geq 0, \alpha \in (\pi, \pi + \theta)\}$$
$$\{x : \langle v_1, x \rangle, \langle v(\theta), x \rangle \leq 0\} := \{(x, y) : y \leq 0, x \leq y\cot(\theta)\}$$

As a consequence, we get by switching to polar coordinates:
$$\gamma(\theta) = \int_0^\infty \int_\pi^{\pi+\theta} \frac{1}{2\pi} \exp\left\{-\frac{(r\cos(\alpha) - \widetilde{\xi}(\theta))^2 + (r\sin(\alpha) - d_1)^2}{2}\right\} d\alpha r dr.$$

Hence, we get by the Leibniz integral rule:
$$\gamma'(\theta) = \int_0^\infty \frac{d}{d\theta}\left(\int_\pi^{\pi+\theta} \frac{1}{2\pi} \exp\left\{-\frac{(r\cos(\alpha) - \widetilde{\xi}(\theta))^2 + (r\sin(\alpha) - d_1)^2}{2}\right\} d\alpha\right) r dr$$

$$= \int_0^\infty \frac{1}{2\pi} \exp\left\{-\frac{(r\cos(\theta) + \widetilde{\xi}(\theta))^2 + (r\sin(\theta) + d_1)^2}{2}\right\} r dr +$$

$$\int_0^\infty \int_\pi^{\pi+\theta} \frac{\partial}{\partial\theta}\left(\frac{1}{2\pi} \exp\left\{-\frac{(r\cos(\alpha) - \widetilde{\xi}(\theta))^2 + (r\sin(\alpha) - d_1)^2}{2}\right\}\right) d\alpha r dr$$

$$= \int_0^\infty \frac{1}{2\pi} \exp\left\{-\frac{(r\cos(\theta) + \widetilde{\xi}(\theta))^2 + (r\sin(\theta) + d_1)^2}{2}\right\} r dr +$$

$$\int_0^\infty \int_\pi^{\pi+\theta} \frac{1}{2\pi} \exp\left\{-\frac{(r\cos(\alpha) - \widetilde{\xi}(\theta))^2 + (r\sin(\alpha) - d_1)^2}{2}\right\}(r\cos(\alpha) - \widetilde{\xi}(\theta))\widetilde{\xi}'(\theta) d\alpha r dr$$

$$= \int_0^\infty \frac{1}{2\pi} \exp\left\{-\frac{(r\cos(\theta) + \widetilde{\xi}(\theta))^2 + (r\sin(\theta) + d_1)^2}{2}\right\} r dr +$$

$$\widetilde{\xi}'(\theta) \int_0^\infty \int_\pi^{\pi+\theta} \frac{1}{2\pi} \exp\left\{-\frac{(r\cos(\alpha) - \widetilde{\xi}(\theta))^2 + (r\sin(\alpha) - d_1)^2}{2}\right\}(r\cos(\alpha) - \widetilde{\xi}(\theta)) d\alpha r dr.$$

For the second term, we have:
$$\int_0^\infty \int_\pi^{\pi+\theta} \frac{1}{2\pi} \exp\left\{-\frac{(r\cos(\alpha) - \widetilde{\xi}(\theta))^2 + (r\sin(\alpha) - d_1)^2}{2}\right\}(r\cos(\alpha) - \widetilde{\xi}(\theta)) d\alpha r dr$$

$$= \int_{-\infty}^0 \int_{-\infty}^{y\cot(\theta)} \frac{1}{2\pi} \exp\left\{-\frac{(x - \widetilde{\xi}(\theta))^2 + (y - d_1)^2}{2}\right\}(x - \widetilde{\xi}(\theta)) dx dy$$

$$= \int_{-\infty}^0 \int_\infty^{(y\cot(\theta) - \widetilde{\xi}(\theta))^2/2} \frac{1}{2\pi} \exp\left\{-t + \frac{(y - d_1)^2}{2}\right\} dt dy$$

$$= \frac{1}{2\pi} \int_{-\infty}^0 \exp\left\{-\frac{(y - d_1)^2}{2}\right\} \cdot (-\exp(-t))\Big|_\infty^{(y\cot(\theta) - \widetilde{\xi}(\theta))^2/2} dy$$

$$= \frac{-1}{2\pi} \int_{-\infty}^0 \exp\left\{-\frac{(y - d_1)^2}{2}\right\} \exp\left\{-\frac{(y\cot(\theta) - \widetilde{\xi}(\theta))^2}{2}\right\} dy.$$

Noting that the above integral is negative, the previous two displays and Claim E.3 now yield:
$$\gamma'(\theta) > 0 \text{ for } \theta \in (0, \pi)$$

concluding the proof of the claim. □

From, Claim E.4, we get that there exists a unique solution $\theta^*$ such that $\gamma(\theta^*) = \gamma$. Observing that $\langle v_1, v_2 \rangle = -\cos(\theta^*)$ establishes the lemma. □

E.2   PROOF OF THEOREM 3.2

Here, we present the proof of Theorem 3.2, stated below for convenience.

**Theorem 3.2.** *For any $n \geqslant 3$ and $\mu, \Sigma$ satisfying Assumption 3.1, there exists an infinite set $\mathcal{S}$:*

$$\forall i, j \in [n], \mu', \Sigma' \in \mathcal{S} : \underset{X \sim \mathcal{N}(\mu, \Sigma)}{\mathbb{P}} \{X_i \geqslant X_j\} = \underset{X \sim \mathcal{N}(\mu', \Sigma')}{\mathbb{P}} \{X_i \geqslant X_j\}.$$

*Proof.* We will establish the result for a mildly different normalization of the parameters, $(\mu, \Sigma)$. Formally, we will assume that:

$$\mu_1 = 0, \quad \Sigma_{1,1} = 0, \quad \text{Tr}(\Sigma) = n - 1 \qquad \text{(DIFF-FORM)}$$

We will establish the equivalence between DIFF-FORM and Assumption 3.1 subsequently. DIFF-FORM is motivated by the observation that for any $X \sim \mathcal{N}(\mu, \Sigma)$, $X'$ generated by the following transformation:

$$\forall i \in [n] : X'_i = t(X_i - X_1)$$

for any $t > 0$ induces the same set of rankings. This results in deterministically fixing the utility of the first item to 0 resulting in the first two constraints and the third is chosen for normalization. As before, we assume that the rank of $\Sigma$ is $n - 1$.

Now, consider any $(\mu, \Sigma)$ satisfying DIFF-FORM. Let $X \sim \mathcal{N}(\mu, \Sigma)$. First, observe that:

$$X_i - X_j \sim \mathcal{N}(\underbrace{\mu_i - \mu_j}_{\mu_{ij}}, \underbrace{\Sigma_{i,i} + \Sigma_{j,j} - 2\Sigma_{i,j}}_{\sigma_{ij}^2}). \qquad \text{(DIFF-PROB)}$$

Note that since $\Sigma$ has rank $n - 1$ (and satisfies $\Sigma e_1 = 0$), $\sigma_{ij} > 0$. We will now establish our results in two different cases:

**Case 1:** $\exists i, j > 1 : \mu_i = \mu_j$. In this case, observe that since $\Sigma$ is of rank $n - 1$, defining:

$$E = e_i e_j^\top + e_j e_i^\top$$

we have for some $\nu > 0$:

$$\forall \delta \in [0, \nu] : \Sigma + \delta E \text{ satisfies DIFF-FORM and has rank } n - 1.$$

Note, furthermore that for any $X' \sim \mathcal{N}(\mu, \Sigma + \delta E)$:

$$\forall k, l \in [n], k \neq l, \{k, l\} \neq \{i, j\} : \mathbb{P}\{X_k > X_l\} = \mathbb{P}\{X'_k > X'_l\}$$

$$\mathbb{P}\{X_i > X_j\} = \frac{1}{2} = \mathbb{P}\{X'_i > X'_j\}.$$

Hence,

$$\mathcal{S}' = \{(\mu, \Sigma + \delta E) : \delta \in [0, \nu]\}$$

consists of an infinite set of solutions with the same pairwise rankings.

**Case 2:** $\exists i > 1 : \mu_i = 0$. For this case, consider the matrix $\widetilde{\Sigma}$:

$$\widetilde{\Sigma}_{kl} = \begin{cases} \Sigma_{ii} + \nu & \text{if } k = l = i \\ \Sigma_{il} - \frac{\nu}{2} & \text{if } k = i \text{ and } 1 < l \neq i \\ \Sigma_{ki} - \frac{\nu}{2} & \text{if } l = i \text{ and } 1 < k \neq i \\ \Sigma_{kl} & \text{otherwise} \end{cases}.$$

Note that for small enough $\nu$, $\widetilde{\Sigma}$ remains positive-semidefinite with rank $n - 1$. Furthermore, $X \sim \mathcal{N}(\mu, \Sigma)$ and $X \sim \mathcal{N}(\mu, \widetilde{\Sigma})$ induce the same distribution over pairwise rankings of the $n$ alternatives. Finally, observe that $(\mu, \widetilde{\Sigma})$ satisfy the conditions of DIFF-FORM except for $\text{Tr}(\widetilde{\Sigma}) = n + \nu$. To restore this, we simply rescale the $(\mu, \widetilde{\Sigma})$ to obtain:

$$t = \frac{n + \nu}{n}, \quad \mu' = \frac{\mu}{\sqrt{t}}, \quad \Sigma' = \frac{\Sigma}{t}.$$

Now, $(\mu', \Sigma')$ induce the same distribution over pairwise preferences $(\mu, \Sigma)$. Furthermore, $(\mu', \Sigma')$ is distinct from $(\mu, \Sigma) - \mu \neq \mu'$ and $\Sigma \neq \Sigma'$. Since $(\mu', \Sigma')$ may be constructed for all small enough $\nu$, there exist infinitely many solutions satisfying DIFF-FORM with the same distribution over pairwise rankings.

**Case 3:**  $\forall i, j \in [n] : \mu_i \neq \mu_j$. In this setting, we start by modifying the parameter $\mu$ as follows:

$$u \neq 0, \|u\| \leqslant 1, u_1 = u_n = 0 : \widetilde{\mu} = \mu + \nu' u$$

for some $\nu'$ to be chosen subsequently. We will now construct an alternative solution from $\widetilde{\mu}$ by setting $\widetilde{\mu}_n$ and the entries of $\widetilde{\Sigma}$, the corresponding covariance matrix accordingly. Note from DIFF-PROB, to ensure that the pairwise ranking probabilities do not change, we must :

$$\forall i \in [2, n-1] : \widetilde{\Sigma}_{ii} = \left( \frac{\widetilde{\mu}_i}{\mu_i} \right)^2 \Sigma_{ii}$$

Next, we will set:

$$\widetilde{\Sigma}_{nn} = n - \sum_{i=2}^{n-1} \widetilde{\Sigma}_{ii} \text{ and } \widetilde{\mu}_n = \sqrt{\frac{\widetilde{\Sigma}_{nn}}{\Sigma_{nn}}} \mu_n.$$

Note that the above settings result in the correct pairwise probabilities as long as $\nu'$ is chosen small enough. From DIFF-PROB, this setting preserves the pairwise ranking probabilities between the first alternative and all other alternatives in the list. To ensure the correct pairwise ranking probabilities between other pairs, $i, j$, with $i \neq j$ and $i, j \neq 1$, we will set the values of $\Sigma_{ij}$ accordingly. To do this, observe from DIFF-PROB, it suffices to set $\Sigma_{ij}$ such that:

$$\frac{\mu_i - \mu_j}{\sqrt{\Sigma_{ii} + \Sigma_{jj} - 2\Sigma_{ij}}} = \frac{\widetilde{\mu}_i - \widetilde{\mu}_j}{\sqrt{\widetilde{\Sigma}_{ii} + \widetilde{\Sigma}_{jj} - 2\widetilde{\Sigma}_{ij}}}.$$

We now prove that $(\widetilde{\mu}, \widetilde{\Sigma})$ is a valid solution satisfying DIFF-FORM. The first three constraints are satisfied by construction. Furthermore, $(\widetilde{\mu}, \widetilde{\Sigma})$ achieve the same pairwise observation probabilities by DIFF-PROB. The only remaining constraint is to prove that $\widetilde{\Sigma}$ is positive semidefinite and has rank $n - 1$. To do this, note that $\sigma_{ij} > 0$ for all $i \neq j$, $\Sigma_{ii} > 0$ for all $i \in [n]$, $\mu_i \neq \mu_j$, and consequently, we have that:

$$\widetilde{\Sigma} = \Sigma + \nu E$$

where $E$ is symmetric, satisfies $E_{1i} = 0$ for all $i \in [n]$, and $\|E\| \leqslant 1$. Furthermore, $\nu$ may be made arbitrarily small by an appropriate choice of $\nu'$. Since the sub matrix corresponding to the last $n - 1$ rows and columns of $\Sigma$ is full rank and positive definite, the corresponding matrix of $\widetilde{\Sigma}$ is also full rank and positive definite by making $\nu$ correspondingly small. Hence, $\widetilde{\Sigma}$ is itself of rank $n - 1$ and positive semidefinite. Since such a solution may be constructed for arbitrary choices of $u$, this concludes the proof in this case as well.

The theorem under Assumption 3.1 is now a consequence of the following claim.

**Claim E.5.** Let

$$\mathcal{V} = \{(\mu, \Sigma) : (\mu, \Sigma) \text{ satisfy Assumption 3.1}\}$$
$$\mathcal{U} = \{(\mu, \Sigma) : (\mu, \Sigma) \text{ satisfy DIFF-FORM}\}.$$

Then, there exists an invertible mapping $f : \mathcal{U} \to \mathcal{V}$, such that for any $(\mu, \Sigma) \in \mathcal{U}$, $f(\mu, \Sigma)$ induces the same distribution over the rankings of the $n$ alternatives.

*Proof.* Letting:

$$M_U := I - \frac{\mathbf{1}\mathbf{1}^\top}{n} \text{ and } M_V := I - \mathbf{1}e_1^\top,$$

we will show that:

$$f(\mu, \Sigma) = \frac{M_U \mu}{\sqrt{t}}, \frac{M_U \Sigma M_U}{t} \text{ where } t = \frac{\mathrm{Tr}(M_U \Sigma M_U)}{n} \tag{MAP}$$

is the required transformation. First, observe:

$$M_V M_U = I - \mathbf{1}e_i^\top - \frac{\mathbf{1}\mathbf{1}^\top}{n} + \frac{\mathbf{1}\mathbf{1}^\top}{n} = I - \mathbf{1}e_i^\top$$

$$M_U M_V = I - \frac{\mathbf{1}\mathbf{1}^\top}{n} - \mathbf{1}e_1^\top + \mathbf{1}e_i^\top = I - \mathbf{1}\mathbf{1}^\top.$$

Consequently, we have:

$$\forall \mu, \Sigma \in \mathcal{U} : M_V M_U \Sigma M_U^\top M_V^\top = \Sigma$$

$$\forall \mu, \Sigma \in \mathcal{V} : M_U M_V \Sigma M_V^\top M_U^\top = \Sigma.$$

Hence, $t > 0$ in MAP and the mapping $f$ is well defined. Next, observe that for any $X \sim \mathcal{N}(\mu, \Sigma)$, $M_U X$ has distribution $\mathcal{N}(M_U \mu, M_U \Sigma M_U)$ and induces the same distribution over rankings. Therefore, $f(\mu, \Sigma)$ induces the same distribution over rankings as $(\mu, \Sigma)$. Finally, to show that $f$ is invertible, define $g : \mathcal{V} \to \mathcal{U}$:

$$g(\mu, \Sigma) = \frac{M_V \mu}{\sqrt{t}}, \frac{M_V \Sigma M_V^\top}{t} \text{ where } t = \frac{\text{Tr}(M_V \Sigma M_V^\top)}{n}. \qquad \text{(INV-MAP)}$$

Note that $g$ is well defined as $t > 0$ due to the definition of $M_U M_V$ and its operation on $(\mu, \Sigma) \in \mathcal{V}$. Similarly, $g(\mu, \Sigma)$ induces the same distribution over rankings as $(\mu, \Sigma)$. Finally, we have:

$$g \circ f(\mu, \Sigma)$$

$$= \frac{M_V M_U \mu}{\sqrt{t_1 t_2}}, \frac{M_V M_U \Sigma M_U^\top M_V}{t_1 t_2} \qquad \left( t_1 = \frac{\text{Tr}(M_U \Sigma M_U)}{n}, \ t_2 = \frac{\text{Tr}(M_V M_U \Sigma M_U^\top M_V^\top)}{n t_1} \right)$$

$$= \frac{M_V M_U \mu}{\sqrt{\text{Tr}(M_V M_U \Sigma M_U^\top M_V^\top)/n}}, \frac{M_V M_U \Sigma M_U^\top M_V}{\text{Tr}(M_V M_U \Sigma M_U^\top M_V^\top)/n}$$

$$= \frac{M_V M_U \mu}{\sqrt{\text{Tr}(\Sigma)/(n)}}, \frac{\Sigma}{\text{Tr}(\Sigma)/(n)} = \mu, \Sigma$$

where the third and fourth inequality follow from the fact that $(\mu, \Sigma) \in \mathcal{U}$. $\qquad \square$

Claim E.5 establishes that there exists a one-to-one mapping between pairs $(\mu, \Sigma)$ satisfying Assumption 3.1 and DIFF-FORM which induce the same distribution over rankings. Hence, identifiability under one parameterization implies identifiability under the other. Therefore, the existence of infinitely many solutions satisfying DIFF-FORM with the same pairwise ranking probabilities as any $(\mu, \Sigma) \in \mathcal{U}$ satisfying implies the existencec of infinitely many solutions satisfying Assumption 3.1 with the same pairwise ranking probabilities as any particlar $(\mu, \Sigma) \in \mathcal{V}$ concluding the proof of the theorem. $\qquad \square$

## F  IMPLEMENTATION DETAILS

Figure 6 shows multiple Gaussian distributions with the same pairwise probabilities.

We split the users into a train and validation sets (90% train, 10%validation) for testing. For the Netflix and MovieLens datasets, we only take movies with more than $n$ ratings, e.g., $n = 10^4$ for nf-10k. For datasets where we convert ratings to rankings, we assume that alternatives with the same rankings can be shuffled in the ranking. To sample the six alternatives, we first sample a user with probability proportional to the number of choices made, then we sample the choices uniformly, lastly, we sort the choices according to the ratings and query format. For the LLM experiments, we use "the movie database" (TMBD) to generate the input, the correctness of the TMBD ID, including mapping correctly to TV show or movie, was manually verified for both datasets. We omit alternatives that do not have clear entries in TMBD. Examples of such movies are double features like "The Fly (1986) / The Fly II (1989) double feature" were omitted.. We use a linear network for all experiments. For the probit experiments, we output the Cholesky decomposition of the covariance matrix. We use the Adam optimizer (Kingma, 2014) with a learning rate of 3e-4 for RUM and 0.01 otherwise. For the matrix factorization model we learn 128 embedding for users and alternatives with a per user and per alternative bias. We do 9000 gradient steps with a batch size of 1024 on RUMs and 1000 gradient steps with a batch size of 10240 on matrix factorization models. We train the matrix factorization models with the Huber loss.

For the probit models learned from 3-way comparisons, we project the problem in 2D using the $P$ matrix introduced in the main body of the text to both speed up the calculation, which is done via numerical integration over a coarse grid, and remove the symmetries present in the loss landscape. To calculate conditioned outcome probability, we use rejection sampling to measure the high-dimensional integrals instead of methods like Gessner et al. (2020).

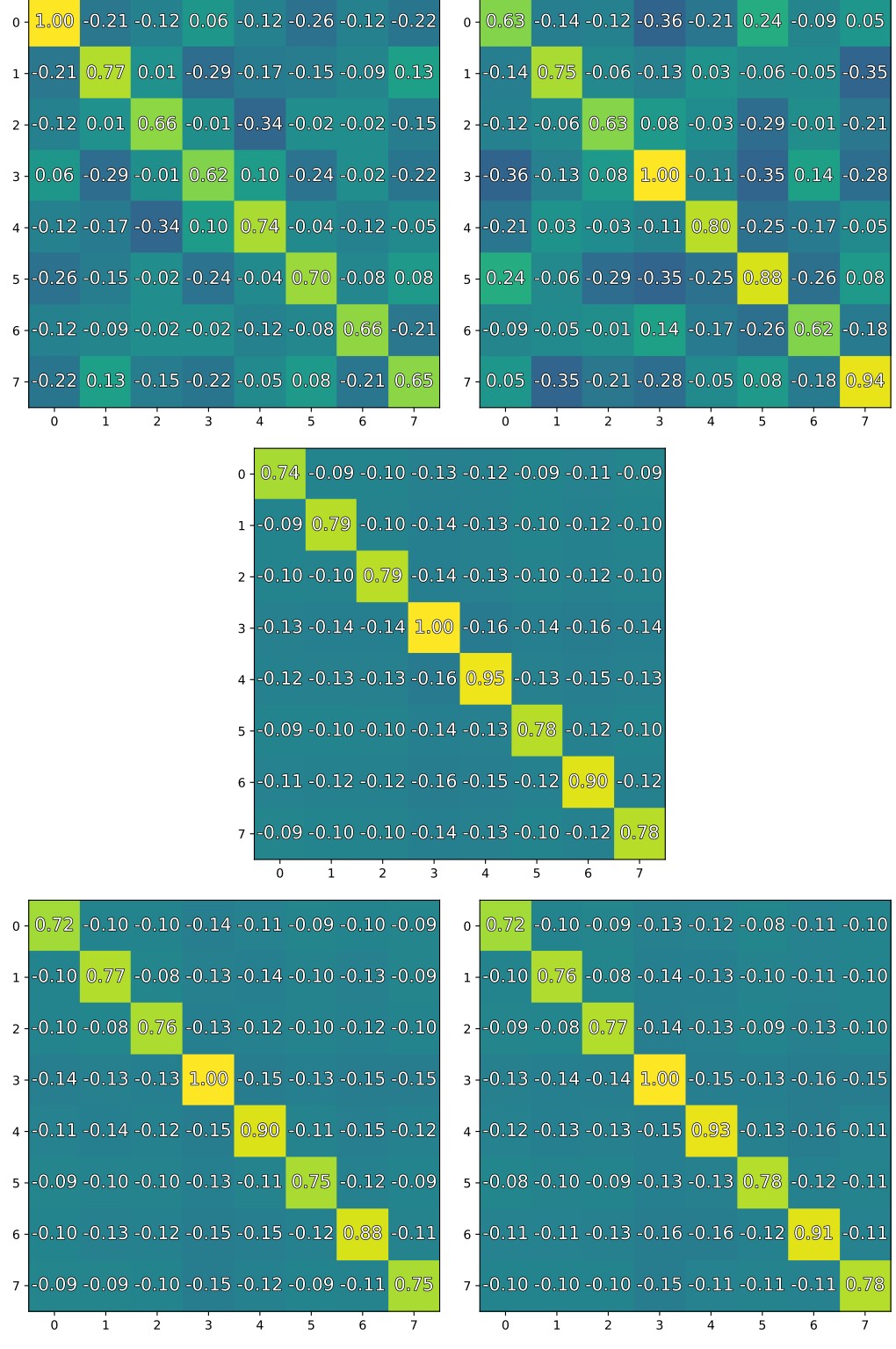

Figure 4: Larger version of the first row of Figure 2. Top to bottom rows: pairwise, ground truth, best-of-three.

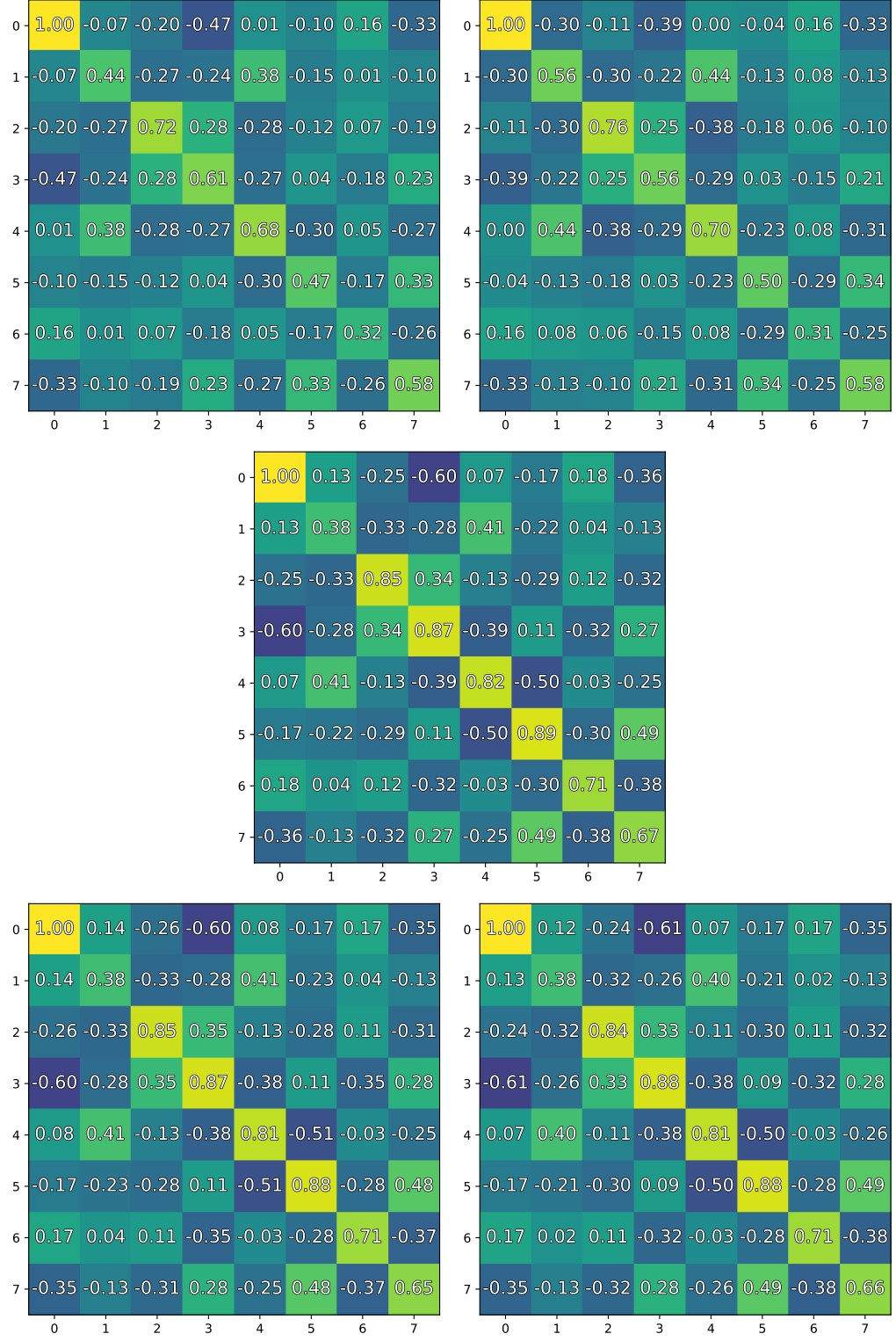

Figure 5: Larger version of the second row of Figure 2. Top to bottom rows: pairwise, ground truth, best-of-three.

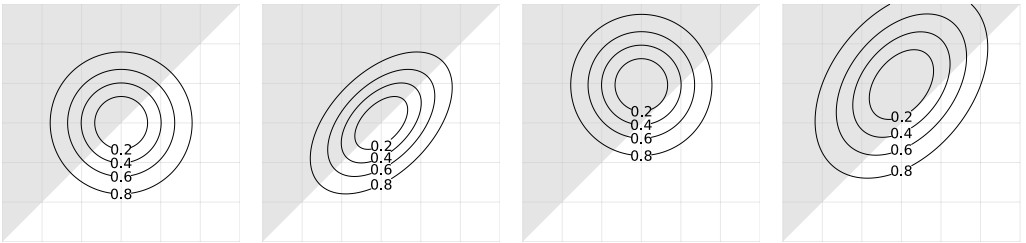

Figure 6: Two probit models where $\mathbb{P}(Y > X) = 0.5$ (two leftmost) and $\mathbb{P}(Y > X) = 0.75$ (two rightmost).

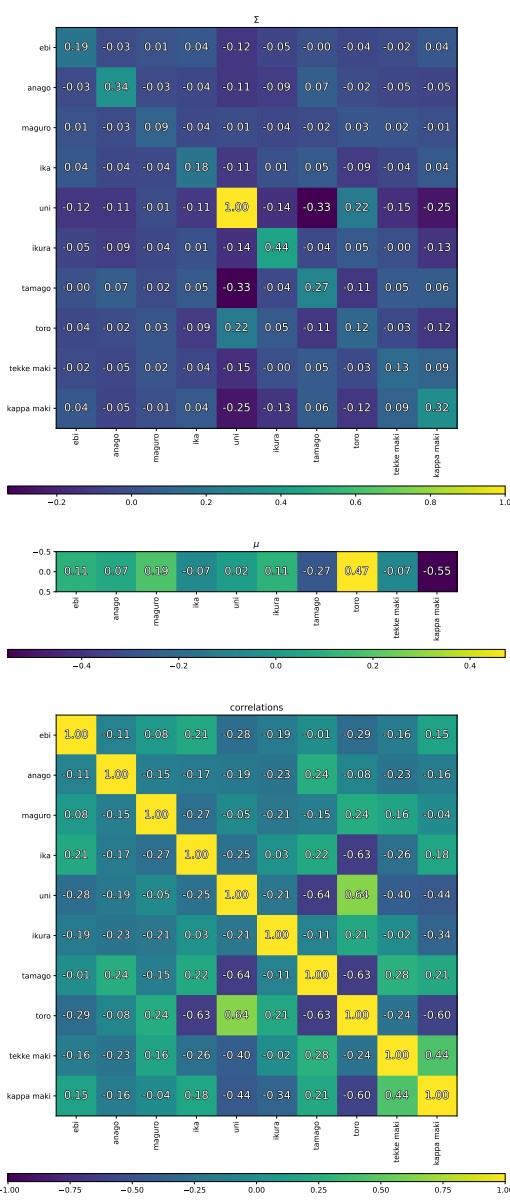

Figure 7: A probit model learned from best-of-three observation from sushi-a

| | | cor. | | | cor. |
|---|---|---|---|---|---|
| Independence Day (1996) | Lost in Translation (2003) | -0.45 | Gone in 60 Seconds (2000) | S.W.A.T. (2003) | 0.40 |
| Maid in Manhattan (2002) | The Royal Tenenbaums (2001) | -0.42 | Lost in Translation (2003) | Napoleon Dynamite (2004) | 0.40 |
| Miss Congeniality (2000) | The Matrix (1999) | -0.41 | Minority Report (2002) | Spider-Man 2 (2004) | 0.40 |
| Memento (2000) | Pearl Harbor (2001) | -0.41 | Dodgeball: A True Underdog Story (2004) | Starsky & Hutch (2004) | 0.40 |
| Double Jeopardy (1999) | Eternal Sunshine of the Spotless Mind (2004) | -0.40 | Kill Bill: Vol. 2 (2004) | Memento (2000) | 0.40 |
| Cheaper by the Dozen (2003) | Memento (2000) | -0.40 | Kill Bill: Vol. 1 (2003) | The Royal Tenenbaums (2001) | 0.40 |
| Independence Day (1996) | The Royal Tenenbaums (2001) | -0.40 | Lost in Translation (2003) | The Godfather (1972) | 0.40 |
| Anchorman: The Legend of Ron Burgundy (2004) | Speed (1994) | -0.40 | Braveheart (1995) | Cold Mountain (2003) | 0.40 |
| Adaptation (2002) | Pearl Harbor (2001) | -0.40 | Men in Black (1997) | Men in Black II (2002) | 0.40 |
| American Beauty (1999) | Maid in Manhattan (2002) | -0.39 | Enemy of the State (1998) | Men in Black II (2002) | 0.40 |
| Fight Club (1999) | Maid in Manhattan (2002) | -0.39 | Gone in 60 Seconds (2000) | The Recruit (2003) | 0.40 |
| Kill Bill: Vol. 1 (2003) | Maid in Manhattan (2002) | -0.39 | Runaway Bride (1999) | Stepmom (1998) | 0.40 |
| Air Force One (1997) | Eternal Sunshine of the Spotless Mind (2004) | -0.39 | Ghostbusters (1984) | Lord of the Rings: The Fellowship of the Ring (2001) | 0.40 |
| Fight Club (1999) | Sweet Home Alabama (2002) | -0.39 | Two Weeks Notice (2002) | What Women Want (2000) | 0.41 |
| Pearl Harbor (2001) | The Royal Tenenbaums (2001) | -0.38 | The Fast and the Furious (2001) | Tomb Raider (2001) | 0.41 |
| Lost in Translation (2003) | Pearl Harbor (2001) | -0.38 | Cheaper by the Dozen (2003) | The Notebook (2004) | 0.41 |
| Being John Malkovich (1999) | Runaway Bride (1999) | -0.38 | Speed (1994) | Top Gun (1986) | 0.41 |
| Adaptation (2002) | Independence Day (1996) | -0.37 | Adaptation (2002) | Pulp Fiction (1994) | 0.41 |
| Cheaper by the Dozen (2003) | Raiders of the Lost Ark (1981) | -0.37 | Steel Magnolias (1989) | You've Got Mail (1998) | 0.41 |
| Double Jeopardy (1999) | Kill Bill: Vol. 1 (2003) | -0.37 | Adaptation (2002) | Sideways (2004) | 0.41 |
| Men in Black II (2002) | Rain Man (1988) | -0.37 | How to Lose a Guy in 10 Days (2003) | The Wedding Planner (2001) | 0.41 |
| The Rock (1996) | The Royal Tenenbaums (2001) | -0.36 | Grease (1978) | Steel Magnolias (1989) | 0.41 |
| American Beauty (1999) | Pearl Harbor (2001) | -0.36 | Dirty Dancing (1987) | Sleepless in Seattle (1993) | 0.41 |
| Eternal Sunshine of the Spotless Mind (2004) | Gone in 60 Seconds (2000) | -0.36 | Memento (2000) | The Royal Tenenbaums (2001) | 0.42 |
| Dirty Dancing (1987) | Kill Bill: Vol. 1 (2003) | -0.36 | Adaptation (2002) | Memento (2000) | 0.42 |
| Miss Congeniality (2000) | The Royal Tenenbaums (2001) | -0.36 | The Silence of the Lambs (1991) | The Usual Suspects (1995) | 0.42 |
| Eternal Sunshine of the Spotless Mind (2004) | The General's Daughter (1999) | -0.36 | Grease (1978) | Pretty Woman (1990) | 0.42 |
| Ghostbusters (1984) | Stepmom (1998) | -0.36 | Radio (2003) | The Notebook (2004) | 0.42 |
| Cheaper by the Dozen (2003) | The Usual Suspects (1995) | -0.36 | Anchorman: The Legend of Ron Burgundy (2004) | Napoleon Dynamite (2004) | 0.42 |
| Fight Club (1999) | Two Weeks Notice (2002) | -0.35 | Pretty Woman (1990) | Runaway Bride (1999) | 0.43 |
| Kill Bill: Vol. 1 (2003) | The Rock (1996) | -0.35 | Hitch (2005) | How to Lose a Guy in 10 Days (2003) | 0.43 |
| Kill Bill: Vol. 2 (2004) | Pearl Harbor (2001) | -0.35 | Dodgeball: A True Underdog Story (2004) | Kill Bill: Vol. 1 (2003) | 0.43 |
| Fight Club (1999) | Radio (2003) | -0.35 | Mr. Deeds (2002) | Spider-Man (2002) | 0.43 |
| Eternal Sunshine of the Spotless Mind (2004) | Pearl Harbor (2001) | -0.34 | Along Came Polly (2004) | Sweet Home Alabama (2002) | 0.43 |
| Maid in Manhattan (2002) | Memento (2000) | -0.34 | Eternal Sunshine of the Spotless Mind (2004) | Pulp Fiction (1994) | 0.43 |
| Eternal Sunshine of the Spotless Mind (2004) | Maid in Manhattan (2002) | -0.34 | Kill Bill: Vol. 2 (2004) | Lord of the Rings: The Fellowship of the Ring (2001) | 0.44 |
| Good Will Hunting (1997) | National Treasure (2004) | -0.34 | Lord of the Rings: The Fellowship of the Ring (2001) | Lord of the Rings: The Return of the King (2003) | 0.44 |
| Hitch (2005) | Pulp Fiction (1994) | -0.34 | Eternal Sunshine of the Spotless Mind (2004) | The Royal Tenenbaums (2001) | 0.44 |
| Armageddon (1998) | Memento (2000) | -0.34 | Being John Malkovich (1999) | Lord of the Rings: The Fellowship of the Ring (2001) | 0.44 |
| How to Lose a Guy in 10 Days (2003) | Ray (2004) | -0.34 | Fight Club (1999) | The Silence of the Lambs (1991) | 0.44 |
| Hitch (2005) | Memento (2000) | -0.34 | Gone in 60 Seconds (2000) | The Fast and the Furious (2001) | 0.44 |
| Anchorman: The Legend of Ron Burgundy (2004) | Miss Congeniality (2000) | -0.34 | Fight Club (1999) | Memento (2000) | 0.45 |
| Adaptation (2002) | S.W.A.T. (2003) | -0.34 | Lord of the Rings: The Fellowship of the Ring (2001) | The Matrix (1999) | 0.45 |
| Pulp Fiction (1994) | Two Weeks Notice (2002) | -0.33 | American Beauty (1999) | Lord of the Rings: The Fellowship of the Ring (2001) | 0.45 |
| Being John Malkovich (1999) | National Treasure (2004) | -0.33 | Napoleon Dynamite (2004) | The Royal Tenenbaums (2001) | 0.45 |
| Pearl Harbor (2001) | Pulp Fiction (1994) | -0.33 | Napoleon Dynamite (2004) | Sideways (2004) | 0.45 |
| Ghostbusters (1984) | John Q (2001) | -0.33 | American Beauty (1999) | Lost in Translation (2003) | 0.45 |
| Armageddon (1998) | The Royal Tenenbaums (2001) | -0.33 | Maid in Manhattan (2002) | You've Got Mail (1998) | 0.45 |
| Eternal Sunshine of the Spotless Mind (2004) | Titanic (1997) | -0.33 | Anchorman: The Legend of Ron Burgundy (2004) | Starsky & Hutch (2004) | 0.45 |
| Indiana Jones and the Last Crusade (1989) | Maid in Manhattan (2002) | -0.33 | Sleepless in Seattle (1993) | You've Got Mail (1998) | 0.46 |
| Bringing Down the House (2003) | Memento (2000) | -0.33 | Kill Bill: Vol. 1 (2003) | Pulp Fiction (1994) | 0.46 |
| Double Jeopardy (1999) | The Matrix (1999) | -0.33 | Entrapment (1999) | Gone in 60 Seconds (2000) | 0.46 |
| Entrapment (1999) | Mean Girls (2004) | -0.33 | Memento (2000) | Pulp Fiction (1994) | 0.46 |
| Lord of the Rings: The Two Towers (2002) | The Recruit (2003) | -0.33 | Gone in 60 Seconds (2000) | Tomb Raider (2001) | 0.46 |
| Napoleon Dynamite (2004) | Pearl Harbor (2001) | -0.33 | Adaptation (2002) | The Royal Tenenbaums (2001) | 0.46 |
| The Patriot (2000) | The Royal Tenenbaums (2001) | -0.33 | Bringing Down the House (2003) | Cheaper by the Dozen (2003) | 0.46 |
| Lost in Translation (2003) | The Patriot (2000) | -0.33 | Maid in Manhattan (2002) | The Wedding Planner (2001) | 0.46 |
| Million Dollar Baby (2004) | The Day After Tomorrow (2004) | -0.32 | John Q (2001) | Men of Honor (2000) | 0.46 |
| Maid in Manhattan (2002) | Pulp Fiction (1994) | -0.32 | American Beauty (1999) | Napoleon Dynamite (2004) | 0.46 |
| Kill Bill: Vol. 2 (2004) | Maid in Manhattan (2002) | -0.32 | Harry Potter and the Chamber of Secrets (2002) | Harry Potter and the Sorcerer's Stone (2001) | 0.47 |
| Fight Club (1999) | The Wedding Planner (2001) | -0.32 | Steel Magnolias (1989) | Stepmom (1998) | 0.47 |
| American Beauty (1999) | Runaway Bride (1999) | -0.32 | Adaptation (2002) | American Beauty (1999) | 0.47 |
| Bringing Down the House (2003) | Kill Bill: Vol. 1 (2003) | -0.32 | Good Will Hunting (1997) | Rain Man (1988) | 0.47 |
| Stepmom (1998) | The Royal Tenenbaums (2001) | -0.32 | Lost in Translation (2003) | Sideways (2004) | 0.47 |
| Gone in 60 Seconds (2000) | Million Dollar Baby (2004) | -0.32 | Ferris Bueller's Day Off (1986) | Ghostbusters (1984) | 0.47 |
| Ghostbusters (1984) | S.W.A.T. (2003) | -0.32 | Braveheart (1995) | Independence Day (1996) | 0.47 |
| Hitch (2005) | Saving Private Ryan (1998) | -0.32 | Runaway Bride (1999) | Speed (1994) | 0.48 |
| Kill Bill: Vol. 1 (2003) | What Women Want (2000) | -0.32 | American Beauty (1999) | Eternal Sunshine of the Spotless Mind (2004) | 0.48 |
| Napoleon Dynamite (2004) | Twister (1996) | -0.32 | Sweet Home Alabama (2002) | Two Weeks Notice (2002) | 0.48 |
| Kill Bill: Vol. 1 (2003) | Miss Congeniality (2000) | -0.32 | Con Air (1997) | Gone in 60 Seconds (2000) | 0.49 |
| Lost in Translation (2003) | Maid in Manhattan (2002) | -0.32 | Adaptation (2002) | Eternal Sunshine of the Spotless Mind (2004) | 0.49 |
| National Treasure (2004) | The Godfather (1972) | -0.32 | Miss Congeniality (2000) | The Wedding Planner (2001) | 0.49 |
| American Beauty (1999) | John Q (2001) | -0.32 | American Beauty (1999) | Pulp Fiction (1994) | 0.49 |
| Bringing Down the House (2003) | Raiders of the Lost Ark (1981) | -0.32 | Maid in Manhattan (2002) | Two Weeks Notice (2002) | 0.49 |
| Ray (2004) | Sweet Home Alabama (2002) | -0.32 | Ocean's Eleven (2001) | Ocean's Twelve (2004) | 0.49 |
| Love Actually (2003) | Troy (2004) | -0.32 | Double Jeopardy (1999) | Entrapment (1999) | 0.50 |
| Lost in Translation (2003) | Pretty Woman (1990) | -0.32 | Fight Club (1999) | Pulp Fiction (1994) | 0.50 |
| Ghostbusters (1984) | Maid in Manhattan (2002) | -0.31 | Being John Malkovich (1999) | Eternal Sunshine of the Spotless Mind (2004) | 0.50 |
| Indiana Jones and the Last Crusade (1989) | S.W.A.T. (2003) | -0.31 | Lost in Translation (2003) | The Royal Tenenbaums (2001) | 0.50 |
| Monsters, Inc. (2001) | The Wedding Planner (2001) | -0.31 | Finding Nemo (Widescreen) (2003) | Monsters, Inc. (2001) | 0.50 |
| S.W.A.T. (2003) | The Royal Tenenbaums (2001) | -0.31 | Being John Malkovich (1999) | Memento (2000) | 0.51 |
| Double Jeopardy (1999) | Memento (2000) | -0.31 | Legally Blonde (2001) | Steel Magnolias (1989) | 0.52 |
| Mean Girls (2004) | The Rock (1996) | -0.31 | Ghost (1990) | Pretty Woman (1990) | 0.52 |
| Along Came a Spider (2001) | Memento (2000) | -0.31 | Maid in Manhattan (2002) | What Women Want (2000) | 0.53 |
| Million Dollar Baby (2004) | Tomb Raider (2001) | -0.31 | Maid in Manhattan (2002) | Sweet Home Alabama (2002) | 0.53 |
| Pearl Harbor (2001) | The Godfather (1972) | -0.31 | Pretty Woman (1990) | Stepmom (1998) | 0.54 |
| Armageddon (1998) | Eternal Sunshine of the Spotless Mind (2004) | -0.31 | Eternal Sunshine of the Spotless Mind (2004) | Memento (2000) | 0.54 |
| Men of Honor (2000) | The Royal Tenenbaums (2001) | -0.31 | Adaptation (2002) | Being John Malkovich (1999) | 0.56 |
| American Beauty (1999) | Swordfish (2001) | -0.31 | Spider-Man (2002) | Spider-Man 2 (2004) | 0.56 |
| Ghostbusters (1984) | Men of Honor (2000) | -0.31 | Adaptation (2002) | Lost in Translation (2003) | 0.57 |
| Finding Neverland (2004) | Speed (1994) | -0.31 | Kill Bill: Vol. 1 (2003) | Kill Bill: Vol. 2 (2004) | 0.57 |
| Big Daddy (1999) | Raiders of the Lost Ark (1981) | -0.31 | Fight Club (1999) | The Usual Suspects (1995) | 0.59 |
| Air Force One (1997) | Mystic River (2003) | -0.31 | Lord of the Rings: The Return of the King (2003) | Lord of the Rings: The Two Towers (2002) | 0.64 |
| Man on Fire (2004) | Raiders of the Lost Ark (1981) | -0.31 | Finding Nemo (Widescreen) (2003) | Shrek (Full-screen) (2001) | 0.65 |
| Hitch (2005) | The Usual Suspects (1995) | -0.31 | American Beauty (1999) | Being John Malkovich (1999) | 0.66 |
| Entrapment (1999) | The Royal Tenenbaums (2001) | -0.31 | Lord of the Rings: The Fellowship of the Ring (2001) | Lord of the Rings: The Two Towers (2002) | 0.66 |
| Indiana Jones and the Last Crusade (1989) | Sweet Home Alabama (2002) | -0.31 | Indiana Jones and the Temple of Doom (1984) | Raiders of the Lost Ark (1981) | 0.67 |
| Lost in Translation (2003) | Runaway Bride (1999) | -0.31 | Lord of the Rings: The Fellowship of the Ring (2001) | Lord of the Rings: The Two Towers (2002) | 0.66 |
| Pulp Fiction (1994) | The League of Extraordinary Gentlemen (2003) | -0.31 | Indiana Jones and the Temple of Doom (1984) | Raiders of the Lost Ark (1981) | 0.67 |
| Radio (2003) | The Usual Suspects (1995) | -0.31 | American Beauty (1999) | Memento (2000) | 0.69 |

Table 4: Left to right, movies with the lowest and highest correlations in a probit learned from the Netflix dataset.

| | | cor. | | | cor. |
|---|---|---|---|---|---|
| Independence Day (1996) | Lost in Translation (2003) | -0.45 | Gone in 60 Seconds (2000) | S.W.A.T. (2003) | 0.40 |
| Maid in Manhattan (2002) | The Royal Tenenbaums (2001) | -0.42 | Lost in Translation (2003) | Napoleon Dynamite (2004) | 0.40 |
| Miss Congeniality (2000) | The Matrix (1999) | -0.41 | Minority Report (2002) | Spider-Man 2 (2004) | 0.40 |
| Memento (2000) | Pearl Harbor (2001) | -0.41 | Dodgeball: A True Underdog Story (2004) | Starsky & Hutch (2004) | 0.40 |
| Double Jeopardy (1999) | Eternal Sunshine of the Spotless Mind (2004) | -0.40 | Kill Bill: Vol. 2 (2004) | Memento (2000) | 0.40 |
| Cheaper by the Dozen (2003) | Memento (2000) | -0.40 | Kill Bill: Vol. 1 (2003) | The Royal Tenenbaums (2001) | 0.40 |
| Independence Day (1996) | The Royal Tenenbaums (2001) | -0.40 | Lost in Translation (2003) | The Godfather (1972) | 0.40 |
| Anchorman: The Legend of Ron Burgundy (2004) | Speed (1994) | -0.40 | Braveheart (1995) | Cold Mountain (2003) | 0.40 |
| Adaptation (2002) | Pearl Harbor (2001) | -0.40 | Men in Black (1997) | Men in Black II (2002) | 0.40 |
| American Beauty (1999) | Maid in Manhattan (2002) | -0.39 | Enemy of the State (1998) | Men in Black II (2002) | 0.40 |
| Fight Club (1999) | Maid in Manhattan (2002) | -0.39 | The Recruit (2003) | Stepmom (1998) | 0.40 |
| Kill Bill: Vol. 1 (2003) | Maid in Manhattan (2002) | -0.39 | Runaway Bride (1999) | Stepmom (1998) | 0.40 |
| Air Force One (1997) | Eternal Sunshine of the Spotless Mind (2004) | -0.39 | Ghostbusters (1984) | Lord of the Rings: The Fellowship of the Ring (2001) | 0.40 |
| Fight Club (1999) | Sweet Home Alabama (2002) | -0.39 | Two Weeks Notice (2002) | What Women Want (2000) | 0.41 |
| Pearl Harbor (2001) | The Royal Tenenbaums (2001) | -0.38 | The Fast and the Furious (2001) | Tomb Raider (2001) | 0.41 |
| Lost in Translation (2003) | Pearl Harbor (2001) | -0.38 | Cheaper by the Dozen (2003) | The Notebook (2004) | 0.41 |
| Being John Malkovich (1999) | Runaway Bride (1999) | -0.38 | Speed (1994) | Top Gun (1986) | 0.41 |
| Adaptation (2002) | Independence Day (1996) | -0.37 | Adaptation (2002) | Pulp Fiction (1994) | 0.41 |
| Cheaper by the Dozen (2003) | Raiders of the Lost Ark (1981) | -0.37 | Steel Magnolias (1989) | You've Got Mail (1998) | 0.41 |
| Double Jeopardy (1999) | Kill Bill: Vol. 1 (2003) | -0.37 | Adaptation (2002) | Sideways (2004) | 0.41 |
| Men in Black II (2002) | Rain Man (1988) | -0.37 | How to Lose a Guy in 10 Days (2003) | The Wedding Planner (2001) | 0.41 |
| The Rock (1996) | The Royal Tenenbaums (2001) | -0.36 | Grease (1978) | Steel Magnolias (1989) | 0.41 |
| American Beauty (1999) | Pearl Harbor (2001) | -0.36 | Dirty Dancing (1987) | Sleepless in Seattle (1993) | 0.41 |
| Eternal Sunshine of the Spotless Mind (2004) | Gone in 60 Seconds (2000) | -0.36 | Memento (2000) | The Royal Tenenbaums (2001) | 0.42 |
| Dirty Dancing (1987) | Kill Bill: Vol. 1 (2003) | -0.36 | Adaptation (2002) | Memento (2000) | 0.42 |
| Miss Congeniality (2000) | The Royal Tenenbaums (2001) | -0.36 | The Silence of the Lambs (1991) | The Usual Suspects (1995) | 0.42 |
| Eternal Sunshine of the Spotless Mind (2004) | The General's Daughter (1999) | -0.36 | Grease (1978) | Pretty Woman (1990) | 0.42 |
| Ghostbusters (1984) | Stepmom (1998) | -0.36 | Radio (2003) | The Notebook (2004) | 0.42 |
| Cheaper by the Dozen (2003) | The Usual Suspects (1995) | -0.36 | Anchorman: The Legend of Ron Burgundy (2004) | Napoleon Dynamite (2004) | 0.42 |
| Fight Club (1999) | Two Weeks Notice (2002) | -0.35 | Pretty Woman (1990) | Runaway Bride (1999) | 0.43 |
| Kill Bill: Vol. 1 (2003) | The Rock (1996) | -0.35 | Hitch (2005) | How to Lose a Guy in 10 Days (2003) | 0.43 |
| Kill Bill: Vol. 2 (2004) | Pearl Harbor (2001) | -0.35 | Dodgeball: A True Underdog Story (2004) | Kill Bill: Vol. 1 (2003) | 0.43 |
| Fight Club (1999) | Radio (2003) | -0.35 | Fight Club (1999) | Kill Bill: Vol. 1 (2003) | 0.43 |
| Eternal Sunshine of the Spotless Mind (2004) | Pearl Harbor (2001) | -0.34 | Mr. Deeds (2002) | Spider-Man (2002) | 0.43 |
| Maid in Manhattan (2002) | Memento (2000) | -0.34 | Along Came Polly (2004) | Sweet Home Alabama (2002) | 0.43 |
| Eternal Sunshine of the Spotless Mind (2004) | Maid in Manhattan (2002) | -0.34 | Eternal Sunshine of the Spotless Mind (2004) | Pulp Fiction (1994) | 0.43 |
| Good Will Hunting (1997) | National Treasure (2004) | -0.34 | Kill Bill: Vol. 2 (2004) | Lord of the Rings: The Fellowship of the Ring (2001) | 0.44 |
| Hitch (2005) | Pulp Fiction (1994) | -0.34 | Lord of the Rings: The Fellowship of the Ring (2001) | Lord of the Rings: The Return of the King (2003) | 0.44 |
| Armageddon (1998) | Memento (2000) | -0.34 | Eternal Sunshine of the Spotless Mind (2004) | The Royal Tenenbaums (2001) | 0.44 |
| How to Lose a Guy in 10 Days (2003) | Ray (2004) | -0.34 | Being John Malkovich (1999) | Lord of the Rings: The Fellowship of the Ring (2001) | 0.44 |
| Hitch (2005) | Memento (2000) | -0.34 | Fight Club (1999) | The Silence of the Lambs (1991) | 0.44 |
| Anchorman: The Legend of Ron Burgundy (2004) | Miss Congeniality (2000) | -0.34 | Gone in 60 Seconds (2000) | The Fast and the Furious (2001) | 0.44 |
| Adaptation (2002) | S.W.A.T. (2003) | -0.34 | Fight Club (1999) | Memento (2000) | 0.45 |
| Pulp Fiction (1994) | Two Weeks Notice (2002) | -0.33 | Lord of the Rings: The Fellowship of the Ring (2001) | The Matrix (1999) | 0.45 |
| Being John Malkovich (1999) | National Treasure (2004) | -0.33 | American Beauty (1999) | Lord of the Rings: The Fellowship of the Ring (2001) | 0.45 |
| Pearl Harbor (2001) | Pulp Fiction (1994) | -0.33 | Napoleon Dynamite (2004) | The Royal Tenenbaums (2001) | 0.45 |
| Ghostbusters (1984) | John Q (2001) | -0.33 | Napoleon Dynamite (2004) | Sideways (2004) | 0.45 |
| Armageddon (1998) | The Royal Tenenbaums (2001) | -0.33 | American Beauty (1999) | Lost in Translation (2003) | 0.45 |
| Eternal Sunshine of the Spotless Mind (2004) | Titanic (1997) | -0.33 | Maid in Manhattan (2002) | You've Got Mail (1998) | 0.45 |
| Indiana Jones and the Last Crusade (1989) | Maid in Manhattan (2002) | -0.33 | Anchorman: The Legend of Ron Burgundy (2004) | Starsky & Hutch (2004) | 0.45 |
| Bringing Down the House (2003) | Memento (2000) | -0.33 | Sleepless in Seattle (1993) | You've Got Mail (1998) | 0.46 |
| Double Jeopardy (1999) | The Matrix (1999) | -0.33 | Kill Bill: Vol. 1 (2003) | Pulp Fiction (1994) | 0.46 |
| Entrapment (1999) | Mean Girls (2004) | -0.33 | Entrapment (1999) | Gone in 60 Seconds (2000) | 0.46 |
| Lord of the Rings: The Two Towers (2002) | The Recruit (2003) | -0.33 | Memento (2000) | Pulp Fiction (1994) | 0.46 |
| Napoleon Dynamite (2004) | Pearl Harbor (2001) | -0.33 | Gone in 60 Seconds (2000) | Tomb Raider (2001) | 0.46 |
| The Patriot (2000) | The Royal Tenenbaums (2001) | -0.33 | Adaptation (2002) | The Royal Tenenbaums (2001) | 0.46 |
| Lost in Translation (2003) | The Patriot (2000) | -0.33 | Bringing Down the House (2003) | Cheaper by the Dozen (2003) | 0.46 |
| Million Dollar Baby (2004) | The Day After Tomorrow (2004) | -0.32 | Maid in Manhattan (2002) | The Wedding Planner (2001) | 0.46 |
| Maid in Manhattan (2002) | Pulp Fiction (1994) | -0.32 | John Q (2001) | Men of Honor (2000) | 0.46 |
| Kill Bill: Vol. 2 (2004) | Maid in Manhattan (2002) | -0.32 | American Beauty (1999) | Napoleon Dynamite (2004) | 0.46 |
| Fight Club (1999) | The Wedding Planner (2001) | -0.32 | Harry Potter and the Chamber of Secrets (2002) | Harry Potter and the Sorcerer's Stone (2001) | 0.47 |
| American Beauty (1999) | Runaway Bride (1999) | -0.32 | Adaptation (2002) | American Beauty (1999) | 0.47 |
| Bringing Down the House (2003) | Kill Bill: Vol. 1 (2003) | -0.32 | Good Will Hunting (1997) | Rain Man (1988) | 0.47 |
| Stepmom (1998) | The Royal Tenenbaums (2001) | -0.32 | Lost in Translation (2003) | Sideways (2004) | 0.47 |
| Gone in 60 Seconds (2000) | Million Dollar Baby (2004) | -0.32 | Ghost (1990) | Grease (1978) | 0.47 |
| Ghostbusters (1984) | S.W.A.T. (2003) | -0.32 | Ferris Bueller's Day Off (1986) | Ghostbusters (1984) | 0.47 |
| Hitch (2005) | Saving Private Ryan (1998) | -0.32 | Braveheart (1995) | Independence Day (1996) | 0.47 |
| Kill Bill: Vol. 1 (2003) | What Women Want (2000) | -0.32 | Runaway Bride (1999) | Speed (1994) | 0.48 |
| Napoleon Dynamite (2004) | Twister (1996) | -0.32 | American Beauty (1999) | Eternal Sunshine of the Spotless Mind (2004) | 0.48 |
| Kill Bill: Vol. 1 (2003) | Miss Congeniality (2000) | -0.32 | Sweet Home Alabama (2002) | Two Weeks Notice (2002) | 0.48 |
| Lost in Translation (2003) | Maid in Manhattan (2002) | -0.32 | Con Air (1997) | Gone in 60 Seconds (2000) | 0.49 |
| National Treasure (2004) | The Godfather (1972) | -0.32 | Miss Congeniality (2000) | Eternal Sunshine of the Spotless Mind (2004) | 0.49 |
| American Beauty (1999) | John Q (2001) | -0.32 | American Beauty (1999) | Pulp Fiction (1994) | 0.49 |
| Bringing Down the House (2003) | Raiders of the Lost Ark (1981) | -0.32 | Ocean's Eleven (2001) | Ocean's Twelve (2004) | 0.49 |
| Ray (2004) | Sweet Home Alabama (2002) | -0.32 | Double Jeopardy (1999) | Entrapment (1999) | 0.50 |
| Love Actually (2003) | Troy (2004) | -0.32 | Fight Club (1999) | Pulp Fiction (1994) | 0.50 |
| Lost in Translation (2003) | Pretty Woman (1990) | -0.32 | Being John Malkovich (1999) | Eternal Sunshine of the Spotless Mind (2004) | 0.50 |
| Ghostbusters (1984) | Maid in Manhattan (2002) | -0.31 | Lost in Translation (2003) | The Royal Tenenbaums (2001) | 0.50 |
| Indiana Jones and the Last Crusade (1989) | S.W.A.T. (2003) | -0.31 | Finding Nemo (Widescreen) (2003) | Monsters, Inc. (2001) | 0.50 |
| Monsters, Inc. (2001) | The Wedding Planner (2001) | -0.31 | Being John Malkovich (1999) | Memento (2000) | 0.51 |
| S.W.A.T. (2003) | The Royal Tenenbaums (2001) | -0.31 | Legally Blonde (2001) | Steel Magnolias (1989) | 0.52 |
| Double Jeopardy (1999) | Memento (2000) | -0.31 | Ghost (1990) | Pretty Woman (1990) | 0.52 |
| Mean Girls (2004) | The Rock (1996) | -0.31 | Maid in Manhattan (2002) | What Women Want (2000) | 0.53 |
| Along Came a Spider (2001) | Memento (2000) | -0.31 | Maid in Manhattan (2002) | Sweet Home Alabama (2002) | 0.53 |
| Million Dollar Baby (2004) | Tomb Raider (2001) | -0.31 | Pretty Woman (1990) | Stepmom (1998) | 0.54 |
| Pearl Harbor (2001) | The Godfather (1972) | -0.31 | Eternal Sunshine of the Spotless Mind (2004) | Memento (2000) | 0.54 |
| Armageddon (1998) | Eternal Sunshine of the Spotless Mind (2004) | -0.31 | Adaptation (2002) | Being John Malkovich (1999) | 0.56 |
| Men of Honor (2000) | The Royal Tenenbaums (2001) | -0.31 | Spider-Man (2002) | Spider-Man 2 (2004) | 0.56 |
| American Beauty (1999) | Swordfish (2001) | -0.31 | Adaptation (2002) | Lost in Translation (2003) | 0.57 |
| Ghostbusters (1984) | Men of Honor (2000) | -0.31 | Kill Bill: Vol. 1 (2003) | Kill Bill: Vol. 2 (2004) | 0.57 |
| Finding Neverland (2004) | Speed (1994) | -0.31 | Fight Club (1999) | The Usual Suspects (1995) | 0.59 |
| Big Daddy (1999) | Raiders of the Lost Ark (1981) | -0.31 | Lord of the Rings: The Return of the King (2003) | Lord of the Rings: The Two Towers (2002) | 0.64 |
| Air Force One (1997) | Mystic River (2003) | -0.31 | Finding Nemo (Widescreen) (2003) | Shrek (Full-screen) (2001) | 0.65 |
| Man on Fire (2004) | Raiders of the Lost Ark (1981) | -0.31 | American Beauty (1999) | Being John Malkovich (1999) | 0.66 |
| Hitch (2005) | The Usual Suspects (1995) | -0.31 | Lord of the Rings: The Return of the King (2003) | Lord of the Rings: The Two Towers (2002) | 0.64 |
| Entrapment (1999) | The Royal Tenenbaums (2001) | -0.31 | Finding Nemo (Widescreen) (2003) | Shrek (Full-screen) (2001) | 0.65 |
| Indiana Jones and the Last Crusade (1989) | Sweet Home Alabama (2002) | -0.31 | American Beauty (1999) | Being John Malkovich (1999) | 0.66 |
| Lost in Translation (2003) | Runaway Bride (1999) | -0.31 | Lord of the Rings: The Fellowship of the Ring (2001) | Lord of the Rings: The Two Towers (2002) | 0.66 |
| Pulp Fiction (1994) | The League of Extraordinary Gentlemen (2003) | -0.31 | Indiana Jones and the Temple of Doom (1984) | Raiders of the Lost Ark (1981) | 0.67 |
| Radio (2003) | The Usual Suspects (1995) | -0.31 | American Beauty (1999) | Memento (2000) | 0.69 |

Table 5: Left to right, movies with the lowest and highest correlations in a probit learned from the MovieLens dataset.

| dds. | var. | feat. | logit accuracy quantile | | | matrix completion accuracy quantile | | | probit (pairwise) accuracy quantile | | | probit (best-of-three) accuracy quantile | | |
|---|---|---|---|---|---|---|---|---|---|---|---|---|---|---|
| | | | 0.25 | 0.50 | 0.75 | 0.25 | 0.50 | 0.75 | 0.25 | 0.50 | 0.75 | 0.25 | 0.50 | 0.75 |
| jokes | onehot | onehot | 0.61 | 0.61 | 0.61 | 0.61 | 0.61 | 0.61 | 0.59 | 0.59 | 0.59 | 0.61 | 0.61 | 0.61 |
| split-jokes | default | onehot | 0.55 | 0.55 | 0.56 | 0.54 | 0.54 | 0.54 | 0.53 | 0.53 | 0.56 | 0.54 | 0.54 | 0.55 |
| ml | 1k | onehot | 0.62 | 0.62 | 0.62 | 0.60 | 0.60 | 0.60 | 0.60 | 0.60 | 0.60 | 0.61 | 0.62 | 0.62 |
| split-ml | ml-1k | onehot | 0.63 | 0.63 | 0.63 | 0.59 | 0.59 | 0.59 | 0.61 | 0.61 | 0.61 | 0.62 | 0.62 | 0.63 |
| ml | 10k | onehot | 0.61 | 0.61 | 0.61 | 0.60 | 0.60 | 0.60 | 0.59 | 0.59 | 0.59 | 0.61 | 0.61 | 0.61 |
| split-ml | ml-10k | onehot | 0.62 | 0.62 | 0.63 | 0.59 | 0.59 | 0.59 | 0.60 | 0.60 | 0.60 | 0.62 | 0.62 | 0.62 |
| ml | 50k | onehot | 0.59 | 0.59 | 0.59 | 0.58 | 0.58 | 0.58 | 0.55 | 0.55 | 0.56 | 0.60 | 0.60 | 0.60 |
| split-ml | ml-50k | onehot | 0.61 | 0.61 | 0.61 | 0.56 | 0.56 | 0.56 | 0.57 | 0.57 | 0.57 | 0.61 | 0.61 | 0.61 |
| nf | 10k | onehot | 0.61 | 0.61 | 0.62 | 0.59 | 0.59 | 0.59 | 0.59 | 0.59 | 0.60 | 0.61 | 0.61 | 0.61 |
| split-nf | nf-10k | onehot | 0.64 | 0.64 | 0.64 | 0.60 | 0.60 | 0.60 | 0.62 | 0.62 | 0.62 | 0.63 | 0.64 | 0.64 |
| nf | 100k | onehot | 0.62 | 0.62 | 0.62 | 0.61 | 0.61 | 0.61 | 0.59 | 0.59 | 0.59 | 0.62 | 0.62 | 0.62 |
| split-nf | nf-100k | onehot | 0.64 | 0.64 | 0.65 | 0.61 | 0.62 | 0.62 | 0.62 | 0.62 | 0.62 | 0.64 | 0.64 | 0.64 |
| nf | 150k | onehot | 0.61 | 0.61 | 0.61 | 0.60 | 0.60 | 0.60 | 0.56 | 0.56 | 0.56 | 0.61 | 0.61 | 0.61 |
| split-nf | nf-150k | onehot | 0.64 | 0.64 | 0.64 | 0.60 | 0.61 | 0.61 | 0.62 | 0.62 | 0.62 | 0.63 | 0.63 | 0.63 |
| sushi | B | default | 0.66 | 0.66 | 0.67 | 0.67 | 0.67 | 0.68 | 0.63 | 0.64 | 0.65 | 0.65 | 0.65 | 0.67 |
| split-sushi | B | default | 0.63 | 0.63 | 0.64 | 0.54 | 0.54 | 0.54 | 0.56 | 0.57 | 0.58 | 0.60 | 0.61 | 0.62 |
| sushi | A | onehot | 0.65 | 0.65 | 0.65 | 0.67 | 0.67 | 0.67 | 0.64 | 0.65 | 0.65 | 0.67 | 0.67 | 0.67 |
| split-sushi | A | onehot | 0.64 | 0.64 | 0.65 | 0.51 | 0.53 | 0.54 | 0.58 | 0.59 | 0.59 | 0.61 | 0.62 | 0.63 |
| sushi | B | onehot | 0.67 | 0.67 | 0.67 | 0.67 | 0.67 | 0.67 | 0.65 | 0.66 | 0.66 | 0.67 | 0.67 | 0.67 |
| split-sushi | B | onehot | 0.62 | 0.62 | 0.63 | 0.53 | 0.54 | 0.54 | 0.57 | 0.58 | 0.58 | 0.58 | 0.60 | 0.60 |

Table 6: Rerun of the experiments in Table 2 with the additional limitation that each user can only supply 5 ratings. This small change tanks the performance of the matrix factorization as the model cannot accurately capture users.

