# OpenReview forum: "Learning Correlated Reward Models: Statistical Barriers and Opportunities"
_ICLR.cc/2026/Conference — ICLR 2026 Poster_

### Official Review · Reviewer_taUP · 2025-10-28

**Soundness:** 3
**Presentation:** 3
**Contribution:** 3
**Rating:** 6
**Confidence:** 2

**Summary:**

The paper demonstrates that pairwise preference data is insufficient to learn correlated utility models in Random Utility Models (RUMs), specifically the correlated probit model. It proves that best-of-three comparisons are both necessary and sufficient for identifiability, proposes a near-optimal estimator, and empirically validates its advantages over traditional models in capturing human preference correlations.

**Strengths:**

1. The paper is clearly written and logically organized, presenting formal identifiability theorems that rigorously establish the necessity and sufficiency of triplet comparisons for learning correlated probit models.

2. The paper delivers a persuasive critique of the Independence of Irrelevant Alternatives (IIA) assumption commonly used in RLHF, and introduces a well-justified alternative that enables more nuanced and personalized preference modeling.

3. The experimental results, though modest in some cases, effectively demonstrate the advantages of triplet-based modeling on real-world datasets (e.g., Netflix, MovieLens, Sushi).

**Weaknesses:**

While I am not a domain expert, I would like to offer a few comments based on my understanding of the paper. One notable limitation is the absence of experiments conducted in an actual RLHF setting (whether in reinforcement learning tasks or fine-tuning large language models). Although the theoretical and empirical results (i.e., the correlations) on general preference datasets are compelling, they may not fully demonstrate the practical value of the proposed approach. Without experiments in the RLHF context, it remains unclear how well the method performs in the scenarios it is ultimately intended to support.

**Questions:**

N/A

---

> ### Author Response · Authors · 2025-11-20
>
> We thank the reviewer for their thoughtful review.
>
> As we show, pairwise comparisons do not carry enough information for identifying the underlying utilities in our setting. However, most of the literature focuses explicitly on pairwise comparisons. Consequently, we were not able to find a suitable existing dataset matching the 3-way comparison structure that our theory calls for. Our primary goal in this work is to characterize when the utilities are identifiable. While releasing a large-scale 3-way RLHF dataset is an important and natural next step for the community, this is simply beyond the scope of the present paper.

---

### Official Review · Reviewer_GvjC · 2025-10-29

**Soundness:** 3
**Presentation:** 3
**Contribution:** 3
**Rating:** 4
**Confidence:** 3

**Summary:**

This paper focusses on learning correlated utility models—parameterized as a correlated probit model——to avoid the IIA assumptions required by RUMs. The paper first proves that pairwise comparison data is insufficient to identify the data generating model. The paper then proves that best-of-three observations are both sufficient and necessary to learn the data generating probit model, and provides finite sample guarantees. Finally, the paper presents a series of experiments evaluating the use of best-of-three observations to learn the parameters of a probit model for 3 real-world datasets and 1 synthetic dataset.

**Strengths:**

This paper explores an interesting question and provides substantive theoretical analysis to support their conclusion: pairwise comparisons are not sufficient to recover the parameters of a correlated choice model, but best-of-three comparisons are. This conclusion is interesting and, to the best of my knowledge, addresses an important gap in existing literature.

**Weaknesses:**

My main concerns are with the experiments in Section 6. Across all datasets, the proposed best-of-three-probit model matches or underperforms the direct matrix completion method. The authors claim that the direct matrix completion method is unrealistic in some scenarios—particularly when the set of alternatives and users is large—-but do not evaluate their method on those scenarios. Therefore, as far as I can tell, how their method also performs with larger alternatives/user sets remains an open question. Given the experimental evidence the authors do provide, there is no clear empirical benefit to using the best-of-three-probit model. I will raise my score if the authors can provide empirical evidence indicating where their best-of-three-probit model outperforms all other baselines.

Also, regarding Figure 3: the authors call this a “welfare maximizing experiment” but then note that the quantity they evaluate by “does not directly correlate with welfare as welfare is sensitive to the magnitude of utility change whereas this plot is not”. The authors should therefore change the name and how they discuss this experiment to avoid confusion.

**Questions:**

When does the best-of-three-probit model outperform the direct matrix completion method?

---

> ### Author Response · Authors · 2025-11-20
>
> > My main concerns are with the experiments in Section 6. Across all datasets, the proposed best-of-three-probit model matches or underperforms the direct matrix completion method.
>
> > particularly when the set of alternatives and users is large—-but do not evaluate their method on those scenarios.
>
> It is worth noting that the matrix completion method requires materializing every or many user scores. Consequently it is orders of magnitude slower than our gradient based method and the standard logit loss for inference. Exacerbating the computational difficulty, matrix completion requires learning per-user features during training, necessitating large user feature buffers. In our experiments, we had to increase the compute budget allocated for the matrix completion method compared to the other methods.
>
> > When does the best-of-three-probit model outperform the direct matrix completion method?
>
> The fundamental assumption of RUMs is that, for every agent, at any given time, there exists a vector of utilities over their preferences. Probits assume that these vectors are normal random variables. In that lens the matrix completion method removes that additional assumption so it includes the probit as a special case. We find it is reasonable that matrix completion methods do not under-perform compared to RUMs.
>
> The matrix completion method performing worse than the probit is a strong indicator that the scores were not learned properly. This learning issue can happen if there are not many observations per user, for instance when user data is anonymized and we only have 3-way comparisons. Or that there are not many users to begin with.
>
> > I will raise my score if the authors can provide empirical evidence indicating where their best-of-three-probit model outperforms all other baselines.
>
> This leaves us with the possibility of anonymizing and shrinking the dataset to hinder the learning of the score function. Concretely, we would resample triplets of ratings to create "virtual" users. Is this reasonable evidence?
>
> > Also, regarding Figure 3: the authors call this a “welfare maximizing experiment” but then note that the quantity they evaluate by “does not directly correlate with welfare as welfare is sensitive to the magnitude of utility change whereas this plot is not”. The authors should therefore change the name and how they discuss this experiment to avoid confusion.
>
> The "menus" are obtained via welfare maximization. The plots, however, are not obtained via welfare maximization as it is not a measurable quantity. Instead we assume that the welfare difference between preferences is always 1. This is a meaningful proxy but not the quantity itself. The sentence hints at the fact that these quantities do not need to have [linear] correlation.

---

> > ### Comment · Reviewer_GvjC · 2025-11-25
> >
> > If matrix completion is orders of magnitude slower than your proposed method, but matches its performance, then perhaps including an analysis of wall-clock-time or some other metric and framing the contribution in terms of improving speed or efficiency would strengthen the paper. Otherwise, by looking at the analysis in the paper, its not clear that your proposed method is better than matrix completion.
> >
> > > This leaves us with the possibility of anonymizing and shrinking the dataset to hinder the learning of the score function. Concretely, we would resample triplets of ratings to create "virtual" users. Is this reasonable evidence?
> >
> > I don't quite follow your proposed analysis, but in general if you can create realistic scenarios where matrix completion underperforms your method that would constitute reasonable evidence that your method is indeed better. Exploring a limited data-regime, where you shrink the dataset, sounds like an interesting experiment!

---

> > > ### Author Response · Authors · 2025-12-03
> > >
> > > | dataset     | variant  | feat    | logit 0.25 | logit 0.50 | logit 0.75 | mf 0.25 | mf 0.50 | mf 0.75 | probit2 0.25 | probit2 0.50 | probit2 0.75 | probit3 0.25 | probit3 0.50 | probit3 0.75 |
> > > |------------|----------|---------|------------|------------|------------|---------|---------|---------|--------------|--------------|--------------|--------------|--------------|--------------|
> > > | jokes      | onehot   | onehot  | 0.61       | 0.61       | 0.61       | 0.61    | 0.61    | 0.61    | 0.59         | 0.59         | 0.59         | 0.61         | 0.61         | 0.61         |
> > > | split-jokes| default  | onehot  | 0.55       | 0.55       | 0.56       | 0.54    | 0.54    | 0.54    | 0.53         | 0.53         | 0.56         | 0.54         | 0.54         | 0.55         |
> > > | ml         | 1k       | onehot  | 0.62       | 0.62       | 0.62       | 0.60    | 0.60    | 0.60    | 0.60         | 0.60         | 0.60         | 0.61         | 0.62         | 0.62         |
> > > | split-ml   | ml-1k    | onehot  | 0.63       | 0.63       | 0.63       | 0.59    | 0.59    | 0.59    | 0.61         | 0.61         | 0.61         | 0.62         | 0.62         | 0.63         |
> > > | ml         | 10k      | onehot  | 0.61       | 0.61       | 0.61       | 0.60    | 0.60    | 0.60    | 0.59         | 0.59         | 0.59         | 0.61         | 0.61         | 0.61         |
> > > | split-ml   | ml-10k   | onehot  | 0.62       | 0.62       | 0.63       | 0.59    | 0.59    | 0.59    | 0.60         | 0.60         | 0.60         | 0.62         | 0.62         | 0.62         |
> > > | ml         | 50k      | onehot  | 0.59       | 0.59       | 0.59       | 0.58    | 0.58    | 0.58    | 0.55         | 0.55         | 0.56         | 0.60         | 0.60         | 0.60         |
> > > | split-ml   | ml-50k   | onehot  | 0.61       | 0.61       | 0.61       | 0.56    | 0.56    | 0.56    | 0.57         | 0.57         | 0.57         | 0.61         | 0.61         | 0.61         |
> > > | nf         | 10k      | onehot  | 0.61       | 0.61       | 0.62       | 0.59    | 0.59    | 0.59    | 0.59         | 0.59         | 0.60         | 0.61         | 0.61         | 0.61         |
> > > | split-nf   | nf-10k   | onehot  | 0.64       | 0.64       | 0.64       | 0.60    | 0.60    | 0.60    | 0.62         | 0.62         | 0.62         | 0.63         | 0.64         | 0.64         |
> > > | nf         | 100k     | onehot  | 0.62       | 0.62       | 0.62       | 0.61    | 0.61    | 0.61    | 0.59         | 0.59         | 0.59         | 0.62         | 0.62         | 0.62         |
> > > | split-nf   | nf-100k  | onehot  | 0.64       | 0.64       | 0.65       | 0.61    | 0.62    | 0.62    | 0.62         | 0.62         | 0.62         | 0.64         | 0.64         | 0.64         |
> > > | nf         | 150k     | onehot  | 0.61       | 0.61       | 0.61       | 0.60    | 0.60    | 0.60    | 0.56         | 0.56         | 0.56         | 0.61         | 0.61         | 0.61         |
> > > | split-nf   | nf-150k  | onehot  | 0.64       | 0.64       | 0.64       | 0.60    | 0.61    | 0.61    | 0.62         | 0.62         | 0.62         | 0.63         | 0.63         | 0.63         |
> > > | sushi      | B        | default | 0.66       | 0.66       | 0.67       | 0.67    | 0.67    | 0.68    | 0.63         | 0.64         | 0.65         | 0.65         | 0.65         | 0.67         |
> > > | split-sushi| B        | default | 0.63       | 0.63       | 0.64       | 0.54    | 0.54    | 0.54    | 0.56         | 0.57         | 0.58         | 0.60         | 0.61         | 0.62         |
> > > | sushi      | A        | onehot  | 0.65       | 0.65       | 0.65       | 0.67    | 0.67    | 0.67    | 0.64         | 0.65         | 0.65         | 0.67         | 0.67         | 0.67         |
> > > | split-sushi| A        | onehot  | 0.64       | 0.64       | 0.65       | 0.51    | 0.53    | 0.54    | 0.58         | 0.59         | 0.59         | 0.61         | 0.62         | 0.63         |
> > > | sushi      | B        | onehot  | 0.67       | 0.67       | 0.67       | 0.67    | 0.67    | 0.67    | 0.65         | 0.66         | 0.66         | 0.67         | 0.67         | 0.67         |
> > > | split-sushi| B        | onehot  | 0.62       | 0.62       | 0.63       | 0.53    | 0.54    | 0.54    | 0.57         | 0.58         | 0.58         | 0.58         | 0.60         | 0.60         |

---

> ### Author Response · Authors · 2025-12-03
>
> We rerun the experiment with the caveat that for the datasets starting with `split`, we resample the choices of the users such that each user makes at most 3 choices in the training set and 6 in the validation set. This means that the test set is now less biased towards users with many ratings and that we throw away most of the data. The columns show the quantiles of the accuracy for different models. As you can see, in many cases, the performance of MF (matrix factorization) is now worse than that of the probit model. We argue that this difference is due to the fact that the matrix factorization model cannot learn user preferences because there is not enough information per user.
>
> Regarding wall-clock time, we make no claim that our implementation is optimized, and thus providing wall-clock time comparison is meaningless. For instance we are not using sparse gradients for the embedding in the MF implementation. Furthermore, the number of users required to estimate the probabilities depends on the distribution that is being modeled, whereas we chose the safe alternative of including all user. Nonetheless, the memory requirements of MF are O(alternatives + users) and the runtime is O(users * query) compared to O(alternative) for RUM (probit/logit/etc) and O(query) runtime where O(query) is a query- and model-dependent factor.

---

### Official Review · Reviewer_ZMzz · 2025-10-30

**Soundness:** 3
**Presentation:** 3
**Contribution:** 4
**Rating:** 6
**Confidence:** 3

**Summary:**

This paper studies RUMs and explores the statistical and computational challenges of learning a correlated probit model that avoids the IIA assumption.
The authors first prove that pairwise preference data is fundamentally insufficient to capture correlation among utilities.
They then show that best-of-three preference data is both identifiable and sufficient, and propose a statistically and computationally efficient estimator that achieves near-optimal performance.

**Strengths:**

I think that the contribution of this paper is significant, particularly Theorem 3.2, which rigorously proves that the classical pairwise comparison paradigm is fundamentally insufficient for recovering the parameters of a correlated probit model.
This finding challenges long-standing assumptions in choice modeling and clearly explains why existing methods fail to capture correlations in human preferences.
Moreover, the paper provides both theoretical and practical advances by establishing the first identifiability and finite-sample guarantees for correlated Random Utility Models.
Overall, the work offers a novel perspective on preference learning to the community.

**Weaknesses:**

Despite its strong theoretical contributions, the paper also has a few weaknesses and open questions.

- In Theorem 5.2, the sample complexity depends on $\gamma^{-24}$. Since $\gamma$ can be extremely small in practical settings, this dependence may lead to an unrealistic sample requirement. It would be important to discuss whether this exponent can be tightened or whether a refined analysis could yield a more favorable dependence on $\gamma$.

- While the paper motivates its setting through RLHF, it treats each item as a single (prompt, response) pair. In RLHF, however, the prompt space is effectively infinite and highly structured. It remains unclear how the proposed framework could be extended to handle such a large or continuous prompt and action space.

- I think this work is also highly related to general preference modeling frameworks, such as [1] and [2].
Could the authors compare their proposed framework with these prior approaches and clarify the key similarities and differences?

---
[1] Ye, Chenlu, et al. "Online iterative reinforcement learning from human feedback with general preference model." NeurIPS, 2024.
[2] Zhang, Yuheng, et al. "Iterative Nash Policy Optimization: Aligning LLMs with General Preferences via No-Regret Learning.", ICLR 2025.

**Questions:**

- How strong is Assumption 3.1? Could the authors elaborate on its necessity and implications?

---

> ### Author Response · Authors · 2025-11-20
>
> We thank the reviewer for their review.
>
> > Since can be extremely small in practical settings, this dependence may lead to an unrealistic sample requirement. It
>
> This is a worst-case bound, we argue that it reflects the difficult nature of problem. We are learning from discontinuous observations of random variables. Furthermore, the experimental results show that this is not a concern when using gradient-based methods.
>
> > While the paper motivates its setting through RLHF, it treats each item as a single (prompt, response) pair. In RLHF, however, the prompt space is effectively infinite and highly structured. It remains unclear how the proposed framework could be extended to handle such a large or continuous prompt and action space.
>
> We don't directly output the model $\mathcal{N}(\mu,\Sigma)$. Instead, for an alternative $i$, we output the average $\mu_i$ plus an embedding $\phi_i$. The mean $\mu$ is the concatenation of all of scalars $\mu_i$s. And $\Sigma_{ij}$ is $\phi_i^\top\phi_j$. Thus, our proposed solution does scale to large spaces as we only require materializing 3 alternatives to calculate the loss.
>
> > I think this work is also highly related to general preference modeling frameworks, such as [1] and [2]. Could the authors compare their proposed framework with these prior approaches and clarify the key similarities and differences?
>
> Our work is complementary to [1] and [2]. They start from a preference oracle $\mathbb{P}(a > b)$ (or samples of the oracle) and propose sampling from the Nash equilibrium of the game defined by this distribution. This leads to interesting sampling strategies. However, our focus is the identifiability of the correlated utilities from orderings of preferences. In principle, methods like [1] and [2] could use our correlated utilities as reward matrix.
>
> > How strong is Assumption 3.1? Could the authors elaborate on its necessity and implications?
>
> Assumption 3.1 is simply our way of choosing a "standard" representation. For instance, given the observation model of the paper, the distributions $\mathcal{N}(\mu,\Sigma)$ and $\mathcal{N}(1 + \mu,\Sigma)$ are equivalent as they have the same preference distribution. However, when converted to the form that Assumtption 3.1 expects, via the transformation $X:=t(X - \sum_i X_i)$ for some $t$, both distributions will match.

---

> > ### Comment · Reviewer_ZMzz · 2025-11-27
> >
> > Thank you for the response. I’ll keep my positive score.

---

### Official Review · Reviewer_AY4G · 2025-10-30

**Soundness:** 3
**Presentation:** 3
**Contribution:** 3
**Rating:** 4
**Confidence:** 4

**Summary:**

This paper considers a correlated probit model of preferences that avoids the Independence of Irrelevant alternatives (IIA) assumption; for this model, the paper shows that pairwise preference data isn't sufficient for provably learning the correlations, and suggests the use of three way preference data to provably solve for the parameters of this model.

**Strengths:**

1. A very clearly written paper, a concrete model / setup and clear theoretical results.
2. Interesting observations, and a particularly relevant problem in the current scheme of RLHF based pipelines for training foundation models.

**Weaknesses:**

1. In principle, this is a fairly stylized model and its applicability to realistic setups, particularly in RLHF is questionable in the sense of how useful it can be compared to optimizing the standard pairwise loss.
2. This specific connection to training improved reward models (in RLHF) is not explored as part of the empirical evaluations which could've helped bolster the results offered by this paper.

**Questions:**

One question that I am interested in thinking about (and getting the authors to weigh in on) is what does this imply for policy learning in RLHF setups -- in particular, I am thinking about situations involving intransitive preferences.

---

> ### Author Response · Authors · 2025-11-20
>
> We thank the reviewer for their thoughtful review.
>
> > In principle, this is a fairly stylized model ...
>
> The probit model is less stylized than the pairwise loss, as the pairwise loss is equivalent to a logit model, which assumes independent utilities (e.g., via i.i.d. Gumbel errors and the resulting independence of irrelevant alternatives). In contrast, the probit model allows for a flexible covariance structure over utilities.
>
> > This specific connection to training improved reward models
>
> As we show, pairwise comparisons do not carry enough information for identifying the underlying utilities in our setting. However, most of the literature focuses explicitly on pairwise comparisons. Consequently, we were not able to find a suitable existing dataset matching the 3-way comparison structure that our theory calls for. Our primary goal in this work is to characterize when the utilities are identifiable. While releasing a large-scale 3-way RLHF dataset is an important and natural next step for the community, this is simply beyond the scope of the present paper.
>
> > One question that I am interested in thinking about
>
> We interpret the question as asking how preferences would be modeled in an RLHF setting. Our method is directly applicable to LLMs. Intuitively, instead of learning a single scalar reward for each dialogue, we learn a Gaussian reward structure across alternatives, with a covariance that captures similarity between dialogues.
>
> Concretely, instead of outputting a single scalar for a given dialogue, the reward model would output, for alternative $i$, a scalar $\mu_i$ and a vector $\phi_i$. These are used to reconstruct the probit over multiple dialogues, where the covariance of the rewards of two dialogues is the dot product of their features, i.e., $\Sigma_{ij} = \phi_i^\top \phi_j. $
>
> In a first step, the alternative with the largest mean $\mu_i$ will be the “best” alternative. In a second step, bundles or “menus” can be created similarly to those in the welfare-maximization experiment. These bundles will be naturally diverse and capture different alternatives that users may find interesting, and such menus can be directly presented to users. Lastly, if a user states their preferences for items in the menus, then in subsequent interactions these preferences can be used to condition the probit model for that specific user, similar to how we predicted preferences in the Netflix dataset. This yields a natural procedure for RLHF systems: construct diverse menus, collect user feedback, and then personalize the reward model by conditioning on the user’s stated preferences

---

### Author Response · Authors · 2025-12-03

We thank the reviewers for their feedback.

We would like to summarize here the main points made during the rebuttal:

Re: lack of RLHF experiments. This paper highlights a fundamental shortcoming of existing datasets. As we show, we cannot learn probit models from pairwise data. As far as we can tell, no RLHF dataset satisfies our requirements, and the reviewers did not point us toward such a dataset. We argue that, given this is the first work to both show this impossibility and provide a solution, curating a dataset is outside the scope of this work.

Re: complexity bounds. We argue that the complexity bound is reflective of the hardness of the task at hand, as we are learning from discontinuous functions (argmax and sorting) applied to random variables. These concerns are also addressed by the proposed gradient-based algorithm and experiments.

Re: performance relative to matrix factorization. We argue that this is not a reasonable objection for the following reason. The matrix factorization model is model-free and does not benefit from the additional assumptions of the probit model, such as rational substitution patterns (increasing the mean of an alternative should never decrease its probability), yet the MF model does not provide any interpretable knob for altering the model. Furthermore, the matrix factorization model covers the space of all possible RUMs, so the objection is, in some capacity, similar to objecting to the fact that for some tasks, neural networks match the performance of model-based regression models. Lastly, as we have shown in our reply to reviewer GvjC, the data requirements of MF are very different from RUMs. Using MF is not appropriate if the dataset does not provide much per-user data. Similarly, the computational requirements of MF are very different: we need to store per-user features and perform an expensive integration at test time over many users.

---

### Public Comment · ~Nihar_B_Shah1 · 2026-06-13

Very cool paper!

In Section 3, it says "model with X ~ N (µ, Σ)". Do you mean to say that when each (comparison/triplet) sample is obtained, it is based on an *independent* draw of vector X from this distribution? Or are dependencies across samples assumed/allowed in any way?

Also, fyi, somewhat related (focusing on triplets):
- "Crowdsourcing Feature Discovery via Adaptively Chosen Comparisons" James Zou, Kamalika Chaudhuri, Adam Kalai
- "Adaptively Learning the Crowd Kernel" Omer Tamuz, Ce Liu, Serge Belongie, Ohad Shamir, Adam Kalai

---

### Meta-Review · Area_Chair_igHu · 2026-01-07

**Summary:**

This paper studies how to learn correlated reward (or preference) models in RUMs, with motivation from RLHF and preference-based learning.
The main theoretical message appears to be timely and important: pairwise comparisons are fundamentally insufficient for identifying correlation structure in correlated probit models, even after standard normalization.
The paper then shows that moving to three-way (best-of-three) comparisons resolves this identifiability issue, and proposes an estimator with provable finite-sample guarantees.
Matching lower bounds demonstrate that the statistical difficulty characterized by the theory is largely unavoidable.

The contribution is primarily theoretical and clarificatory.
The paper explains why existing pairwise-based approaches cannot, in principle, recover correlated utilities, and what type of feedback is required to do so.
Empirical results on several real-world preference datasets are included to illustrate the behavior of the estimator and to compare against baselines.

**Reviewer Concerns:**

While the motivation is framed around RLHF, the method fundamentally relies on three-way comparisons, whereas most existing RLHF datasets and systems use pairwise preferences. The rebuttal argues convincingly that this reflects a limitation of current datasets rather than the theory.

Reviewers raised concerns about the strong dependence of the sample complexity on an observability parameter. The authors explain that this reflects worst-case hardness due to discontinuous observations and that practical performance can be much better, but the gap between worst-case theory and realistic regimes may still exist.

There was a concern about weak experiments, particularly, on whether there is any "clear empirical benefit to using the best-of-three-probit model"
The authors added experiments in a limited data regime, which helps clarify when the method is useful. Still, more thorough experiments  would strengthen the paper (while the paper's contribution is mainly theoretical.)

Some dependence on the sample complexity is extremely large (e.g., $\gamma^{-24}$), making the results far from practical benefits.

**Reviewer Scores:**

One reviewer explicitly stated they would keep their positive score (6).

Reviewers who were already positive but cautious would likely remain unchanged, as their main concern (lack of direct RLHF validation) was acknowledged.

Reviewers who were borderline negative indicated they might raise their score if clearer empirical regimes favoring the proposed method were shown. The added limited-data experiments partially address this. It is reasonable to assume that some of these reviewers would update/increase their scores.

Overall, I believe this paper is interesting and relevant to the research community.

---

### Decision · Program_Chairs · 2026-01-26

Accept (Poster)